# What is the Relationship between Tensor Factorizations and Circuits (and How Can We Exploit it)?

**Lorenzo Loconte**⚡                                    *l.loconte@sms.ed.ac.uk*
*School of Informatics, University of Edinburgh, UK*

**Antonio Mari**⚡🔵                                    *antonio.mari@epfl.ch*
*École Polytechnique Fédérale de Lausanne (EPFL), Switzerland*

**Gennaro Gala**⚡                                    *g.gala@tue.nl*
*Eindhoven University of Technology, NL*

**Robert Peharz**                                    *robert.peharz@tugraz.at*
*Graz University of Technology, Austria*

**Cassio de Campos**                                    *c.decampos@tue.nl*
*Eindhoven University of Technology, NL*

**Erik Quaeghebeur**                                    *e.quaeghebeur@tue.nl*
*Eindhoven University of Technology, NL*

**Gennaro Vessio**                                    *gennaro.vessio@uniba.it*
*Computer Science Department, University of Bari Aldo Moro, IT*

**Antonio Vergari**                                    *avergari@ed.ac.uk*
*School of Informatics, University of Edinburgh, UK*

**Reviewed on OpenReview:** *https://openreview.net/forum?id=Y7dRmpGiHj*

**Code repository:**🖥 *https://github.com/april-tools/uni-circ-le*

## Abstract

This paper establishes a rigorous connection between circuit representations and tensor factorizations, two seemingly distinct yet fundamentally related areas. By connecting these fields, we highlight a series of opportunities that can benefit both communities. Our work generalizes popular tensor factorizations within the circuit language, and unifies various circuit learning algorithms under a single, generalized hierarchical factorization framework. Specifically, we introduce a modular "Lego block" approach to build tensorized circuit architectures. This, in turn, allows us to systematically construct and explore various circuit and tensor factorization models while maintaining tractability. This connection not only clarifies similarities and differences in existing models, but also enables the development of a comprehensive pipeline for building and optimizing new circuit/tensor factorization architectures. We show the effectiveness of our framework through extensive empirical evaluations, and highlight new research opportunities for tensor factorizations in probabilistic modeling.

---

⚡Shared first authorship.

🔵Work partly done when visiting the University of Edinburgh.

🖥The codebase for this paper is based on ten-pcs, an older version of the currently maintained cirkit package.

# 1 Introduction

This paper aims at bridging two apparently distant, but in fact intimately related fields: *circuit representations* (Darwiche & Marquis, 2002; Choi et al., 2020; Vergari et al., 2021) and *tensor factorizations* (Kolda, 2006; Sidiropoulos et al., 2017). Specifically, we establish a formal connection between the two representations and show how the latter can bring a unified perspective on the many learning algorithms devised to learn the former, as well as create research opportunities for both communities.

Tensors are multidimensional generalizations of matrices that are extensively used to represent high-dimensional data (Kroonenberg, 2007). Tensor factorizations are well-understood mathematical objects to compactly represent tensors in terms of simple operations acting on lower-dimensional tensors (Kolda, 2006). They have been extensively applied in ML and AI, e.g., in computer vision (Vasilescu & Terzopoulos, 2002; Savas & Eldén, 2007; Panagakis et al., 2021), graph analysis (Kolda et al., 2005), computational neuroscience (Vos et al., 2007; Tresp et al., 2021), neuro-symbolic AI (Nickel et al., 2015; Balazevic et al., 2019; Gema et al., 2023; Loconte et al., 2023), language modeling (Ma et al., 2019; Hu et al., 2022; Xu et al., 2023), and as ways to encode probability distributions (Jaini et al., 2018b; Novikov et al., 2021; Amiridi et al., 2022; Hood & Schein, 2024). While usually defined in terms of *shallow factorizations*, tensor factorizations can be also expressed as a hierarchy of factorizations (Grasedyck, 2010), sometimes represented in the graphical formalism of tensor networks (Orús, 2013; Biamonte & Bergholm, 2017; Glasser et al., 2019).

Circuit representations (Darwiche & Marquis, 2002; Choi et al., 2020; Vergari et al., 2021), on the other hand, are structured computational graphs introduced in the context of logical reasoning and probabilistic modeling (Darwiche, 2003; Poon & Domingos, 2011; Kisa et al., 2014). *Probabilistic circuits* (PCs) (Vergari et al., 2019b; Choi et al., 2020), in particular, are circuits that encode tractable probability distributions. They support a number of applications requiring exact and efficient inference routines, e.g., lossless compression (Liu et al., 2022), biomedical generative modeling (Dang et al., 2022b), reliable neuro-symbolic AI (Ahmed et al., 2022; Loconte et al., 2023) and constrained text generation (Zhang et al., 2023). Many algorithms to learn PCs from data have been proposed in the past (see e.g., Sidheekh & Natarajan (2024) for a review), with one paradigm emerging: building *overparameterized* circuits, comprising millions or even billions of parameters (Liu et al., 2023a; Gala et al., 2024a), and training these parameters by gradient-ascent, expectation-maximization (Peharz et al., 2016; 2020c), or regularized variants (Dang et al., 2022a).

Both hierarchical tensor factorizations and PCs have been introduced as alternative representations of probabilistic graphical models (Song et al., 2013; Robeva & Seigal, 2017; Glasser et al., 2020; Bonnevie & Schmidt, 2021), and the connection between certain circuits and factorizations has been hinted in some works (Jaini et al., 2018b; Glasser et al., 2019). However, they mainly differ in how they are applied: tensor factorizations are usually used in tasks where a ground-truth tensor to approximate is available or a dimensionality reduction problem can be formulated (aka *tensor sketch*), whereas PCs are usually learned from data in the same spirit generative models are trained. Similar to tensor factorizations, however, modern PC representations are overparameterized and usually encoded as a collection of tensors as to leverage parallelism and modern deep learning frameworks (Vergari et al., 2019a; Peharz et al., 2020c; Mari et al., 2023). This begs the question: is there any formal and systematic connection between circuits and tensor factorizations? Our answer is affirmative, as we show that *a circuit can be cast as a generalized sparse hierarchical tensor factorization*, where its parameters encode the lower-dimensional tensors of the factorization itself. Or alternatively, *a hierarchical tensor factorization is a special case of a deep circuit with a particular tensorized architecture*. When it comes to PCs, this implies decomposing probability distributions represented as non-negative tensors (Cichocki & Phan, 2009). At the same time, classical tensor factorizations can be exactly encoded as (shallow) circuits. By affirming the duality of tensor factorizations and circuits, we systematize previous results in the literature, open up new perspectives in representing and learning circuits, and suggest possible ways to construct new and extend existing (probabilistic) factorizations.

Specifically, in this paper we will first derive a compact way to denote several tensorized circuit architectures, and represent them as computational graphs using a *"Lego blocks"* approach that stacks (locally) dense tensor factorizations while preserving the structural properties of circuits required for tractability. This enables us to use novel "blocks" in a plug-and-play manner. Then, we unify the many different algorithms for learning PCs that have been proposed in the literature so far (Peharz et al., 2020c;a; Liu & Van den Broeck,

2021b), which come from different perspectives and yield circuits that are considered as different models. In particular, we show that their differences reduce to factorizations and syntactic transformations of their tensor parameters, since they can be understood under the same generalized (hierarchical) factorization based on the Tucker tensor factorization (Tucker, 1966) and its specializations (Kolda & Bader, 2009). Therefore, we argue the different performances that are often reported in the literature are actually the result of different hyperparameters and learning methods more than different inductive biases (Liu et al., 2023b).

Furthermore, after making this connection, we exploit tensor factorizations to further compress the parameters of modern PC architectures already represented in tensor format. By doing so, we introduce PCs that are more parameter-efficient than previous ones, and we show that finding the best circuit architecture for a certain setting is far from solved. Lastly, we highlight how this connection with circuits can spawn interesting research opportunities for the tensor factorization community—highlighted as boxes throughout the paper—ranging from learning to decompose tensors from data, to interpreting tensor factorizations as latent-variable probabilistic models, to inducing sparsity via the specification of background knowledge.

**Contributions.** **i)** We generalize popular tensor factorization methods and their hierarchical formulation into the language of *circuits* (Section 2). **ii)** We connect PCs to non-negative tensor factorizations and highlight how the latter can be interpreted as latent variable models, and as such they can be used as generative models and for neuro-symbolic AI (Section 3). **iii)** Within our framework, we abstract away the many options used to build and learn modern overparameterized architectures to arrive at a general algorithmic pipeline (Section 4) to represent and learn hierarchical tensor factorizations as tensorized circuits. **iv)** This allows us to analyze how existing, different parameterizations of circuits are related to each other by leveraging tensor factorizations, while proposing more parameter-efficient modeling choices that retain some of the expressiveness (Section 5). **v)** We evaluate several algorithmic choices in our framework on a wide range of distribution estimation tasks, highlighting the major trade-offs in terms of time and space complexity, and resulting performance (Section 6).

## 2 From Tensor Factorizations to Circuits

**Symbols notation.** We will adapt most of the notation and nomenclature from Kolda & Bader (2009). We denote sets of random variables with $\mathbf{X}$, $\mathbf{Y}$ and $\mathbf{Z}$, and we use $[n]$ to express the set $\{1, 2, \ldots, n\}$ with $n > 0$. The domain of a variable $X$ is denoted as $\mathsf{dom}(X)$, and we denoted as $\mathsf{dom}(\mathbf{X}) = \mathsf{dom}(X_1) \times \cdots \times \mathsf{dom}(X_n)$ the joint domain of variables $\mathbf{X} = \{X_i\}_{i=1}^n$. We denote scalars with lower-case letters (e.g., $a \in \mathbb{R}$), vectors with boldface lower-case letters (e.g., $\mathbf{a} \in \mathbb{R}^N$), matrices with boldface upper-case letters (excluding those used for variables, e.g., $\mathbf{A} \in \mathbb{R}^{M \times N}$), and tensors with boldface calligraphic letters (e.g., $\boldsymbol{\mathcal{A}} \in \mathbb{R}^{I_1 \times I_2 \times I_3}$). Moreover, we use subscripts to denote entries of tensors (e.g., $a_{ijk}$ is the $(i, j, k)$-th entry in $\boldsymbol{\mathcal{A}}$).

**Matrix and tensor operations notation.** We make use of ":" to denote tensor slicing (e.g., $\mathbf{A}_{:j:} \in \mathbb{R}^{I_1 \times I_3}$ is obtained by selecting the $j$-th matrix slice of $\boldsymbol{\mathcal{A}}$ along the second dimension). Furthermore, we denote with $\odot$ the Hadamard (or element-wise product) of tensors having the same dimensions, and we denote with $\circ$ the outer products of vectors, i.e., given $\mathbf{u} \in \mathbb{R}^M, \mathbf{v} \in \mathbb{R}^N$ we have that their outer product $\mathbf{A} = \mathbf{u} \circ \mathbf{v} \in \mathbb{R}^{M \times N}$ is defined such that $a_{ij} = u_i v_j$ for all $(i, j) \in [M] \times [N]$. We denote with $||$ the concatenation operator over vectors, i.e., $\mathbf{u} \,||\, \mathbf{v} = [u_1, \ldots, u_M, v_1, \ldots, v_N]^\top \in \mathbb{R}^{M+N}$. We use $\otimes$ to express the Kronecker product between vectors, i.e., $\mathbf{u} \otimes \mathbf{v} \in \mathbb{R}^{MN}$ is the *row-wise* flattening of $\mathbf{u} \circ \mathbf{v}$ into an $MN$-dimensional vector. Finally, we use $\times_n$ to denote the tensor-matrix dot product along the $n$-th dimension, i.e., given a tensor $\boldsymbol{\mathcal{T}} \in \mathbb{R}^{I_1 \times \cdots \times I_d}$ and a matrix $\mathbf{A} \in \mathbb{R}^{J \times I_n}$, $n \in [d]$, then we have that $\boldsymbol{\mathcal{T}} \times_n \mathbf{A} \in \mathbb{R}^{I_1 \times \cdots \times I_{n-1} \times J \times I_{n+1} \cdots \times I_d}$ is defined in element-wise notation as $(\boldsymbol{\mathcal{T}} \times_n \mathbf{A})_{i_1 \cdots i_{n-1} j\, i_{n+1} \cdots i_d} = \sum_{i_n=1}^{I_n} t_{i_1 \cdots i_d} a_{j i_n}$, with $j \in [J]$.

### 2.1 Shallow Tensor Factorizations are Shallow Circuits

**Tucker tensor factorization.** Tensor factorizations *approximate* high-dimensional tensors by a collection of lower-dimensional ones. Formally, given a tensor $\boldsymbol{\mathcal{T}} \in \mathbb{R}^{I_1 \times \cdots \times I_d}$, whose size grows exponentially with respect to the dimensions $d$, we seek a low-rank factorization for it (Kroonenberg, 2007). Many popular tensor factorization methods, such as the *canonical polyadic* decomposition (CP) (Carroll & Chang, 1970),

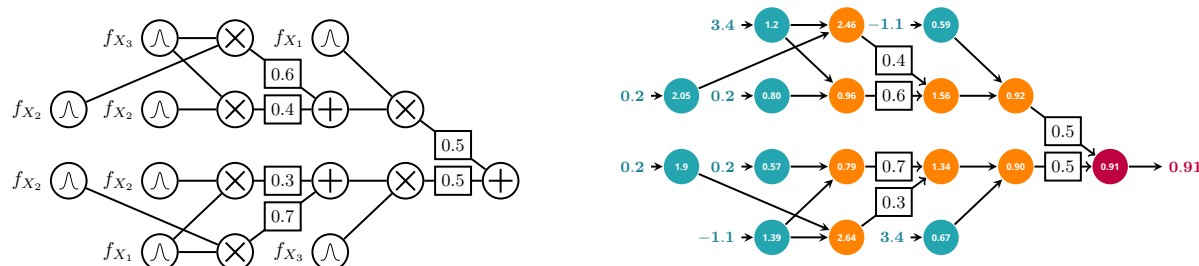

Figure 1: **Example of a circuit (left) and its evaluation (right)** for a circuit encoding the joint density over three continuous random variables $X_1, X_2, X_3$. We denote input units with $\bigwedge$ as they are univariate Gaussian distributions and label them with their scopes (left) while later on we will draw generic input units with an empty circle. To compute the joint density for $p(X_1 = -1.1, X_2 = 0.2, X_3 = 3.4)$, one has to first evaluate the Gaussian densities at the inputs (blue) and propagate the computed values. These densities are then multiplied across product units $\otimes$ and then passed through sums $\oplus$ (both in orange), whose parameters are here explicitly drawn in boxes. We will omit drawing the sum units weights in other pictures to avoid clutter. The value of $p(X_1 = -1.1, X_2 = 0.2, X_3 = 3.4) = 0.91$ is obtained by collecting the output of the last unit (in purple). See Section 3 for more circuits encoding distributions.

*RESCAL* (Nickel et al., 2011), and the *higher-order singular value decomposition* (HOSVD) (De Lathauwer et al., 2000) are all particular cases of the *Tucker* factorization (Tucker, 1964; 1966). For this reason, our treatment of tensor factorizations will focus on Tucker first, and its hierarchical formulation (Grasedyck, 2010) later. Our results will generalize to special cases such as CP, RESCAL and HOSVD.

**Definition 1** (Tucker factorization (Tucker, 1964)). Let $\mathcal{T} \in \mathbb{R}^{I_1 \times \cdots \times I_d}$ be a $d$-dimensional tensor. The multilinear rank-$(R_1, \ldots, R_d)$ *Tucker factorization* of $\mathcal{T}$ factorizes it as a *core tensor* multiplied by a matrix along each dimension, i.e.,

$$\mathcal{T} \approx \mathcal{W} \times_1 \mathbf{V}^{(1)} \times_2 \mathbf{V}^{(2)} \ldots \times_d \mathbf{V}^{(d)} \tag{1}$$

where $\mathcal{W} \in \mathbb{R}^{R_1 \times \cdots \times R_d}$ is the core tensor, $\mathbf{V}^{(j)} \in \mathbb{R}^{I_j \times R_j}$ with $j \in [d]$ are the *factor matrices*, and $\approx$ denotes the approximation of the tensor on the left-hand side given by the right-hand side factorization. The above equation can be rewritten in element-wise notation as

$$t_{i_1 \cdots i_d} \approx \sum_{r_1=1}^{R_1} \cdots \sum_{r_d=1}^{R_d} w_{r_1 \cdots r_d} \, v_{i_1 r_1}^{(1)} \cdots v_{i_d r_d}^{(d)}. \tag{2}$$

Focusing on the element-wise notation, we can view the factorization of $\mathcal{T}$ as a function $c$ over $d$ discrete variables $\mathbf{X} = \{X_j\}_{j=1}^d$, each having domain $\mathsf{dom}(X_j) = [I_j]$, such that $t_{\mathbf{x}} \approx c(\mathbf{x})$ for any assignment $\mathbf{x} = \langle i_1, \ldots, i_d \rangle$ to variables $\mathbf{X}$. In other words, each assignment to $\mathbf{X}$ is mapped to one scalar tensor entry, whose value is computed by $c$. Eq. (2) highlights that such a tensor factorization encodes a polynomial defined over the factor matrix values associated to assignments to variables $\mathbf{X}$ (Kolda, 2006). Therefore, we can represent the factorization encoded in $c$ as a *circuit*, i.e., a computational graph consisting of sums and products as atomic operators, formally defined next.

**Definition 2** (Circuit (Choi et al., 2020; Vergari et al., 2021)). A *circuit* $c$ is a parameterized directed acyclic computational graph[1] over variables $\mathbf{X}$ encoding a function $c(\mathbf{X})$, and comprising three kinds of computational units: *input*, *product*, and *sum* units. Each product or sum unit $n$ receives the outputs of other units as inputs, denoted with the set $\mathsf{in}(n)$. Each unit $n$ encodes a function $c_n$ defined as: (i) $f_n(\mathsf{sc}(n))$ if $n$ is an input unit, where $f_n$ is a function over variables $\mathsf{sc}(n) \subseteq \mathbf{X}$, called its *scope*, (ii) $\prod_{j \in \mathsf{in}(n)} c_j(\mathsf{sc}(j))$ if $n$ is a product unit, and (iii) $\sum_{j \in \mathsf{in}(n)} w_j c_j(\mathsf{sc}(j))$ if $n$ is a sum unit, with $w_j \in \mathbb{R}$ denoting the weighted sum parameters. The scope of a product or sum unit $n$ is the union of the scopes of its inputs, i.e., $\mathsf{sc}(n) = \bigcup_{j \in \mathsf{in}(n)} \mathsf{sc}(j)$. The size of a circuit $c$, denoted as $|c|$, is the number of edges between the computational units.

---

[1]In our figures, the direction of the circuit edges is always assumed to be from input to output units, but it is not graphically shown to avoid clutter.

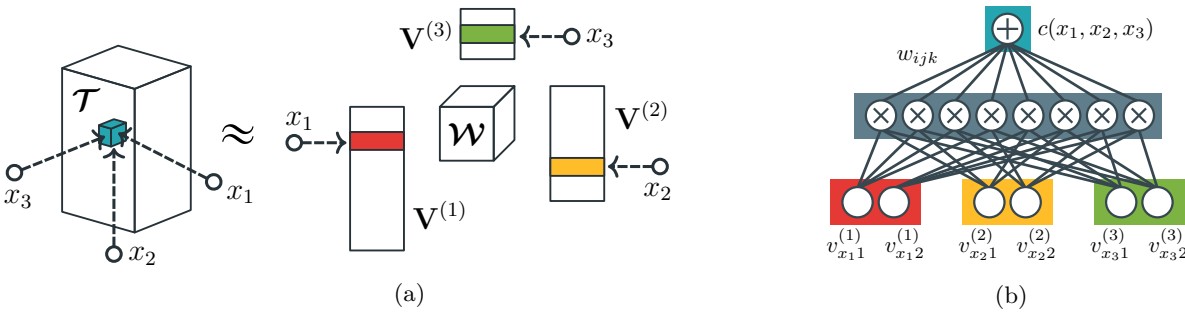

(a)  (b)

Figure 2: **Tucker tensor factorizations are circuits.** Given a tensor $\boldsymbol{\mathcal{T}} \in \mathbb{R}^{I_1 \times I_2 \times I_3}$ and its multilinear rank-$(2, 2, 2)$ Tucker factorization $\boldsymbol{\mathcal{T}} \approx \boldsymbol{\mathcal{W}} \times_1 \mathbf{V}^{(1)} \times_2 \mathbf{V}^{(2)} \times_3 \mathbf{V}^{(3)}$ (a), we can encode it as a circuit $c$ whose evaluation corresponds to computing an entry of the decomposed tensor, i.e., $t_{x_1 x_2 x_3} \approx c(x_1, x_2, x_3)$ for any entry index $(x_1, x_2, x_3)$ (b). The directionality of the circuit connections goes from input units to output units, but it is not shown to avoid clutter. The sum unit is parameterized by the entries $w_{ijk}$ of the core tensor $\boldsymbol{\mathcal{W}}$, while the input units are parameterized by the factor matrices $\mathbf{V}^{(1)}, \mathbf{V}^{(2)}, \mathbf{V}^{(3)}$. For instance, evaluating the two input units depending on the index $x_1$ (b, in red) translates to indexing the $x_1$-th row of $\mathbf{V}^{(1)}$, i.e., $\mathbf{v}_{x_1:} = [v_{x_1 1}^{(1)} \ v_{x_1 2}^{(1)}]^\top$ (a, in red).

Circuits can be understood as multilinear polynomials with exponentially many terms, but compactly encoded in a deep computational graph of polynomial size (Darwiche, 2003; Zhao et al., 2016; Choi et al., 2020). From this perspective, it is possible to intuit how they are related to, but also different from, tensor factorizations. In fact, while also the latter encode compact multilinear operators (Eq. (2)), the indeterminates of the circuit polynomials can be more than just entries of matrices as per Def. 2, e.g., potentially non-linear input functions. For example, a circuit can encode the joint density over a collection of continuous random variables, and input functions $f_n$ could encode Gaussian densities (Fig. 1). See also Opportunity 4 for a discussion on the many ways to encode input units in circuits.

Evaluating the function $c$ encoded in a circuit is done by traversing its computational graph in the usual *feedforward* way – inputs before outputs, see Fig. 1. Furthermore, the circuit definition we provided can be more general than tensor factorizations as it can represent *sparse* computational graphs, i.e., where units are irregularly connected. As we will argue later, this does not need to be the case. Circuits can be, in fact, designed to be locally-dense as it is common in many modern implementations (Section 4). Locally-dense architectures are also how tensor factorizations will look like, when turned into circuits, as we demonstrate in the following constructive proposition for a general Tucker factorization (Def. 1).

**Proposition 1** (Tucker as a circuit). Let $\boldsymbol{\mathcal{T}} \in \mathbb{R}^{I_1 \times \cdots \times I_d}$ be a tensor being decomposed via a multilinear rank-$(R_1, \ldots, R_d)$ Tucker factorization, as in Eq. (1). Then, there exists a circuit $c$ over variables $\mathbf{X} = \{X_j\}_{j=1}^d$ with $\mathsf{dom}(X_j) = [I_j]$, $j \in [d]$ computing the same factorization. Moreover, we have that $|c| \in \mathcal{O}(d \prod_{j=1}^d R_j)$.

Appendix A.1 details our proof construction and Fig. 2 illustrates it for the Tucker factorization of a three dimensional tensor. In a nutshell, we build a *shallow* circuit $c$ over the same variables that, when evaluated, outputs the reconstructed tensor entry for a set of coordinates, i.e., it encodes Eq. (2). Its input functions $f_n$, in fact, map variable states to *embeddings*, i.e., the real values contained in the matrices obtained from the Tucker factorization, see Fig. 1. Note that one can easily particularize our construction to obtain circuits corresponding to other factorizations such as CP, RESCAL and HOSVD.

As a concrete example of our construction, consider the following. Let $\boldsymbol{\mathcal{T}} \in \mathbb{R}^{3 \times 3 \times 3}$ be a three-dimensional tensor defined as

$$\boldsymbol{\mathcal{T}} = \left( \begin{pmatrix} -1.68 & 4.02 & -1.84 \\ 0.63 & \mathbf{-1.50} & 0.68 \\ 0.25 & -0.59 & 0.27 \end{pmatrix}, \begin{pmatrix} 16.83 & -40.24 & 18.36 \\ -6.27 & 14.99 & -6.84 \\ -2.48 & 5.918 & -2.7 \end{pmatrix}, \begin{pmatrix} 21.88 & -52.31 & 23.87 \\ -8.15 & 19.49 & -8.89 \\ -3.22 & 7.69, & -3.51 \end{pmatrix} \right) \quad (3)$$

and whose multilinear rank-$(2, 2, 2)$ Tucker decomposition is given by a tensor $\mathcal{W} \in \mathbb{R}^{2 \times 2 \times 2}$ whose entries are all 0.5 and by matrices

$$\mathbf{V}^{(1)} = \begin{pmatrix} \mathbf{0.1} & \mathbf{0.2} \\ -2.0 & -1.0 \\ 1.5 & -5.4 \end{pmatrix}, \quad \mathbf{V}^{(2)} = \begin{pmatrix} 1.1 & 9.1 \\ \mathbf{-3.3} & \mathbf{-0.5} \\ 0.7 & -2.2 \end{pmatrix}, \quad \mathbf{V}^{(3)} = \begin{pmatrix} -2 & 0.9 \\ \mathbf{0.23} & \mathbf{2.4} \\ -1.4 & 0.2 \end{pmatrix}. \tag{4}$$

Then, we can build a circuit $c$ with the same structure as the one in Fig. 2, equipping its input units with embeddings taken from $\mathbf{V}^{(1)}$, $\mathbf{V}^{(2)}$ or $\mathbf{V}^{(3)}$, depending on their scope, and by setting the sum unit parameters to be the vector $\mathbf{w} \in \mathbb{R}^8$ obtained by vectorizing the tensor $\mathcal{W}$ and therefore having values $= (0.5, \dots, 0.5)$. Now, to compute the approximate value of the $t_{1,2,2}$ entry in $\mathcal{T}$, we can evaluate the circuit $c$ in a feed-forward way—evaluating inputs before outputs—to compute $c(1, 2, 2)$. This would yield the following computation:

$$\mathbf{w}^\top \left( \begin{pmatrix} \mathbf{0.1} & \mathbf{0.2} \end{pmatrix}^\top \otimes \begin{pmatrix} \mathbf{-3.3} & \mathbf{-0.5} \end{pmatrix}^\top \otimes \begin{pmatrix} \mathbf{0.23} & \mathbf{2.4} \end{pmatrix}^\top \right) \approx \mathbf{-1.4991}. \tag{5}$$

Note how the color-coded blocks inside the brackets correspond to the outputs of the input functions in the circuits (Fig. 2), and how the vector outer products ($\otimes$) realize the product units in $c$ while the dot product with $\mathbf{w}$ is encoded in the final sum unit. We invite the reader to play with this example and try to recover other entries in the tensor, until they are comfortable with the translation of a tensor factorization into our circuit format. Furthermore, since circuits can represent factorizations, they inherit the same non-uniqueness issue commonly arising in many tensor factorization methods (e.g., Tucker). That is, the tensor factorization encoded by a circuit is not unique: one can change the circuit parameters while still encoding the same function. Finally, we remark that the multilinear-rank of the factorization now translates into the number of the input units in the circuit representation. Later, for hierarchical factorizations turned into deep circuits (Section 2.2) ranks will turn into the number of units located at different depths as well.

Representing tensor factorizations as computational graphs of this kind will offer a number of opportunities for extending the former model class, in which case we will highlight them in boxes throughout the paper. At the same time, we can better understand why these factorizations already support the tractable computation of certain quantities of interest, e.g., the computation of integrals, information theoretic measures or maximization (Vergari et al., 2021). This can be done in a systematic way in the framework of circuits, that maps these computations to the presence of certain structural properties of the computational graph, precisely defining sufficient (and sometimes necessary) conditions for tractability. We start by defining *smoothness* and *decomposability*, two structural properties of circuits that allow to tractably compute summations over exponentially many variable assignments, which are often intractable to compute for other models.

**Definition 3** (Unit-wise smoothness and decomposability (Darwiche & Marquis, 2002))**.** A circuit is *smooth* if for every sum unit $n$, its input units depend all on the same variables, i.e., $\forall i, j \in \mathsf{in}(n)\colon \mathsf{sc}(i) = \mathsf{sc}(j)$. A circuit is *decomposable* if for every product unit $n$, its input units depend on mutually disjoint sets of variables, i.e., $\forall i, j \; i \neq j\colon \mathsf{sc}(i) \cap \mathsf{sc}(j) = \varnothing$.

For a smooth and decomposable circuit one can exactly compute summations of the form $\sum_{\mathbf{z} \in \mathsf{dom}(\mathbf{Z})} c(\mathbf{y}, \mathbf{z})$, where $\mathbf{Z} \subseteq \mathbf{X}$, $\mathbf{Y} = \mathbf{X} \backslash \mathbf{Z}$, called *marginals*, in a single feedforward pass of its computational graphs (Choi et al., 2020). See also our discussion in Section 3 for more use cases of smoothness and decomposability. It is easy to verify that a Tucker tensor factorization represented as a circuit (e.g., Fig. 2) is both smooth and decomposable, and hence inherits tractable marginalization. In addition, under this light, one can understand the expressiveness of these factorizations, for multilinear polynomials expressiveness is usually characterized in terms of circuits with these structural properties (Shpilka & Yehudayoff, 2010; Martens & Medabalimi, 2014; de Colnet & Mengel, 2021).

**Where do circuits and tensor factorizations come from?** Now that we have established a first link between tensor factorizations and circuits, as the former can be rewritten as computational graphs with structural properties in the language of the latter, we also point out a first difference in how the two communities obtain and approach these objects. Tensor factorizations arise from the need to compressing a *given* high-dimensional tensor, which is usually *explicitly* represented (if not on memory, on disk). A factorization is then retrieved as the output of an optimization problem, e.g., find the factors that minimize

a certain reconstruction loss (Sidiropoulos et al., 2017; Cichocki et al., 2007). In contrast, modern circuits are *learned from data*. While this can be done both in a supervised and unsupervised way, the latter is more common as circuits are learned to encode a probability distribution. Such a distribution can be thought as an *implicit* tensor that is never observed, but from which we sampled data points. Section 3 formalizes this and the circuit learning problem. Even if reconstructing tensors is generally done differently than learning circuit from data, *once a factorization is given, by looking at it as a circuit, we can open up new opportunities to use it and exploit it*. We highlight them as boxes in the following sections. Next, we discuss how the framework of circuits also generalizes hierarchical (or deeper) tensor factorizations, which will also provide the entry point of our pipeline for *learning* both circuits and tensor factorizations (Section 4).

## 2.2 Hierarchical Tensor Factorizations are Deep Circuits

Tensor factorizations can be stacked together to form a *deep* or *hierarchical* factorization that can be much more space-efficient (i.e., of much lower rank) than its *shallow* materialization. For instance, Grasedyck (2010) proposed *hierarchical Tucker*, which stacks many low-rank Tucker factorizations according to a fixed hierarchical partitioning of tensor dimensions. Cohen et al. (2015) showed that in most cases equivalent or even approximate shallow factorizations would instead require an exponential rank with respect to the number of dimensions. Similar theoretical results have been also shown for circuits, i.e., deep circuits can be exponentially smaller than shallow circuits, where the size of a circuit is the number of unit connections (Delalleau & Bengio, 2011; Martens & Medabalimi, 2014; Jaini et al., 2018b).

In this section, we first introduce the hierarchical Tucker factorization, show that it is a deep circuit, and later use this connection to describe modern tensorized circuit representations (Section 4). To do so, we borrow a tool from the circuit literature: a hierarchical partitioning of the scope of a circuit (Vergari et al., 2021), aka *region graph* (RG) (Dennis & Ventura, 2012). As we formalize next, a RG is a bipartite graph whose nodes are either sets of variables, i.e., the dimensions of the tensor, or indicate how they are partitioned.

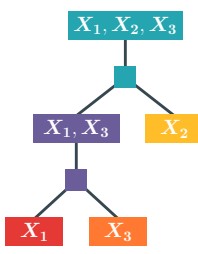

**Definition 4** (Region graph (Dennis & Ventura, 2012)). Given a set of variables $\mathbf{X}$, a *region graph* $\mathcal{R}$ is a bipartite and rooted directed acyclic graph (DAG) whose nodes are either *regions*, denoting subsets of $\mathbf{X}$, or *partitions*, specifying how a region is partitioned into other regions. The root is the region node $\mathbf{X}$.

Without loss of generality, we assume binary RGs, i.e., each region is partitioned into two others, as shown in Fig. 3. Similarly to our graphical notation of circuits (Def. 2), we remove the directionality of node connections from the figures and assume that edges are oriented from region nodes of more variables towards regions of fewer variables. Next, we define the hierarchical variant of Tucker.

Figure 3: A tree RG.

**Definition 5** (Hierarchical Tucker factorization). Let $\mathcal{T} \in \mathbb{R}^{I_1 \times \cdots \times I_d}$ be a $d$-dimensional tensor, and let $\mathbf{X}$ be the region root of a tree-shaped binary RG $\mathcal{R}$ whose leaves have exactly one variable, where $\mathsf{dom}(X_j) = [I_j]$ for all $X_j \in \mathbf{X}$. The *hierarchical Tucker factorization* of $\mathcal{T}$ is given by recursively applying Tucker factorizations according to the partitioning of indices induced by $\mathcal{R}$. There are three cases:

- First, for every leaf region $\mathbf{Z} = \{X_j\}$ in $\mathcal{R}$, we define $u_{x_j r}^{(\mathbf{Z})}$ to be an alias of the $(x_j, r)$-th entry of the factor matrix $\mathbf{V}^{(j)} \in \mathbb{R}^{I_j \times R_{\mathbf{Z}}}$ associated to $\mathbf{Z}$.

- Next, for every non-leaf region $\mathbf{Y} \subseteq \mathbf{X}$ partitioned into $(\mathbf{Z}_1, \mathbf{Z}_2)$ in $\mathcal{R}$, i.e., $\mathbf{Y} = \mathbf{Z}_1 \cup \mathbf{Z}_2$ with $\mathbf{Y} = \{Y_j\}_{j=1}^l$, $\mathbf{Z}_1 = \{Z_{1,j}\}_{j=1}^m$, $\mathbf{Z}_2 = \{Z_{2,j}\}_{j=1}^n$, we recursively define the Tucker factorization associated to $\mathbf{Y}$ as

$$u_{y_1 \cdots y_l s}^{(\mathbf{Y})} \approx \sum_{r_1=1}^{R_{\mathbf{Z}_1}} \sum_{r_2=1}^{R_{\mathbf{Z}_2}} w_{s\, r_1 r_2}^{(\mathbf{Y})}\, u_{z_{1,1} \cdots z_{1,m} r_1}^{(\mathbf{Z}_1)}\, u_{z_{2,1} \cdots z_{2,n} r_2}^{(\mathbf{Z}_2)} \qquad \text{with } s \in [R_{\mathbf{Y}}], \tag{6}$$

where $(R_{\mathbf{Y}}, R_{\mathbf{Z}_1}, R_{\mathbf{Z}_2})$ denotes the multilinear rank of the Tucker factorization. Moreover, $\boldsymbol{\mathcal{W}}^{(\mathbf{Y})} \in \mathbb{R}^{R_{\mathbf{Y}} \times R_{\mathbf{Z}_1} \times R_{\mathbf{Z}_2}}$ is the corresponding core tensor, and $\mathbf{y} = \langle y_1, \ldots, y_l \rangle$, $\mathbf{z}_1 = \langle z_{1,1}, \ldots, z_{1,m} \rangle$, $\mathbf{z}_2 = \langle z_{2,1}, \ldots, z_{2,m} \rangle$ are assignments to variables $\mathbf{Y}, \mathbf{Z}_1, \mathbf{Z}_2$, respectively.

- Finally, in the case of the root region $\mathbf{Y} = \mathbf{X}$ in the recursive rule in Eq. (6), we define $R_{\mathbf{Y}} = 1$ and $u_{x_1 x_2 \cdots x_d 1}^{(\mathbf{Y})}$ in Eq. (6) becomes an alias of the entry $t_{x_1 x_2 \cdots x_d}$ of $\mathcal{T}$.

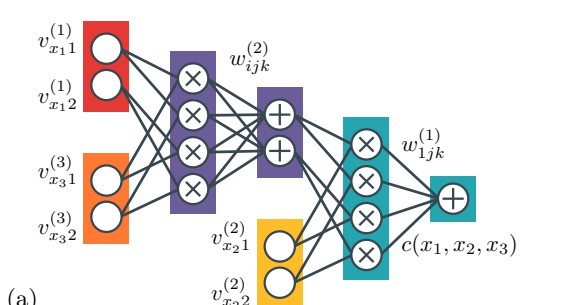
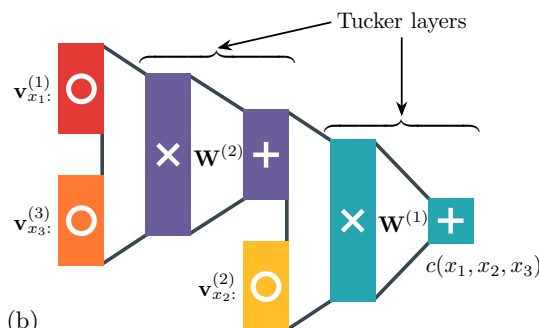

Figure 4: **Hierarchical Tucker factorizations are deep (tensorized) circuits** as shown here with the circuit representation of the hierarchical Tucker factorization of a three dimensional tensor (a), which is obtained by stacking two Tucker factorizations according to the RG in Fig. 3. Evaluating the circuit from left to right for some entry $(x_1, x_2, x_3)$ computes the corresponding tensor entry. In (b) we show the equivalent tensorized architecture (Def. 7) obtained by grouping units into layers, according to the graphical convention introduced in Def. 7. Input layers map indices into rows of factor matrices, while products layers compute Kronecker products of their inputs, and sum units compute a matrix-vector product. The core tensors $\mathcal{W}^{(2)} \in \mathbb{R}^{2 \times 2 \times 2}, \mathcal{W}^{(1)} \in \mathbb{R}^{1 \times 2 \times 2}$ that parameterize the sum units in (a) are reshaped into matrices $\mathbf{W}^{(2)} \in \mathbb{R}^{2 \times 4}, \mathbf{W}^{(1)} \in \mathbb{R}^{1 \times 4}$ in (b). In Section 4 we will refer to the composition of Kronecker product and sum layers simply as *Tucker layer*, as showed in (b).

We provide an example of a hierarchical Tucker factorization, as to show an application of the recursive Tucker factorization shown in Eq. (6). Given a three-dimensional tensor $\mathcal{T} \in \mathbb{R}^{I_1 \times I_2 \times I_3}$, we factorize it via hierarchical Tucker according to the RG shown in Fig. 3. Since the RG in Fig. 3 has two partitionings, we recursively perform two Tucker factorizations (as in Eq. (6)), and choose $(R_{\{X_1, X_2, X_3\}}, R_{\{X_2\}}, R_{\{X_1, X_3\}})$ and $(R_{\{X_1, X_3\}}, R_{\{X_1\}}, R_{\{X_3\}})$ as the respective multilinear ranks, i.e., each entry of $\mathcal{T}$ is approximated as

$$t_{x_1 x_2 x_3}^{\{X_1, X_2, X_3\}} \approx \sum_{r_1=1}^{R_{\{X_2\}}} \sum_{r_2=1}^{R_{\{X_1, X_3\}}} w_{1 r_1 r_2}^{\{X_1, X_2, X_3\}} \, v_{x_2 r_1}^{\{X_2\}} \, u_{x_1 x_3 r_2}^{\{X_1, X_3\}},$$

where $\mathcal{W} \in \mathbb{R}^{1 \times R_{\{X_2\}} \times R_{\{X_1, X_3\}}}$ is the core tensor of the first Tucker factorization, $\mathbf{V}^{\{X_2\}} \in \mathbb{R}^{I_2 \times R_{\{X_2\}}}$ is the factor matrix associated to $\{X_2\}$, and $\mathcal{U}^{\{X_1, X_3\}} \in \mathbb{R}^{I_1 \times I_3 \times R_{\{X_1, X_3\}}}$ consists of $R_{\{X_1, X_3\}}$ matrices of shape $I_1 \times I_3$ being factorized according to the second Tucker factorization,[2] i.e.,

$$u_{x_1 x_3 r_2}^{\{X_1, X_3\}} = \sum_{r_3=1}^{R_{\{X_1\}}} \sum_{r_4=1}^{R_{\{X_3\}}} w_{r_2 r_3 r_4}^{\{X_1, X_3\}} \, v_{x_1 r_3}^{\{X_1\}} \, v_{x_3 r_4}^{\{X_3\}},$$

where $\mathbf{W}^{\{X_1, X_3\}} \in \mathbb{R}^{R_{\{X_1, X_3\}} \times R_{\{X_1\}} \times R_{\{X_3\}}}$, $\mathbf{V}^{\{X_1\}} \in \mathbb{R}^{I_1 \times R_{\{X_1\}}}$, and $\mathbf{V}^{\{X_3\}} \in \mathbb{R}^{I_3 \times R_{\{X_3\}}}$.

Following this recursive definition of a hierarchical Tucker factorization, we now build an equivalent circuit $c$ encoding the same factorization, i.e., $t_{\mathbf{x}} \approx c(\mathbf{x})$, by stacking weighted sum and product units together as to construct a *deep* circuit. In the following constructive proposition we present this construction.

**Proposition 2** (Hierarchical Tucker as a deep circuit). Let $\mathcal{T} \in \mathbb{R}^{I_1 \times \dots \times I_d}$ be a tensor being decomposed using hierarchical Tucker factorization according to a RG $\mathcal{R}$. Then, there exists a circuit $c$ over variables $\mathbf{X} = \{X_j\}_{j=1}^d$ with $\mathsf{dom}(X_j) = [I_j]$, computing the same factorization. Furthermore, given $\{\mathbf{Y}^{(i)}\}_{i=1}^m \subset 2^{\mathbf{X}}$ the set of all non-leaf region nodes $\mathbf{Y}^{(i)} \subseteq \mathbf{X}$ being factorized into $(\mathbf{Z}_1^{(i)}, \mathbf{Z}_2^{(i)})$ in $\mathcal{R}$, with corresponding Tucker factorization multilinear rank $(R_{\mathbf{Y}^{(i)}}, R_{\mathbf{Z}_1^{(i)}}, R_{\mathbf{Z}_2^{(i)}})$, we have that $|c| \in \mathcal{O}\left(\sum_{i=1}^m R_{\mathbf{Y}^{(i)}} R_{\mathbf{Z}_1^{(i)}} R_{\mathbf{Z}_2^{(i)}}\right)$.

---

[2]The Tucker factorization of a three-dimensional tensor into only two factor matrices implicitly assumes the identity matrix as third factor, and it is also called *Tucker2 factorization* (Tucker, 1966; Kolda & Bader, 2009).

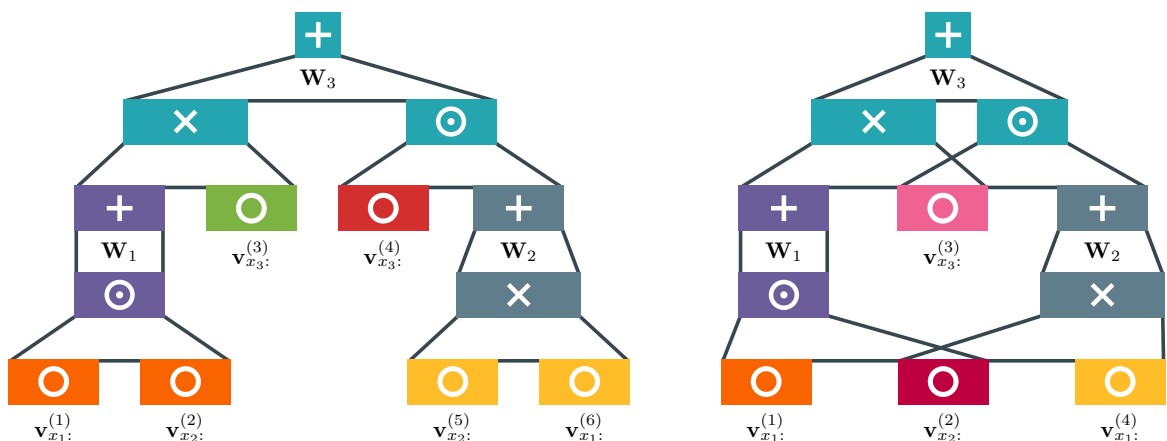

Figure 6: **Tensorized circuits can encode novel hierarchical multilinear factorizations by mixing different structures and layers.** Section 2.3 and Fig. 7 formalize and illustrate our tensorized circuit formalism, respectively. **The figure on the left** shows a tensorized circuit over variables $\mathbf{X} = \{X_1, X_2, X_3\}$ encoding an allowed multilinear factorization of a three-dimensional tensor (as it is smooth and decomposable, see Def. 8). Note that each input layer has its own factor matrix $\mathbf{V}^{(j)}$ with $j \in [6]$, and the architecture consists of a mix of Hadamard, Kronecker and sum inner layers. Overall, this tensorized circuit do not map to a known hierarchical factorization. **The figure on the right** shows a similar tensorized circuit, where the factor matrices $\mathbf{V}^{(2)}$ and $\mathbf{V}^{(3)}$ are instead shared while still encoding an allowed multilinear factorization.

Appendix A.2 shows the construction, also illustrated in Fig. 4a for a hierarchical Tucker factorization based on the RG showed in Fig. 3. In the very same way one can extend any tensor factorization to be hierarchical, one can represent such a construction as a circuit. However, in the circuit literature we found many architectures that are not limited to RGs that are trees nor to those having univariate input regions.

---

**Opportunity 1. A wider choice of factorization structures**

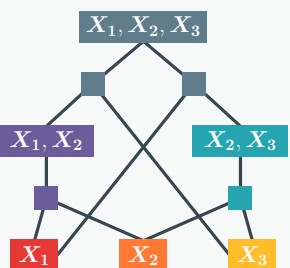

Figure 5: Region nodes can be shared between partitionings in a DAG RG.

Def. 4 allows for arbitrary DAGs and arbitrarily scoped-regions, while hierarchical tensor factorizations are usually presented in terms of RGs with a tree structure having region leaves containing *exactly one variable* (e.g., the RG in Fig. 3), which are sometimes called *dimension trees* or *mode cluster trees* (Grasedyck, 2010) in the tensor factorization community, and often *vtree* in circuit literature (Pipatsrisawat & Darwiche, 2008; Kisa et al., 2014; Wedenig et al., 2024a). More intricate RG structures can increase the expressiveness as well as flexibility in building deep circuits/hierarchical factorizations. Intuitively, we can share factor matrices among multiple factorizations, and therefore reduce the number of model parameters, making a more space-efficient implementation possible. See Peharz et al. (2020c;a) for more details. We provide an example of such a RG in Fig. 10 and Fig. 5 here on the left.

The reader can check that this RG encodes the hierarchical scope partitioning of the decomposable circuit in figure Fig. 1. Fig. 9 then illustrates a fragment of this RG and shows how tensor factorizations conforming to it can be constructed as circuits in Section 4. The circuit literature provides several ways to build RGs that are suitable for certain data modalities (e.g., image, sequence, tabular data), which can also be learned from data. Section 4.1 provides an overview of such techniques.

---

Imposing a particular factorization structure by leveraging a RG, and picking a particular parameterization for each region in it (as it will be discussed in Section 4), represents one way to encode novel hierarchical factorizations that do not correspond to existing ones. Fig. 6 shows some examples. There, we represent circuits in a layer-wise formalism as described later in Section 2.3. Note that instantiating tensor factorizations

from RGs defined as above preserve decomposability, and that circuits built from RGs in the literature are typically also smooth (Def. 3). Hierarchical Tucker and its variants are also smooth and decomposable and therefore support the tractable computation of a number of (probabilistic) inference tasks (Section 3). These hierarchical factorizations (and the corresponding deep circuits) that follow a tree-shaped RG with univariate leaves satisfy an additional structural property, called *structured-decomposability*. Structured decomposability enables the tractable computation of harder operations for which smoothness and decomposability are not enough. For instance, squaring particular tensor factorizations formalized in the graphical language of tensor networks, known as the Born rule in physics (Feynman, 1987; Glasser et al., 2019) (see also Section 2.4). We define structured decomposability below.

**Definition 6** (Structured decomposability (Pipatsrisawat & Darwiche, 2008))**.** A circuit is *structured decomposable* if (1) it is smooth and decomposable, and (2) any pair of product units $n, m$ having the same scope decompose their scope at their input units in the same way.

We can easily check that hierarchical Tucker yields a structured decomposable circuit, as it is obtained by stacking Tucker factorizations (which are computed by decomposable circuits) based on a tree RG, which in turn synchronizes all product units to decompose in the same way. We emphasize that eliciting the few structural properties that can explain the tractable computation of many different quantities of interest can help save effort aimed at (re)discovering and (re)engineering algorithms for specific hierarchical factorizations.

---

### Opportunity 2. Efficient Compositional Operations over Factorizations

Given one or more tensor factorizations appearing as operands in a computation of interest, how can we automatically devise a tractable algorithm for it without having to materialize the exponentially large tensor operands? The circuit literature holds the answer and offers other structural properties that can unlock the tractable computation of many complex inference scenarios, in a *reusable* fashion. E.g., when two deep circuits conform to the same tree RG, they are said to be *compatible* (Vergari et al., 2021). Given two compatible hierarchical tensor factorizations $p$ and $q$ over $\mathbf{X}$, one can compute general expectations of the form

$$\sum_{\mathbf{x}} p(\mathbf{x})q(\mathbf{x}) \qquad \text{(expectations)}$$

in closed form in time $\mathcal{O}(|p||q|)$, where $|p|$ and $|q|$ are the size of the corresponding circuits encoding such factorizations. On the other hand, maximization problems as in maximum-a-posteriori inference

$$\max_{\mathbf{y}} p(\mathbf{y}, \mathbf{E} = \mathbf{e}) \qquad \text{(MAP inference)}$$

where $\mathbf{e}$ is the *evidence* assignment to variables $\mathbf{E} \subset \mathbf{X}$, and $\mathbf{y}$ is the assignment to the remaining variables $\mathbf{Y} = \mathbf{X} \backslash \mathbf{E}$ for which we want to maximize $p$, can be solved exactly and efficiently if $p$ is a decomposable circuit that supports an additional property, *determinism* (Darwiche, 2009). In a nutshell, sum units in a deterministic circuit receive inputs from functions with disjoint support (see Choi et al. (2020) for details). While determinism is a consolidated property in the circuit literature, it is off the radar for (hierarchical) tensor factorizations. Furthermore, the circuit literature provides a systematic way to quickly devise the tractability conditions for a given mathematical expression that involves sums, products, powers, exponentials and logarithms, and therefore automatically distill corresponding tractable algorithms (Vergari et al., 2021). For example, if one wants to compute Rényi's $\alpha$-divergence between two factorizations $p$ and $q$ over variables $\mathbf{X}$, for $\alpha \in \mathbb{N}$, defined as

$$(1-\alpha)^{-1} \log \sum_{\mathbf{x}} p^{\alpha}(\mathbf{x}) q^{1-\alpha}(\mathbf{x}), \qquad (\alpha\text{-divergence})$$

then this can be done quickly if $p$ and $q$ can be represented as smooth, decomposable and compatible circuits and $q$ is also deterministic. Vergari et al. (2021) show how to automatically distill the tractable computation of more information-theoretic quantities.

---

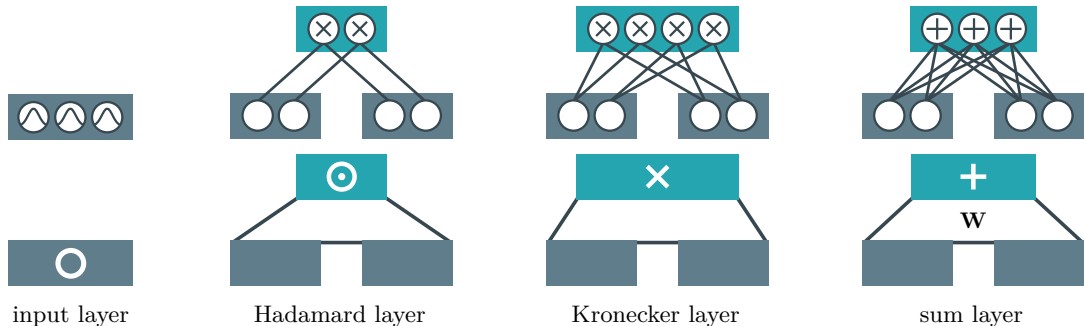

Figure 7: **Tensorized "Lego blocks".** In the rest of the figures we will abstract away from individual connections between units (as we did so far, and as we do in the top row illustrations) and represent layers as (colored) blocks (bottom row). Input layers are the only layers that do not have any other layer as input, i.e., they take a variables assignment and output a vector computed by a function $f$. Hadamard and Kronecker product layers receive inputs from at least two other layers (represented in gray), and compute the Hadamard and Kronecker products of their inputs, respectively. A sum layer parameterized by a weight matrix $\mathbf{W}$ concatenates its input layers into a single vector, and then multiplies it by $\mathbf{W}$.

## 2.3 Representing Circuits in a Tensorized Formalism

Representing (hierarchical) tensor factorization as (deep) circuits highlights how circuit units can be naturally grouped together by type and scope into *layers*, as hinted already in Fig. 2. This perspective presents a new opportunity: defining and representing certain circuit structures as *tensorized computational graphs*. While circuits in the literature are defined in terms of scalar computational units, sum, product and inputs and single connections (Def. 2), many successful implementations of circuits nowadays already group units into tensors (Vergari et al., 2019a; Peharz et al., 2020c;a; Liu & Van den Broeck, 2021b; Loconte et al., 2024) with the goal of speeding up computation by using the acceleration provided by GPUs. Following these ideas, we now provide a general tensorized circuit definition that offers a *modular* way to build overparameterized circuit architectures. This will allow us to design a single learning pipeline that subsumes many existing architectures (Section 4), and also suggest a way to create novel ones by mixing and reusing small "blocks".

**Definition 7** (Tensorized circuit). A *tensorized circuit* $c$ is a computational graph composed of three kinds of layers: *input*, *product* and *sum*. Each layer $\boldsymbol{\ell}$ consists of computational units defined over the same scope $\mathsf{sc}(\boldsymbol{\ell})$. Every non-input layer receives the output vectors of other layers as inputs, denoted with the set $\mathsf{in}(\boldsymbol{\ell})$. The three kinds of layers are defined as follows:

- Each input layer $\boldsymbol{\ell}$ has scope $\mathbf{Y} \subseteq \mathbf{X}$ and computes a vector function $f \colon \mathsf{dom}(\mathbf{Y}) \to \mathbb{R}^K$.

- Each product layer $\boldsymbol{\ell}$ computes either an Hadamard product $(\bigodot_{\boldsymbol{\ell}_j \in \mathsf{in}(\boldsymbol{\ell})} \boldsymbol{\ell}_j)$ or Kronecker product $(\bigotimes_{\boldsymbol{\ell}_j \in \mathsf{in}(\boldsymbol{\ell})} \boldsymbol{\ell}_j)$ over the vectors it receives from its input layers $\boldsymbol{\ell}_j$.

- A sum layer with $S$ sum units computes the matrix-vector product $\mathbf{W}(\|_{\boldsymbol{\ell}_j \in \mathsf{in}(\boldsymbol{\ell})} \boldsymbol{\ell}_j(\mathsf{sc}(\boldsymbol{\ell}_j)))$, where $\|$ denotes vector concatenation and $\mathbf{W} \in \mathbb{R}^{S \times K}$, $K > 0$ are the sum layer parameters.

Note that if a sum layer $\boldsymbol{\ell}$ receives only one input vector, i.e., $|\mathsf{in}(\boldsymbol{\ell})| = 1$, then it simply computes $\mathbf{W}\boldsymbol{\ell}_1(\mathsf{sc}(\boldsymbol{\ell}_1))$. Fig. 7 illustrates the layer types of a tensorized circuit, together with the unit-wise representation (Def. 2). Furthermore, we retrieve the previous scalar unit-wise definition by setting $K$, the size of each layer, to 1. The above four types of layers constitute the basic "Lego blocks" that we will later use to create more sophisticated layers (Section 4.3, Section 5) and reproduce all modern circuit architectures (Table 1).

As a first example on how this definition can help to abstract away from details in circuit architectures, see Fig. 4. There, sum and Kronecker product layers are used to stack two Tucker tensor factorizations to represent a hierarchical one. We provide in Section 4 a systematic way to stack different layers and build a deep circuit in this way. We can now easily extend the unit-wise definition of structural properties in Def. 3 to this layer-wise representation, by defining the scope of each layer.

Figure 8: **A MPS/TT represented as a deep tensorized circuit** with Hadamard product layers (b). To obtain the parameters of the circuit, the tensor $\boldsymbol{\mathcal{A}}^{(2)}$ in the MPS/TT (a, showed in Penrose graphical notation) is firstly factorized into matrices $\mathbf{V}^{(2)}, \mathbf{B}, \mathbf{C}$ through a CANDECOMP/PARAFAC decomposition (Carroll & Chang, 1970). Then, $\mathbf{V}^{(1)}, \mathbf{V}^{(3)}, \mathbf{W}$ are obtained as in the figure (a). See Loconte et al. (2024) for the detailed circuit construction. In (b) we denote with $\mathbf{1}$ a row-matrix whose entries are all ones.

**Definition 8** (Layer-wise smoothness and decomposability). A tensorized circuit over variables $\mathbf{X}$ is *smooth* if for every sum layer $\boldsymbol{\ell}$, its inputs depend all on the same variables, i.e., $\forall \boldsymbol{\ell}_i, \boldsymbol{\ell}_j \in \mathsf{in}(\boldsymbol{\ell}) \colon \mathsf{sc}(\boldsymbol{\ell}_i) = \mathsf{sc}(\boldsymbol{\ell}_j)$, where $\mathsf{sc}(\boldsymbol{\ell}) \subseteq \mathbf{X}$ is the scope of layer $\boldsymbol{\ell}$, i.e., the scope of the units in $\boldsymbol{\ell}$. It is *decomposable* if for every product layer $\boldsymbol{\ell}$ in it, its inputs depend on disjoint sets of variables, i.e., $\forall \boldsymbol{\ell}_i, \boldsymbol{\ell}_j \in \mathsf{in}(\boldsymbol{\ell}), i \neq j \colon \mathsf{sc}(\boldsymbol{\ell}_i) \cap \mathsf{sc}(\boldsymbol{\ell}_j) = \varnothing$.

Note that by assuming that every layer is composed by units sharing the same scope, and by using the three layers defined in Def. 7, we obtain tensorized circuits that are smooth and decomposable *by design*. Furthermore, if the RG of a deep circuit is a tree, then the tensorized circuit will be structured-decomposable (Def. 6) as well. It is possible to quickly read these properties out of the graphical representation of hierarchical Tucker as a tensorized circuit in Fig. 4b. Next, we use this layered abstraction to bridge to the popular *tensor networks*, and show how they can be naturally encoded as deep circuits.

## 2.4 Tensor Networks as Deep Circuits

Tensor networks (TNs) are often the preferred way to represent hierarchical tensor factorizations in fields such as physics and quantum computing (Markov & Shi, 2008; Schollwoeck, 2010; Biamonte & Bergholm, 2017). TNs come with a graphical language – Penrose notation – to encode tensor dot products in a compact graphical formalism (also called *tensor contractions*). See Orús (2013) for a review. Perhaps, the most popular TN factorization is the matrix-product state (MPS) (Pérez-García et al., 2007), also called tensor-train factorization (TT) (Oseledets, 2011; Glasser et al., 2019; Novikov et al., 2021). For instance, given a tensor $\boldsymbol{\mathcal{T}} \in \mathbb{R}^{I_1 \times \cdots \times I_d}$, its rank-$R$ MPS/TT factorization is defined in element-wise notation as

$$t_{i_1 \cdots i_d} \approx \sum_{r_1=1}^{R} \sum_{r_2=1}^{R} \cdots \sum_{r_{d-1}=1}^{R} a_{i_1,r_1}^{(1)} a_{i_2,r_1,r_2}^{(2)} \cdots a_{i_{d-1},r_{d-2},r_{d-1}}^{(d-1)} a_{i_{d-1},r_{d-1}}^{(d)} \tag{7}$$

where $\mathbf{A}^{(1)} \in \mathbb{R}^{I_1 \times R}$, $\mathbf{A}^{(d)} \in \mathbb{R}^{I_d \times R}$, and $\boldsymbol{\mathcal{A}}^{(j)} \in \mathbb{R}^{I_j \times R \times R}$ with $1 < j < d$. That is, an MPS factorization decomposes $\boldsymbol{\mathcal{T}}$ into the complete contraction of a chain of smaller tensors $\mathbf{A}^{(1)}, \mathbf{A}^{(d)}$, and $\{\boldsymbol{\mathcal{A}}^{(j)}\}_{j=2}^{d-1}$. Fig. 8a shows an example of a MPS/TT represented in Penrose graphical notation, i.e., where nodes denote the tensors $\mathbf{A}^{(1)}, \boldsymbol{\mathcal{A}}^{(2)}, \ldots, \boldsymbol{\mathcal{A}}^{(d-1)}, \mathbf{A}^{(d)}$, edges denote summations over shared indices, and $X_1, \ldots, X_d$ denote the tensor indices whose assignment yield the corresponding tensor entry. Loconte et al. (2024) showed how an MPS can be represented as a deep tensorized circuit by encoding summations and products in Eq. (7) into sum and (Hadamard) product layers, respectively.

**Proposition 3** (MPS as deep tensorized circuits (Loconte et al., 2024)). Let $\boldsymbol{\mathcal{T}} \in \mathbb{R}^{I_1 \times \cdots \times I_d}$ be a tensor being decomposed via a rank $R$ matrix-product state (MPS) factorization. Then, there exists a structured decomposable tensorized circuit $c$ over variables $\mathbf{X} = \{X_j\}_{j=1}^{d}$ with $\mathsf{dom}(X_j) = [I_j], j \in [d]$ computing the same factorization, i.e., $t_{\mathbf{x}} \approx c(\mathbf{x})$ for all entries $\mathbf{x}$. In addition, we have that $|c| \in \mathcal{O}(dN^2)$ with $N \leqslant \min\{R^2, R \max\{I_1, \ldots, I_d\}\}$.

In Fig. 8 we show a tensorized circuit representing a MPS/TT over variables $\mathbf{X} = \{X_1, X_2, X_3\}$, and, as detailed in the proof of Proposition 3 in Loconte et al. (2024), the parameters of its input and dense layers are obtained by decomposing the tensors $\{\boldsymbol{\mathcal{A}}^{(j)}\}_{j=2}^{d-1}$ of the MPS/TT. Similarly to the tensorized circuit representation of hierarchical Tucker (Proposition 2), Proposition 3 yields a tensorized circuit that is structured decomposable (Def. 6). Structured-decomposability is the crucial property in MPS/TTs that allows to perform certain operations over them tractably, for instance squaring them as to recover a *Born machine* – a probabilistic model devised to simulate quantum many-body systems in physics (Orús, 2013; Glasser et al., 2019). Understanding this enables practitioners to design alternative Born machine architectures that are not limited to a sequence of tensor operations as encoded in a "linear" RG, without having to prove the tractability of the square operation over these architectures from scratch (Shi et al., 2005). This is one of the opportunities we highlighted for hierarchical tensor factorizations once represented as circuits (Opportunity 1 and Opportunity 2). Further opportunities will be presented in the next section and directly translates to TNs as well as classical tensor factorizations.

**Next steps.** Until now, we discussed the generic decomposition of a real-valued tensor. However, tensor factorizations that are tailored for non-negative data (e.g. images), called *non-negative tensor factorizations*, factorize tensors into non-negative factors that can be easily interpreted (Cichocki & Phan, 2009). In Section 3, we connect non-negative tensor factorizations to the literature of circuits for probabilistic modeling, which allows us to interpret them as deep latent-variable models. In addition, by bridging non-negative tensor factorizations and their representation as (deep) circuits, we showcase future research opportunities related to both parameterizing tensor factorizations and performing probabilistic inference with them.

## 3 From Non-negative Factorizations to Circuits for Probabilistic modeling

Much attention has been paid in machine learning on circuit representations for tractable *probabilistic* modeling, i.e., for modeling probability distributions that support tractable inference. Circuits built with such a purpose are usually called *probabilistic circuits* (PCs) (Vergari et al., 2019b; Choi et al., 2020). In this section, we connect non-negative tensor factorizations and PCs, showing a number of research opportunities for the tensor factorization community within the probabilistic machine learning panorama.

First, we bridge non-negative (hierarchical) tensor factorizations with the discrete latent variable interpretation of (deep) PCs, showing examples of available algorithms for linear-time probabilistic inference that exploit this interpretation (not only marginals, as discussed in the previous section, but also sampling). Second, we show how the rich literature on PCs provides several compact parameterization techniques that can yield non-linear factorizations. At the same time, we leverage optimization tricks from the non-negative tensor literature to learn PCs. Finally, we connect with the literature of infinite-dimensional tensor factorizations showing their relationship with PCs encoding probability density functions, as well as with PCs equipped with infinite-dimensional sum units. We start by describing how to represent a probability distribution over finitely-discrete random variables as a tensor factorization.

Let $p(\mathbf{X})$ be a probability mass function (PMF) over finitely-discrete random variables $\mathbf{X} = \{X_j\}_{j=1}^{d}$, where each $X_j \in \mathbf{X}$ takes values in $\mathsf{dom}(X_j) = [I_j]$. Then, the simplest representation of $p(\mathbf{X})$ is that of a *probability tensor* $\boldsymbol{\mathcal{T}} \in \mathbb{R}_+^{I_1 \times \cdots \times I_d}$ such that every entry encodes the probability of a joint configuration of $\mathbf{X}$, i.e., $t_{x_1 \cdots x_d} = p(x_1, \ldots, x_d)$ for any $\mathbf{x} = \langle x_1, \ldots, x_d \rangle \in \mathsf{dom}(\mathbf{X})$. Clearly, this representation is inefficient, as it scales exponentially in space with respect to the number of variables $d$. A natural way to compactly model $p(\mathbf{X})$ is via a non-negative tensor factorization, e.g., the non-negative version of Tucker (Kim & Choi, 2007), where the factor matrices $\{\mathbf{V}^{(j)}\}_{j=1}^{d}$ and the core tensor $\boldsymbol{\mathcal{W}}$ shown in Eq. (2) are restricted to have non-negative entries only. By trivially specializing Proposition 2, we can encode the non-negative hierarchical Tucker factorization (Vendrow et al., 2021) in a circuit $c$ that outputs non-negative values, also called a PC.

**Definition 9** (Probabilistic circuit (Choi et al., 2020))**.** A *probabilistic circuit* (PC) over variables $\mathbf{X}$ is a circuit encoding a function $c(\mathbf{X})$ that is non-negative for all assignments to $\mathbf{X}$, i.e., $\forall \mathbf{x} \in \mathsf{dom}(\mathbf{X}): c(\mathbf{x}) \geqslant 0$.

A sufficient condition to ensure a circuit is a PC is constraining both the parameters of sum units and the outputs of input units to be non-negative, resulting in a circuit that is called *monotonic* (Shpilka &

Yehudayoff, 2010).[3] For instance, the circuit encoding a non-negative hierarchical Tucker factorization that we mentioned above is a monotonic PC, as its sum unit weights (i.e., the entries of the core tensor $\mathcal{W}$) and the outputs of its input units (i.e., the entries of the factor matrices $\{\mathbf{V}^{(j)}\}_{j=1}^d$) are restricted to be non-negative. Smoothness and decomposability in circuits allow for the tractable computation of summation and integrals (Section 2.1), which translates into exactly computing any marginal or conditional distribution for a PC with these structural properties (Vergari et al., 2019b). However, these PCs are not just tractable probabilistic models, they are also *generative models* from which it is possible to sample exactly.

### 3.1 Non-negative Tensor Factorizations as Generative Models

As non-negative factorizations—such as non-negative hierarchical Tucker—are smooth and (structured) decomposable PCs (Defs. 3 and 6), they inherit the ability of PCs to perform tractable inference and to generate new data points, i.e., certain configurations of the variables they are defined on. To the best of our knowledge, this treatment of tensor factorizations as generative models has gone unnoticed so far. We discuss it in the following, showing how one can devise (faster) sampling algorithms for these representations.

First, we review the simplest way to sample from a non-negative factorization. Consider a non-negative (hierarchical) Tucker factorization (Def. 5) encoding $p(\mathbf{X})$ and modeled as tensorized monotonic PC $c$. We can sample a data point $\mathbf{x} = \langle x_1, \ldots, x_d \rangle$ from $p(\mathbf{X})$ by autoregressively sampling one variable at a time, conditioned to the previously sampled variable assignments. That is, we can first marginalize all variables except $X_1$, and then sample from the distribution $p(X_1)$, i.e., $x_1 \sim p(X_1)$. This can be done in time $\mathcal{O}(|c|)$, as $c$ is both smooth and decomposable (Def. 3, Def. 8). Then, for all $d > 1$, we condition w.r.t. to the assignments to variables $\{X_i\}_{i=1}^{d-1}$ and sample $X_d$, i.e., $x_d \sim p(X_d \mid X_1, \ldots, X_{d-1})$. This "naive" sampling procedure requires worst-case time $\mathcal{O}(d|c|)$, where $|c|$ is the circuit size (see Def. 2). This can be inefficient in case of large $d$. However, for smooth and decomposable circuits, we can sample in $\mathcal{O}(|c|)$ only, by interpreting them as discrete *latent variable models* (Peharz et al., 2017; Vergari et al., 2018).

---

**Opportunity 3. Tensor factorizations as discrete latent variable models**

Each sum unit $n$ in a smooth PC can be thought as a *mixture model* computing:

$$c_n(\mathbf{X}) = \sum_{i=1}^K w_{n,i}\, c_{n,i}(\mathbf{X}), \quad \text{where} \quad \sum_{i=1}^K w_{n,i} = 1, \quad w_{n,i} > 0, \tag{8}$$

i.e., a convex combination of the its $K$ inputs, each one representing a distribution. At the same time, this can be interpreted as summing out a discrete latent variable $Z_n$ that has $K$ different states,

$$p_n(\mathbf{X}) = \sum_{i=1}^K p(Z_n = i)\, p_{n,i}(\mathbf{X} \mid Z_n = i)$$

where the non-negative weights $w_{n,i}$ are the marginal probabilities of this latent variable. As such, the whole circuit, and hence the corresponding non-negative tensor factorization, can be seen as a *hierarchical latent variable model* (Peharz et al., 2016; Choi et al., 2011), with as many discrete latent variables as the number of sum units. Therefore, as for any mixture model, to sample $\mathbf{x}$ we can first sample the latent variables, and then sample the mixture components. In practice, this sampling procedure can be done efficiently by performing a backward traversal of the circuit computational graph (Vergari et al., 2019a; Dang et al., 2022a). We provide this algorithm for tensorized circuits in Algorithm C.1, which sample a batch of $N$ data points in parallel and discuss it in Appendix C. Other efficient probabilistic inference tasks can be "imported" from the circuit literature for smooth and decomposable PCs. See Vergari et al. (2021) for more details.

---

### 3.2 How to Parameterize Probability Tensor Factorizations?

Circuits and tensor factorizations are the output of two different optimization problems that however share some common challenges. Understanding them can open new opportunities for both communities. In

---

[3]Non-monotonic PCs, which allow negative weights while ensuring non-negative outputs, are possible Loconte et al. (2024).

application scenarios of (non-negative) tensor factorizations, the main task is to compress or reconstruct a given tensor, which is generally explicitly represented in memory. Hence, the parameters of the factorization are optimized as to minimize a reconstruction loss (Cichocki et al., 2007). In contrast, modern PCs are *learned from data.* That is, one is given a dataset of $N$ datapoints $\{\mathbf{x}^{(i)}\}_{i=1}^{N}$ that are assumed to be drawn i.i.d. from and *unknown* distribution $p(\mathbf{X})$ (Bishop & Nasrabadi, 2006). The probability tensor that encodes $p(\mathbf{X})$ is therefore implicit and cannot be fully materialized, as the probability distribution is unknown, but also because of its possible exponential size (or even infinite, see Section 3.4).

As learning in PCs often reduces to an optimization problem, i.e., maximizing the data (log-)likelihood (Peharz et al., 2016), enforcing the non-negativity of the circuit is done by using one or more *reparameterizations*, i.e., mapping real-valued parameters to positive sum unit weights. This is necessary as the sum weights of a monotonic PC need to form a convex combination to yield a valid distribution (as shown in Eq. (8)). For instance, we can squash the $K$ parameters $\boldsymbol{\theta} \in \mathbb{R}^K$ of a sum unit with $K$ inputs through a softmax function, i.e. $\mathbf{w} = \mathsf{softmax}(\boldsymbol{\theta})$. Using such a reparameterization together with input functions encoding probability distributions delivers a PC whose normalization constant is 1, as the probabilities of all variable assignments sum up to one. This is direct consequence of having the weights of each sum unit summing up to one. For tensorized circuits, this reparameterization would act row-wise on the parameter matrix of every sum layer.

Luckily, if the circuit is smooth and decomposable (Def. 3), we can still compute its normalization constant exactly and efficiently even if sum weights are not normalized (Peharz et al., 2015). This allows us to use alternative ways to reparameterize a monotonic PC $c$, even if its reparameterization delivers an unnormalized distribution, i.e., a distribution not integrating to 1. In fact, we can still recover a distribution $p(\mathbf{X})$ efficiently via normalization, i.e., $p(\mathbf{X}) = c(\mathbf{X})/Z$ with $Z = \sum_{\mathbf{x} \in \mathsf{dom}(\mathbf{X})} c(\mathbf{x})$ being the normalization constant. For instance, we can enforce each sum unit parameter $\theta$ to be non-negative via exponentiation, i.e. $w = \exp(\theta)$. In this paper, we introduce a third way, a simpler implementation trick that we borrow from the literature on gradient-based optimization for non-negative tensor factorizations (Cichocki et al., 2007): projecting the sum unit parameters in the positive orthant after every optimization step, i.e.,

$$w = \max(\epsilon, \theta), \qquad \theta \in \mathbb{R} \tag{9}$$

where $\epsilon$ is a positive threshold close to zero. Each reparameterization can yield a different loss landscape and lead to different solution during optimization. In our experiments (Section 6), we found this third reparameterization to be the most effective to learn PCs. When it comes to input units in monotonic PCs, they need to model *valid distributions.* Common parameterizations can include simple PMFs (or densities, see Section 3.4) such as Bernoulli or Categorical distributions, or even other probabilistic models as long as they can be tractably marginalized. This yields a set of possible parameterizations that go beyond the simple mappings from indices to matrix entries, as usually used in tensor factorizations (Proposition 1 and Fig. 2).

---

**Opportunity 4. A wide range of possible parameterizations**

Estimating a PMF $p(\mathbf{X})$ via a probabilistic model is another way to perform an implicit tensor compression. If this model is a circuit, then this compression exactly maps to a non-negative hierarchical tensor factorization but over a number of *basis functions*, which are the circuit input units. These input units (thus also input layers in our tensorized formalism) can encode more memory efficient and more expressive functions than indicators. For instance, one can use Binomial distributions instead of categoricals as to drastically reduce the number of parameters of the factorizations (Peharz et al., 2020c). In the case of infinite-dimensional probability tensors (see Section 3.4 below), discrete variables with infinite support can instead be modeled by using Poisson distributions (Molina et al., 2017) or generative models as input layers, such as normalizing flows (Papamakarios et al., 2021; Sidheekh et al., 2023), variational auto-encoders (Tan & Peharz, 2019), or also non-linear functions that can be integrated efficiently, e.g., splines (Novikov et al., 2021; Loconte et al., 2024). Parameterizing input units in this way yields a tensor factorization that uses *non-linearities.* Along this direction, in the circuit literature parameters of sum layers have been directly parameterized by neural networks (Shao et al., 2020; 2022; Gala et al., 2024a). These non-linear cases have only been explored very recently in the matrix and tensor factorization literature (Leplat et al., 2023; Awari et al., 2024).

---

### 3.3 Reliable Neuro-Symbolic Integration

A prominent use case for tractable inference with PCs is in safety-critical applications, where it is necessary to enforce *hard constraints* over the predictions of *neural* classifiers (Ahmed et al., 2022; van Krieken et al., 2024). Such constraints can be expressed as logical formulas over symbols extracted by a perceptual component (a classifier). For example, the safety rule that a self-driving car must stop in front of a pedestrian or a traffic light (Marconato et al., 2024b;a) can be written as a propositional logical formula $\phi : (\text{P} \vee \text{R} \implies \text{S})$, where P, R and S are Boolean variables representing that a Pedestrian and a Red-light have been detected in the video stream of the car and the action to Stop must be taken.

Circuits are especially suitable for this neuro-symbolic integration (De Raedt et al., 2019), because they can represent both probability distributions and logical formulas. These two representations can be used in a single classifier to guarantee that the predictions that will violate the given constraint will always have 0 probability. Formally, we can implement such a classifier, mapping inputs $\mathbf{x}$ to outputs $\mathbf{y}$ that have to satisfy a constraint $\phi$, as (Ahmed et al., 2022):

$$p(\mathbf{y} \mid \mathbf{x}) \propto q(\mathbf{y} \mid \mathbf{x})\mathbb{1}\{\mathbf{y} \models \phi\}, \tag{10}$$

where $q(\mathbf{y} \mid \mathbf{x})$ is a conditional distribution encoded in a circuit that can be parameterized by a neural network (see Opportunity 4) and $\mathbb{1}\{\mathbf{y} \models \phi\}$ is an indicator function that is 1 when the predictions $\mathbf{y}$ satisfy ($\models$) the constraint $\phi$. For instance, $\mathbf{y}$ is a Boolean assignment to variables P, R, S in our self-driving car example, and $\mathbb{1}\{\mathbf{y} \models \phi\}$ is 1 iff substituting $\mathbf{y}$ to variables in $\phi$ yields "true" ($\top$). This indicator function can be compactly represented as a circuit made of sum and product units through a process called *knowledge compilation* (Darwiche & Marquis, 2002; Chavira & Darwiche, 2008; Choi et al., 2013).[4] If both the probability distribution $q$ and the indicator function for the constraint $\phi$ are compatible circuits (Opportunity 2), one can efficiently multiply them and renormalize by computing the partition function (Vergari et al., 2021), which equals the probability that the hard constraint $\phi$ holds given $\mathbf{x}$, i.e.,

$$\sum_{\mathbf{y}} q(\mathbf{y} \mid \mathbf{x})\mathbb{1}\{\mathbf{y} \models \phi\} = \mathbb{E}_{\mathbf{y} \sim q(\mathbf{y}|\mathbf{x})}\left[\mathbb{1}\{\mathbf{y} \models \phi\}\right] = p(\phi = \top \mid \mathbf{x}) \tag{11}$$

also called the *weighted model count* (Chavira & Darwiche, 2008; van Krieken et al., 2024) which is the crucial quantity to compute when combining logical and probabilistic reasoning (Darwiche, 2009; Zeng et al., 2020). This possible integration, as far as we can tell, is off the radar of the tensor factorizations community.

---

**Opportunity 5. Structured sparsity via logical constraints**

Circuits encoding logical formulas are generally very sparse, nonetheless, they still represent a (sparse) factorization of a tensor, in this case a Boolean one. Analogously to the probability tensor described at the beginning of Section 3, this Boolean tensor would encode the logical formula as an exponentially large table of zeros and ones. Multiplying a probability tensor compactly encoded as circuit $q$ as in Eq. (10) with this compact representation of a Boolean tensor equals to a structured form of *masking*: all the invalid (according to the logical constraint $\phi$) entries in the probability tensor are forcefully set to zero, thus making such entries not predictable. A possible opportunity is therefore to connect with the vast literature of knowledge compilation (Darwiche & Marquis, 2002; Choi et al., 2013; Oztok & Darwiche, 2017) to impose structured sparsity to tensor factorizations.

Possible applications include neuro-symbolic integration for graph data (Loconte et al., 2023) as well as representing rankings and user preferences (Choi et al., 2015), scaling cryptographic attacks (Wedenig et al., 2024b), enforcing constraints over the output of LLMs (Zhang et al., 2023) and promoting their self-consistency (Calanzone et al., 2025).

---

[4]Note that arbitrary ANDs and ORs in a logical formula do not directly correspond to products and sums in our circuit language. It is necessary to *compile* the formula in a new representation that contains ANDs over sub-formulas with disjoints scopes – corresponding to decomposable products – and XORs – corresponding to deterministic sum units, and pushes negation towards the input functions (Darwiche & Marquis, 2002).

### 3.4 Infinite-Dimensional Probability Tensors and Continuous Factorizations

Until now, we discussed circuits representing a (hierarchical) factorization of a tensor having finite dimensions, i.e., where the number of entries in every dimension is finite. That is, these circuits are defined over a set of discrete variables, each having a finite number of states. In this section, we focus on factorizations of tensors that can have dimensions having an infinite (and possibly uncountable) number of entries or *quasi-tensors* (Townsend & Trefethen, 2015). Analogously to the symmetry between (hierarchical) tensor factorizations and circuits (Section 2) we show that quasi-tensors can be represented as circuits defined over at least one variable having infinite (and possibly uncountable) domain. Furthermore, by connecting with a very recent class of circuits equipped with integral units, we point out at opportunities regarding the parameterization of infinite-rank (hierarchical) tensor factorizations, i.e., factorizations whose rank is not necessarily finite. We ground these ideas to the problem of modeling a probability density function (PDF).

Formally, let $p(\mathbf{X})$ be a PDF over continuous variables $\mathbf{X} = \{X_j\}_{j=1}^d$, where each $X_j \in \mathbf{X}$ takes values in $\mathsf{dom}(X_j) = \mathbb{R}$. Then, $p(\mathbf{X})$ can be represented as an infinite-dimensional probability tensor $\boldsymbol{\mathcal{T}}$ such that $t(x_1, \ldots, x_d) = p(x_1, \ldots, x_d)$ for any $\mathbf{x} \in \mathsf{dom}(\mathbf{X})$. Infinite-dimensional tensors such as $p(\mathbf{X})$ can be decomposed into a finite number of sums and products of factor matrices that live in Hilbert spaces of generic functions. For instance, we can re-adapt the Tucker factorization shown in Def. 1 as a different factorization method where, instead of having factor matrices $\mathbf{V}^{(j)} \in \mathbb{R}^{I_j \times R_j}$ for all $j$, we encode a vector of $R_j$ functions $\mathcal{F}^{(j)} = \{f_{r_j}^{(j)} \colon \mathsf{dom}(X_j) \to \mathbb{R}\}_{r_j=1}^{R_j}$. That is, we factorize $\boldsymbol{\mathcal{T}}$ as

$$t(x_1, \ldots, x_d) \approx \sum_{r_1=1}^{R_1} \cdots \sum_{r_d=1}^{R_d} w_{r_1 \cdots r_d} f_{r_1}^{(1)}(x_1) \cdots f_{r_d}^{(d)}(x_d). \tag{12}$$

Here, we have $\boldsymbol{\mathcal{W}} \in \mathbb{R}^{R_1 \times \cdots \times R_d}$. Then, one can trivially modify Proposition 1 such that this Tucker factorization of $p(\mathbf{X})$ can be represented as a PC of the same size where the input units over variable $X_j$ now encode the functions in $\mathcal{F}^{(j)}$. Similarly, one can retrieve PCs encoding mixed probability distributions over discrete *and* continuous variables (Molina et al., 2018), thus encoding factorizations of a quasi-tensor. In the same way, one can easily re-adapt hierarchical Tucker to factorize $p(\mathbf{X})$, thus yielding an equivalent deep circuit over continuous variables.

Note that, while Eq. (12) is a factorization of an infinite-dimensional tensor, it is still a *finite* factorization. That is, the ranks $R_1, \ldots, R_d$ are finite, and therefore the circuit representing the same factorization has a sum unit having $R_1 \cdots R_d$ inputs (see Fig. 2). Very recent works have proposed to augment the circuit definition (Def. 2) with *integral units* which, roughly speaking, encode a sum over an infinite and uncountable number of inputs (Gala et al., 2024a;b). We can consider such PCs to encode continuous factorizations of a probability tensor, which can be though of as infinite-rank factorizations. For instance, consider the problem of factorizing a finite-dimensional tensor $\boldsymbol{\mathcal{T}} \in \mathbb{R}^{I_1 \times \cdots \times I_d}$. Instead of considering a finitely-dimensional core tensor $\boldsymbol{\mathcal{W}} \in \mathbb{R}^{R_1 \times \cdots \times R_d}$ in Tucker (Eq. (2)), we can use a function $\omega \colon \mathsf{dom}(\mathbf{Z}) \to \mathbb{R}$ over continuous variables $\mathbf{Z} = \{Z_i\}_{i=1}^d$, where each $Z_i$ has domain $\mathsf{dom}(Z_j) = \mathbb{R}$. Similarly, we replace each factor matrix $\mathbf{V}^{(j)} \in \mathbb{R}^{I_j \times R}$ with a vector of $I_j$ functions $\{f_{i_j}^{(j)} \colon \mathsf{dom}(Z_j) \to \mathbb{R}\}_{i_j=1}^{I_j}$, for all $j$. By doing so and since $\mathbf{Z}$ consists of continuous variables, we are in practice replacing the summations in Eq. (2) with a multivariate integral over $\mathbf{Z}$. That is, we factorize $\boldsymbol{\mathcal{T}}$ as

$$t_{x_1 \cdots x_d} \approx \int_{\mathsf{dom}(\mathbf{Z})} \omega(z_1, \ldots, z_d) \, f_{x_1}^{(1)}(z_1) \cdots f_{x_d}^{(d)}(z_d) \, \mathrm{d}\mathbf{z}. \tag{13}$$

Similarly, one can retrieve hierarchical versions of such continuous tensor factorizations, with applications for probabilistic modeling (Gala et al., 2024b). In case the integral in Eq. (13) is intractable to compute, quadrature rules can be applied as to approximate it. See Gala et al. (2024a) for the details.

In the following section (Section 4), we present a generic pipeline that can be used to build both finite-dimensional and infinite-dimensional hierarchical probability tensor factorizations as deep tensorized PCs (Def. 7). Before that, in the following opportunity box, we stress how circuits can also be used as alternative representations of probability distributions *that do not correspond to probability tensor factorizations.*

> ### Opportunity 6. More factorizations of alternative representations of distributions
>
> Instead of explicitly encoding $p(\mathbf{X})$ by modeling its PMF or PDF one can instead encode its *probability generating function*, *characteristic function* or its *cumulative density function*. Circuits have been used to compactly represent these alternative representations of distributions. For instance, Yu et al. (2023) proposed to build circuits that encode characteristic functions to represent and learn distribution over mixed discrete and continuous data domains. These characteristic circuits have also found application in causal probabilistic inference (Poonia et al., 2024). Similarly to the correspondence between circuits and tensor factorizations shown in the previous sections, a characteristic circuit can be seen as a hierarchical factorization of a tensor encoding a characteristic function, i.e., a factorization of a tensor with complex entries that however still implicitly encodes a probability distribution.

Table 1: **De-structuring circuit and tensor factorization architectures, and their implementations, into simpler design choices** conforming to our pipeline: which region graphs (Section 4.1) and sum-product layers to use (Section 4.3), and whether to apply folding (Section 4.4). New designs are possible by mix & matching these existing base ingredients. Furthermore, we propose new region graphs that deliver more efficient tensorized circuit: QG, QT-2 and QT-4. By leveraging tensor factorizations of the weights of folded circuits, we propose two new sum-product layers: CP, $\text{CP}^{\text{S}}$ and $\text{CP}^{\text{XS}}$. Check mark ✓ means that even if the original implementation of HCLTs does not implement folding as we describe it here, they achieve similar parallelism by custom CUDA kernels. In Appendix B we present a detailed discussion on the design choices of our pipeline that are implicitly made in each PC architecture.

| PC Architecture | Region Graph | Sum-Product Layer | Fold |
|---|---|---|---|
| Poon&Domingos (Poon & Domingos, 2011) | PD | $\text{CP}^{\top}$ | ✗ |
| RAT-SPN (Peharz et al., 2020c) | RND | Tucker | ✗ |
| EiNet (Peharz et al., 2020a) | { RND, PD } | Tucker | ✓ |
| HCLT (Liu & Van den Broeck, 2021b) | CL | $\text{CP}^{\top}$ | ✓ |
| HMM/MPS$_{\mathbb{R}_{\geqslant 0}}$ (Glasser et al., 2019) | LT | $\text{CP}^{\top}$ | ✗ |
| BM (Han et al., 2018) | LT | $\text{CP}^{\top}$ | ✗ |
| TTDE (Novikov et al., 2021) | LT | $\text{CP}^{\top}$ | ✗ |
| NPC$^2$ (Loconte et al., 2024) | { LT, RND } | { $\text{CP}^{\top}$, Tucker } | ✓ |
| TTN (Cheng et al., 2019) | QT-2 | Tucker | ✗ |
| Mix & Match (our pipeline) | $\left\{ \begin{array}{l} \text{RND, PD, LT,} \\ \text{CL, QG, QT-2, QT-4} \end{array} \right\}$ $\times$ | $\{ \text{Tucker, CP, CP}^{\top} \} \cup$ $\{ \text{CP}^{\text{S}}, \text{CP}^{\text{XS}} \mid \text{Fold } ✓ \}$ | $\times \{ ✗, ✓ \}$ |

## 4 How to Build and Scale Circuits: A Tensorized Perspective

We now have all the necessary background to start exploiting the connections between (hierarchical) tensor factorizations and (deep) circuits. In particular, in this section, we will show how we can understand and unify many—apparently different—ways to build circuits (and other factorizations) in a single pipeline leveraging tensor factorizations as modular abstractions. By doing so, we can "disentangle" what are the key ingredients to build and effectively learn overparameterized circuits, i.e., circuits with a very large number of parameters (Table 1).

Fig. 9 summarizes our pipeline: **i)** first, one builds a RG structure to enforce the necessary structural properties (Section 4.1), then, **ii)** populates such a template by introducing units and grouping them into layers (Section 4.2), following the many possible tensor factorizations abstractions (Section 4.3), optionally, **iii)** these layers can be "folded", i.e., stacked together to exploit GPU parallelism (Section 4.4). Finally, the circuit parameters can be optimized by gradient descent or expectation maximization (Peharz et al., 2016; Zhao et al., 2016).

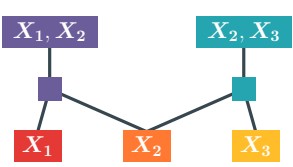

(**i**) Build a region graph.

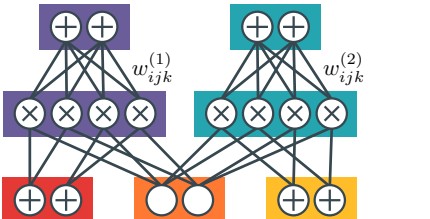
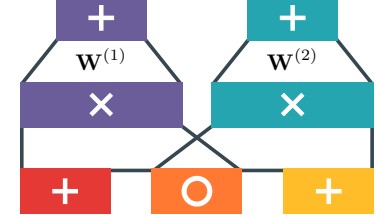

(**ii**) Overparameterize & tensorize.

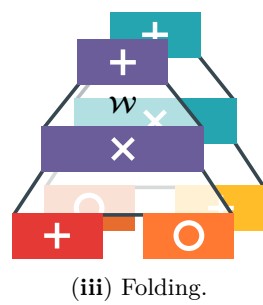

(**iii**) Folding.

Figure 9: **Our pipeline for building overparameterized circuits.** Given a (fragment of) region graph (i), we overparameterize it with sum, product and input units. In this case, the connections between sum and product units encode a Tucker factorization (e.g., as in Fig. 2). Then, we tensorize it by grouping units into layers (ii). In the final folding step, we can fuse together those layers that can be evaluated in parallel (iii). To do so, we stack the parameter matrices $\mathbf{W}^{(1)}, \mathbf{W}^{(2)}$ into a tensor $\mathcal{W}$.

## 4.1 Building and Learning Region Graphs

The first step of our pipeline is to construct a RG (Def. 4). It specifies a hierarchical partitioning of the input variables according to which we build deep circuit architectures. In particular, PCs that are built out of RGs satisfying crucial structural properties such as smoothness and decomposability by design (and structured-decomposability if the RG is a tree and has univariate leaves, see Section 2.2), which in turn guarantee tractable inference for many queries of interest (Section 2). RGs are explicitly used to build PCs in some papers (Peharz et al., 2020c;a), but as we show next, they can be implicitly found in many other PC and tensor factorization architectures. We also introduce a novel way to quickly build RGs for images that are dataset-agnostic but exploit the structure of pixels.

**Linear tree RGs (LT).** A simple way to instantiate a RG is by building partitionings that factorize one variable at a time. That is, given an ordering $\pi$ over variables $\mathbf{X}$, each $i$-th partition node factorizes its scope $\{X_{\pi(1)}, \ldots, X_{\pi(i)}\}$ into regions $\{X_{\pi(1)}, \ldots, X_{\pi(i-1)}\}$ and $\{X_{\pi(i)}\}$. We call the resulting RG a linear tree (LT) RG, and show an example for it in case of three variables in Fig. 3. The ordering of the variables can be the lexicographical one or depending on additional information such as time when modeling sequence data. This sequential RG is the one adopted by chain-like tensor network factorizations, such as MPS, TTs or BMs (Pérez-García et al., 2007; Oseledets, 2011), as well as hidden Markov models (HMMs) when represented as PCs (Rabiner & Juang, 1986; Liu et al., 2023a).

**Randomized tree RGs (RND).** A slightly more sophisticated way to build a RG is to construct a tree that is balanced. This can be done in a dataset-agnostic way by randomly partitioning variables recursively. That is, the root region $\mathbf{X}$ is recursively partitioned by randomly splitting variables in approximately even subsets, until no further partitionings are possible. This approach, which we label with RND, has been introduced to build *randomized-and-tensorized sum-product networks* (RAT-SPNs) (Peharz et al., 2020c). A similar approach has been described by Di Mauro et al. (2017; 2021), with the difference that some randomly-chosen subsets of the data are also taken into account when parameterizing the circuit, thus entangling the construction of the RG with the circuit parameterization.

**Poon-Domingos construction (PD).** One can devise other RG algorithms that are tailored for specific data modalities, but that are still dataset-agnostic. In the case of images where variables are associated to pixel values, Poon & Domingos (2011) proposed to split them as to form a deep hierarchy of patches, by recursively performing horizontal and vertical cuts. However, the main drawback of this approach, labeled PD, is that it generally yields very deep circuit architectures that are hard to optimize (Section 6), as it

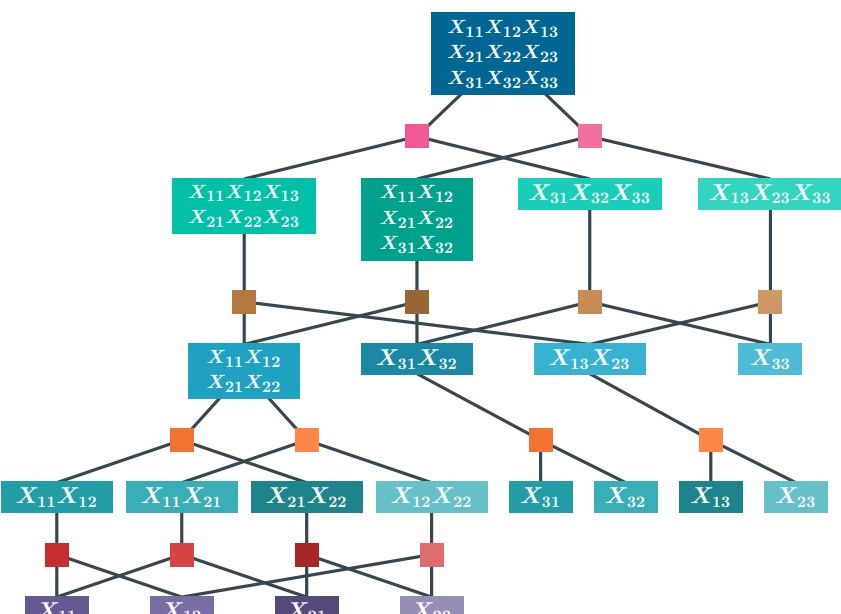

Figure 10: **The quad graph (QG).** We illustrate the quad graph RG delivered by Algorithm D.1 passing $H = 3, W = 3$ and isTree = False as input arguments. The region graph is *unbalanced* as the image size ($3 \times 3$) is not a power of 2. Differently from our quad trees (QTs), QGs have regions partitioned in more than a single way (e.g., the root region node), and regions can be shared among partitions. For example, in a QT, the top region could only be partitioned in a single way into two or four subregions, respectively called QT-2 and QT-4 region graphs.

considers *all* the possible ways to recursively split an image into patches whose number grows fast with respect to the image size. The PD RG has been extensively used in the circuit literature, e.g., for architectures like EiNets (Peharz et al., 2020a).

**Novel RGs for image data: quad graphs (QG) and trees (QT).** We want to devise RGs that are dataset-agnostic but still aware of the pixel structure as PD, while at the same time not falling prey of the same optimization issues. Therefore, we propose a much simpler way to construct image-tailored RGs that delivers smaller circuits that can achieve better performances, even when compared to RGs learned from data (see Section 6). Algorithm D.1 in the Appendix details our construction. Similarly to PD, it builds a RG by recursively splitting image patches of approximately the same size, but differently from PD it only splits them into four parts (a one vertical and horizontal cut) sharing the newly created patches. We call such RG *quad-graph* (QG). Fig. 10 shows an example of a QG RG for a 3x3 image.

Alternatively, one can obtain a tree RG by splitting the patches both horizontally and vertically, but without sharing patches. We call such tree RG *quad-tree* (QT). Since regions of such RGs are associated to image patches, we can choose to partition them in different ways. In particular, we will denote with QT-2 a QT whose regions are partitioned in two parts (bottom and top parts of the patch), and with QT-4 a QT whose regions are partitioned into four parts (following a quadrant division partitioning). With QT-2 we retrieve tensor factorizations tailored for image-data used in prior work (Cheng et al., 2019).

**Learning RGs from data.** The approaches discussed so far do not depend on the training data. To exploit the data in the construction of RGs, one can test the statistical independence of subset of features inside a region node $\mathbf{Y} \subseteq \mathbf{X}$. This is the approach used in the seminal LearnSPN algorithm (Gens & Domingos, 2013), later extended in many other works (Molina et al., 2018; Di Mauro et al., 2019). All these variants never mention a RG, but one is built implicitly by performing these statistical test and by introducing regions that are associated to a different "chunk" of data obtained by clustering (Vergari et al., 2015). Alternatively, one can split regions according to some heuristics over the data that result in region nodes being shared (Jaini et al., 2018a). The same idea is at the base of the Chow-Liu algorithm to learn the tree-shaped PGM that better approximates the data likelihood (Chow & Liu, 1968b). The Chow-Liu algorithm (CL) can be used to implicitly build a RG as well, as done in many structure learning variants (Vergari et al., 2015; Rahman et al., 2014; Choi et al., 2011). A more recent approach that leverages this idea and that generally yields state-of-the-art performance first learns the Chow-Liu tree, then treats it as a latent tree model (Choi et al., 2011) that is finally compiled into a PC (Liu & Van den Broeck, 2021b).

---

**Algorithm 1** overparamAndTensorize($\mathcal{R}, \mathcal{F}, K$)

---

**Input:** a RG $\mathcal{R}$ over variables $\mathbf{X}$, the sum layers width $K$, and the type of input functions $\mathcal{F}$.
**Output:** A tensorized circuit $c$ over $\mathbf{X}$.

1: $\mathcal{L} \leftarrow$ emptyMap $\qquad \triangleright$ From regions to layers
2: **for each** region $\mathbf{Y} \in$ postOrderTraversal($\mathcal{R}$) **do**
3: $\quad$ **if** $\mathbf{Y}$ is partitioned into $\{(\mathbf{Z}_1^{(i)}, \mathbf{Z}_2^{(i)})\}_{i=1}^N$ **then**
4: $\qquad \Lambda \leftarrow \varnothing$
5: $\qquad C \leftarrow 1$ **if** $\mathbf{Y} = \mathbf{X}$ **else** $K$
6: $\qquad$ **for** $i = 1$ **to** $N$ **do**
7: $\qquad\quad \ell \leftarrow$ SumProdLayer($\mathcal{L}[\mathbf{Z}_1^{(i)}], \mathcal{L}[\mathbf{Z}_2^{(i)}], C, K$)
8: $\qquad\quad \Lambda \leftarrow \Lambda \cup \{\ell\}$
9: $\qquad \mathcal{L}[\mathbf{Y}] \leftarrow$ pop($\Lambda$) **if** $|\Lambda|{=}1$ **else** SumLayer($\Lambda$)
10: $\quad$ **else** $\qquad\qquad \triangleright \mathbf{Y}$ is a leaf region in $\mathcal{R}$
11: $\qquad \mathcal{L}[\mathbf{Y}] \leftarrow$ InputLayer($\mathbf{Y}, K, \mathcal{F}$)
12: **return** A circuit having $\mathcal{L}[\mathbf{X}]$ as output layer

---

**Algorithm 2** SumProdLayer($\ell_1, \ell_2, C, K$)

---

**Input:** Layers $\ell_1, \ell_2$ with width $K$ and output width $C$.
**Output:** A composition of sum & product layers.

1: **procedure** (parameterizeTucker)
2: $\quad$ Let $\mathbf{W} \in \mathbb{R}^{C \times K^2}$ be the sum layer parameters
3: $\quad$ **return** $\ell$ computing $\mathbf{W}\left(\ell_1(\mathbf{Z}_1) \otimes \ell_2(\mathbf{Z}_2)\right)$

4: **procedure** (parameterizeCP)
5: $\quad$ Let $\mathbf{Q}^{(1)}, \mathbf{Q}^{(2)} \in \mathbb{R}^{C \times K}$ be the parameters
6: $\quad$ **return** $\ell$ computing $(\mathbf{Q}^{(1)} \ell_1(\mathbf{Z}_1)) \odot (\mathbf{Q}^{(2)} \ell_2(\mathbf{Z}_2))$

---

**Algorithm 3** SumLayer($\{\ell_i\}_{i=1}^N$)

---

**Input:** Input layers $\{\ell_i\}_{i=1}^N$ having scope $\mathbf{Y}$ and width $K$, with $N > 1$. **Output:** A sum layer.

1: Let $\mathbf{W} \in \mathbb{R}^{K \times (NK)}$ be the sum layer parameters
2: **return** $\ell$ computing $\mathbf{W}( \|_{i=1}^N \ell_i(\mathbf{Y}))$

---

The construction of this *hidden* Chow-Liu *tree* (HCLT) exactly follows the steps in our pipeline, once one disentangles the role of the RG from the rest.

The construction of other PC and tensor factorization architectures mentioned so far (i.e., RAT-SPNs, EiNets, MPSs, BMs, etc) also follows the same pattern, and can be easily categorized in our pipeline (Table 1). They not only differ in terms of the RGs they are built from, but also on the kind of the chosen sum and product layers. In the next section, we provide a generic algorithm that builds a tensorized circuit architecture from a given RG, given a selection of sum and product layers encoding tensor factorizations.

## 4.2 Overparameterize & Tensorize Circuits

Given a RG, the simplest way to build a circuit is to associate a single input distribution unit per leaf region, a single sum per inner region, and an single product unit per partition, and then connect them following the RG structure. This would deliver a smooth and (structured-)decomposable circuit that is sparsely connected, and it is in fact the strategy that the many structure learning algorithms discussed in the previous section were implicitly using (Gens & Domingos, 2013; Vergari et al., 2015; Molina et al., 2018). We can adapt this strategy to the "deep learning recipe", and output instead an *overparameterized* circuit that is locally densely-connected. With overparameterization we refer to the process of "populating" a RG with not one but many sum, product and input units of the same scope. The resulting tensorized computational graph (Def. 7) has many more learnable parameters and lends itself to be parallelized on GPU, as we can vectorize computational units sharing the same scope as to form dense layers. Algorithm 1 details the overparameterization and tensorization process. The algorithm takes as input: a RG $\mathcal{R}$, the type of input functions $\mathcal{F}$ (e.g., Gaussians), and the number of sum units $K$ which governs the expressiveness of the circuit, or equivalently the rank of the factorization.[5] Furthermore, we can customize the choice of input layers as well as how to stack sum and product layers together, yielding many ways to build circuits with different degrees of efficiency and expressiveness.

**Constructing input layers.** The first step of Algorithm 1 consists of associating input units to leaf regions, i.e., regions that are not further decomposed. Leaf regions are often univariate, i.e., of the form $\mathbf{Y} = \{X_j\}$ for some variable $X_j \in \mathbf{X}$. For each leaf region over a variable $X_j$ we introduce $K$ input units, each computing a function $f_i : \text{dom}(X_j) \to \mathbb{R}$. To guarantee the non-negativity of the output in monotonic PCs, $f_i$ are often chosen to be non-negative, e.g., by choosing them to be probability mass or density functions (Choi et al., 2020). However, one can possibly choose $f_i$ from a much wider set of expressive function families, e.g., polynomial splines (de Boor, 1971; Loconte et al., 2024), neural networks (Shao et al., 2020; Correia et al.,

---

[5]As in a (hierarchical) Tucker factorization, we can select $d$ different numbers of units, one for each layer, thus encoding a $K_1, \ldots, K_d$-rank factorization. For simplicity, we assume that $K_1 = K_2 = \ldots = K_d$.

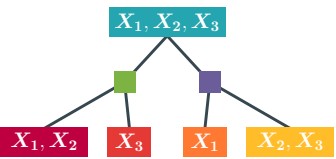
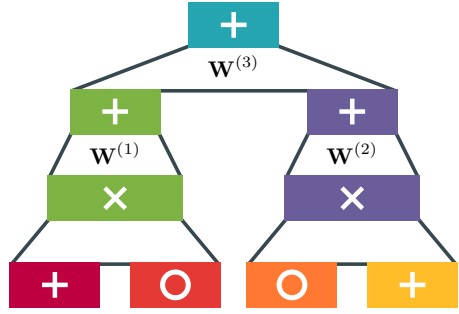

Figure 12: **A region node split into two partitionings** $(\{X_1, X_2\}, \{X_3\})$ and $(\{X_1\}, \{X_2, X_3\})$ of $\{X_1, X_2, X_3\}$ (above) is overparameterized using Tucker layers having parameters $\mathbf{W}^{(1)}, \mathbf{W}^{(2)} \in \mathbb{R}^{K \times K^2}$, and with an additional sum layer parameterized by $\mathbf{W}^{(3)} \in \mathbb{R}^{S \times (2K)}$, for some $S > 0$ (right).

2023; Gala et al., 2024a;b) and normalizing flows (Sidheekh et al., 2023). See also Opportunity 4. Then, the input units can be tensorized by effectively replacing them with an input layer $\boldsymbol{\ell} \colon \mathsf{dom}(X_j) \to \mathbb{R}^K$ such that $\boldsymbol{\ell}(X_j)_i = f_i(X_j)$ with $i \in [K]$ can be computed in parallel (L11 in Algorithm 1). Next, sum and product layers are built and connected according to the variables partitioning specified in the given RG.

### 4.3 Abstracting Sum and Product Layers into Modules

Alongside input layers, we introduced the other atomic "Lego blocks" for tensorized circuits in Def. 7: sum layers, Hadamard and Kronecker product layers. In the following, we will use these blocks to create *composite* layers that will act as further abstractions that can be seamlessly plugged in Algorithm 1. These composite layers include: ***Tucker*** (Fig. 11), ***CP*** (Fig. 15) and $\boldsymbol{CP}^\top$ (Fig. 16) layers. Each of these layers encodes a local factorization and stacks and connects internal sum and product units in a different way as to increase expressiveness or efficiency.

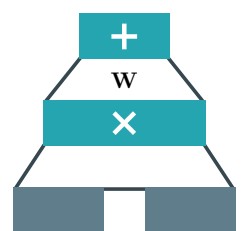

Note that given our semantics for tensorized layers, stacking these composite abstractions by applying Algorithm 1 over a RG will always output a tensorized circuit that is smooth and (structured-)decomposable (Def. 8).

We start by considering composite layers that adopt the connectivity of computational units in the Tucker factorization, as shown in Fig. 2. This is a pattern introduced in architectures such as RAT-SPNs (Peharz et al., 2020c) and EiNets (Peharz et al., 2020a). There, a region node over $\mathbf{Y} \subseteq \mathbf{X}$ and partitioned into $(\mathbf{Z}_1, \mathbf{Z}_2)$ is parameterized as a layer $\boldsymbol{\ell}$ that is a composition of a Kronecker product layer followed by a sum layer, i.e., computing

Figure 11: Tucker layer

$$\boldsymbol{\ell}(\mathbf{Y}) = \mathbf{W}\left(\boldsymbol{\ell}_1(\mathbf{Z}_1) \otimes \boldsymbol{\ell}_2(\mathbf{Z}_2)\right), \qquad \text{(Tucker-layer)}$$

where $\mathbf{W} \in \mathbb{R}^{K \times K^2}$ is the parameter matrix for a given number of units $K$, and $\boldsymbol{\ell}_1, \boldsymbol{\ell}_2$ are its input layers (in grey in Fig. 11), each computing a $K$-dimensional vector obtained via overparameterization and tensorization of region nodes over $\mathbf{Z}_1, \mathbf{Z}_2$, respectively. Algorithm 2 composes sum and product layers as to construct Eq. (Tucker-layer), and it is called to overparameterize the circuit (see L6-8 in Algorithm 1). However, note that the flexibility provided by Algorithm 1 allows us to define other possible parameterizations in Algorithm 2, without changing the rest of the algorithm.

**Overparameterizing multiple partitionings, the Mixing layer case.** Some RGs can have multiple partitionings for a same region, as shown in Fig. 12. More formally, given a region node $\mathbf{Y} \subseteq \mathbf{X}$, we can split it into $N > 1$ different partitioning, i.e., $\{(\mathbf{Z}_1^{(i)}, \mathbf{Z}_2^{(i)})\}_{i=1}^N$, with $\mathbf{Y} = \mathbf{Z}_1^{(i)} \cup \mathbf{Z}_2^{(i)}$ for every $i$. This is the case of the PD RG used in EiNets (Peharz et al., 2020a), and the proposed QG (see Section 4.1 and Fig. 10). We illustrate an example of such RG in Fig. 12. The design adopted in EiNets to overparameterize them is to build an *apparently* special layer called *mixing layer* by Peharz et al. (2020a) computing

$$\boldsymbol{\ell}(\mathbf{Y}) = \sum_{i=1}^N \mathbf{w}_{i:} \odot \boldsymbol{\ell}_i(\mathbf{Y}) \qquad \text{(Mixing-layer)}$$

where $\mathbf{W} \in \mathbb{R}^{N \times K}$ denote the parameter matrix of $\boldsymbol{\ell}$, and each $\boldsymbol{\ell}_i$ is a layer that outputs a $K$-dimensional vector. However, we observe that Eq. (Mixing-layer) can be computed by a simple sum layer that already conforms to our Def. 7. In fact, Eq. (Mixing-layer) can be rewritten as $\boldsymbol{\ell}(\mathbf{Y}) = \mathbf{W}'(\|_{i=1}^{N} \boldsymbol{\ell}_i(\mathbf{Y}))$, where $\mathbf{W}' \in \mathbb{R}^{K \times (NK)}$ is the parameter matrix obtained by concatenating $N$ diagonal matrices $\{\mathbf{W}'_i\}_{i=1}^{N}$ along the columns, with $\mathbf{W}'_i \in \mathbb{R}^{K \times K}$ for $i \in [K]$. This observation demystifies the need of treating mixing layers as yet another type of layers, which happens to be the case in current EiNets implementations (Peharz et al., 2020a; Braun, 2021). Algorithm 3 specifies the construction of the generalization of the mixing layer as a single sum layer (used in L9 of Algorithm 1). Section 4.3 illustrates the overparameterization and tensorization of a region being decomposed in more than one partitioning. Note that from our perspective it becomes clear that such a sum layer does not necessarily increase the expressiveness, i.e., at the scalar unit level one could merge connected sum units as a single sum unit. For this reason, in Section 6 we experiment with mixing layers whose parameter entries are fixed during learning.[6]

## 4.4 Folding to Further Accelerate Learning and Inference

The final and optional step of our proposed pipeline (Fig. 9) consists of stacking together the layers that share the same functional form as to increase GPU parallelism. We name this step *folding*. Note that folding is only a syntactic transformation of the circuit, i.e., it does not change the encoded function and hence it preserves its expressiveness. This simple syntactic "rewriting" of a circuit can however significantly impact learning and inference performance. In fact, folding is the core ingredient of the additional speed-up introduced by EiNets (Peharz et al., 2020a) with respect to the same non-folded circuit architectures such as RAT-SPNs (Peharz et al., 2020c) which share with EiNets the other architecture details, e.g., the use of Tucker layers (see Table 1). As such, the difference in performance that is usually reported when treating RAT-SPNs and EiNets as two different PC model classes (see e.g., Liu et al. (2023a)) must depend on other factors, such as the choice of the RG or a discrepancy in other hyperparameters used to learn these models, e.g., the chosen optimizer. By disentangling these aspects in our pipeline, we can design experiments that truly highlight which factors are responsible for increased performance (see Section 6).

**Folding layers.** To retrieve the folded representation of the Tucker layer (Eq. (Tucker-layer)), we need to stack the parameter matrices along a newly-introduced dimension, which we call *the fold dimension*. Then, we can compute products accordingly to such extra dimension. For instance, given a set $\{\boldsymbol{\ell}^{(n)}\}_{n=1}^{F}$ of Tucker layers having scopes $\{\mathbf{Y}^{(n)}\}_{n=1}^{F}$, respectively, we evaluate them *in parallel with a single folded layer* $\boldsymbol{\ell}$ computing a $F \times K$ matrix and defined as

$$\boldsymbol{\ell}\left(\bigcup_{n=1}^{F} \mathbf{Y}^{(n)}\right)_{n:} = \mathbf{W}_{n::}\left[\boldsymbol{\ell}_1\left(\bigcup_{n=1}^{F} \mathbf{Z}_1^{(n)}\right)_{n:} \otimes \boldsymbol{\ell}_2\left(\bigcup_{n=1}^{F} \mathbf{Z}_2^{(n)}\right)_{n:}\right] \qquad \text{with } n \in [F] \qquad \text{(Tucker-folded)}$$

where $\boldsymbol{\ell}_1$ (resp. $\boldsymbol{\ell}_2$) denotes a folded layer computing the $F$ left (resp. right) inputs to $\boldsymbol{\ell}^{(n)}$, each defined over variables $\mathbf{Z}_1^{(n)}$ (resp. $\mathbf{Z}_2^{(n)}$), and each $\mathbf{W}_{n::} \in \mathbb{R}^{K \times K^2}$ is the parameter matrix of $\boldsymbol{\ell}^{(n)}$. In other words, $\mathbf{W}_{n::}$ is the $n$-th slice along the first dimension of a tensor $\mathcal{W} \in \mathbb{R}^{F \times K \times K^2}$ obtained by stacking together the parameter matrices of each Tucker layer. Since the same region node can possibly take part in multiple partitionings of other region nodes (e.g., see Fig. 9i), we might have folded inputs $\boldsymbol{\ell}_1, \boldsymbol{\ell}_2$ computing the same outputs. We illustrate an example of this in Fig. 9iii, which shows the folding of two Tucker sum-product layers sharing one input. In Appendix F, we report a pytorch snippet implementing a folded Tucker layer with an einsum operation. For this reason, while folding provides considerable speed-up when evaluating a tensorized circuit, it might come at the cost of increased memory usage depending on the chosen RG.

**How to choose the layers to fold?** It remains to decide how to choose the layers to fold together. The simplest way is traversing the tensorized circuit top-down (i.e., from the outputs towards the inputs) and to fold layers located at the same depth in the computational graph. However, note that we can also fold layers at different depth. For example, all input layers can be folded together if they encode the same input functional for all variables. This is the approach adopted in EiNets (Peharz et al., 2020a) and the one that will be used in all our experiments and benchmarks (see Section 6). However, note that this is

---

[6]And we found that empirically this speeds up learning and does improve performances a bit.

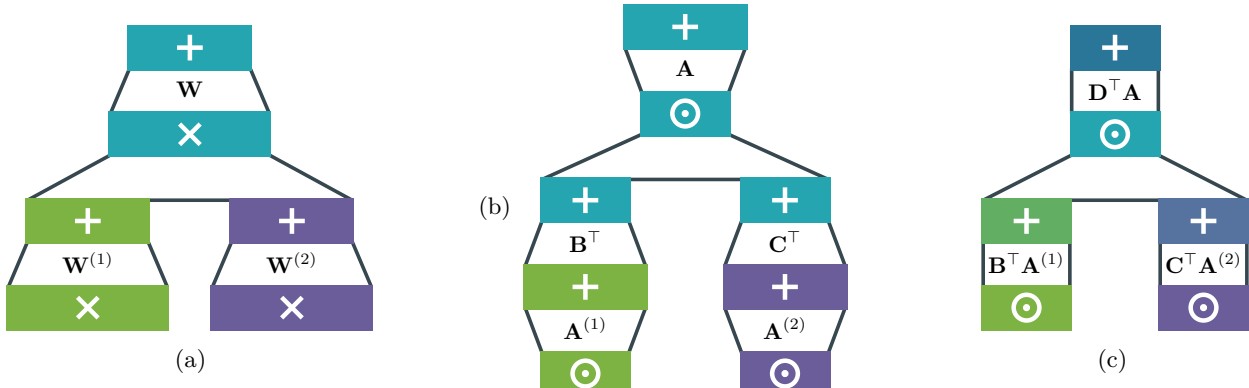

Figure 14: **Compressing Tucker layers into CP layers.** Given a (fragment of) tensorized circuit equipped with Tucker layers (a), we compress it by computing a CP factorization for each parameter matrix $\mathbf{W}^{(1)}$, $\mathbf{W}^{(2)}$, and $\mathbf{W}$. By doing so, we recover a different parameterization given by the factor matrices of the CP factorizations, and the product layers now compute a Hadamard product of their inputs (b). Finally, we can simplify the circuit by *collapsing* consecutive sum layers (c). The circuit structure showed in (a) is typical of RAT-SPNs and EiNets architectures, while the one in (c) captures the connectivity in HCLTs, MPS/TTs, BMs and more (Table 1), since it interleaves Hadamard product and sum layers (e.g., see MPS/TT in Fig. 8).

not regarded as the optimal way to fold layers, and different ways of choosing the layers to fold might bring additional speed-up and memory savings when tailored for specific architectures. While we do not investigate different ways of folding layers other than the one mentioned above, the disentanglement of the folding and overparameterization steps (Section 4.2) in our proposed pipeline will foster future work to rely on the wide literature on parallelizing generic computational graphs (Shah et al., 2023).

# 5 Compressing Circuits and Sharing Parameters via Tensor Decompositions

In this section, we exploit again the literature of tensor factorizations to improve the design and learning of circuit architectures. We start by observing that as the parameters in circuit layers in our pipeline are stored in large tensors (see e.g., Eqs. (Tucker-layer) and (Tucker-folded)) they *can in principle be factorized again.* And since factorizations are circuits (Proposition 1), in the end we obtain several variants of circuit architectures and layers, some of which are new and offer an interesting trade-off between speed and accuracy (Section 5.2), while others are implicitly being used in the construction of existing circuits and tensor factorizations (Table 1). Again, we start from Tucker layers, with the aim of *compressing* a deep circuit using them, i.e., approximating it by using less parameters.

## 5.1 Compressing Tucker layers

Although expressive, Tucker layers in circuits require learning and storing $K^3$ parameters, encoded in the matrix $\mathbf{W} \in \mathbb{R}^{K \times K^2}$ in Eq. (Tucker-layer), which can be reshaped as the three dimensional tensor $\mathcal{W} \in \mathbb{R}^{K \times K \times K}$. More in general, by relaxing the assumption of binary RGs made so far, a Tucker layer taking $N$ input layers will be parameterized by $K^{N+1}$ parameters. To retrieve a more space efficient parameterization of a Tucker layer, we propose to compress its parameter tensor $\mathcal{W}$ via a rank-$R$ *canonical polyadic* (CP) factorization, which we define below.

**Definition 10** (CP factorization (Carroll & Chang, 1970)). Let $\mathcal{T} \in \mathbb{R}^{I_1 \times \cdots \times I_d}$ be a $d$-dimensional tensor. The rank-$R$ *canonical polyadic* (CP) of $\mathcal{T}$ factorizes it as a sum of $R$ rank-1 tensors, i.e.,

$$\mathcal{T} \approx \sum_{r=1}^{R} \mathbf{v}_{:r}^{(1)} \circ \cdots \circ \mathbf{v}_{:r}^{(d)} \qquad \text{or in element-wise notation} \qquad t_{i_1 \cdots i_d} \approx \sum_{r=1}^{R} v_{i_1 r}^{(1)} \cdots v_{i_d r}^{(d)} \qquad (14)$$

where $\mathbf{V}^{(j)} \in \mathbb{R}^{I_j \times R}$ with $j \in [d]$ are factor matrices.

Note that a CP factorization can be represented as a circuit as it is a special case of the Tucker factorization (Proposition 1). With this in mind, we proceed by decomposing the $\boldsymbol{\mathcal{W}} \in \mathbb{R}^{K \times K \times K}$ parameter tensor of a Tucker layer via a rank-$R$ CP factorization such that $R \ll K$, i.e.,

$$\boldsymbol{\mathcal{W}} \approx \sum_{r=1}^{R} \mathbf{a}_{:r} \circ \mathbf{b}_{:r} \circ \mathbf{c}_{:r} \qquad \text{or in element-wise notation} \qquad w_{ijk} \approx \sum_{r=1}^{R} a_{ir} b_{jr} c_{kr} \tag{15}$$

where $\mathbf{A}, \mathbf{B}, \mathbf{C} \in \mathbb{R}^{K \times R}$ are newly-introduced parameter matrices. This new parameterization requires only $3KR$ parameters and unlocks a faster evaluation of Tucker layers. That is, we can rewrite the function computed by a Tucker layer $\boldsymbol{\ell}$ in element-wise notation as

$$\boldsymbol{\ell}(\mathbf{Y}) = \mathbf{A} \left[ \left( \mathbf{B}^{\top} \boldsymbol{\ell}_1(\mathbf{Z}_1) \right) \odot \left( \mathbf{C}^{\top} \boldsymbol{\ell}_2(\mathbf{Z}_2) \right) \right] \tag{16}$$

where $\boldsymbol{\ell}_1, \boldsymbol{\ell}_2$ are input layers to $\boldsymbol{\ell}$ having width $K$ and scopes $\mathbf{Z}_1, \mathbf{Z}_2$, respectively.

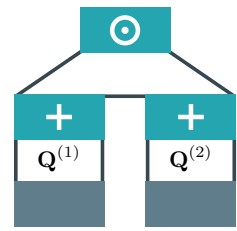

Figure 15: CP layer.

Therefore, evaluating a compressed Tucker layer that has undergone the CP factorization requires time $\mathcal{O}(KR)$ (Eq. (16)), rather than $\mathcal{O}(K^3)$ (Eq. (Tucker-layer)). On top of this, we observe that if we use a CP factorization to all Tucker layers in a PC, we will obtain a circuit in which sum and product layers are not alternated anymore. For example, starting from the Tucker layers in Fig. 14a, we would obtain a new architecture where product layers can be followed by two sum layers, as in Fig. 14b, e.g., one parameterized by $\mathbf{A}^{(1)} \in \mathbb{R}^{K \times R}$ feeding another sum layer parameterized by $\mathbf{B}^{\top} \in \mathbb{R}^{R \times K}$. As we can always rewrite any composition of consecutive sum layers with a single sum layer parameterized by a product of matrices (e.g., by $\mathbf{B}^{\top} \mathbf{A}^{(1)} \in \mathbb{R}^{R \times R}$), we can *collapse* the adjacent sum layers as to obtain the simplified architecture in Fig. 14c. More formally, under such observation and by assuming that $\boldsymbol{\ell}_1, \boldsymbol{\ell}_2$ are also Tucker layers being decomposed, we can rewrite Eq. (16) as

$$\boldsymbol{\ell}(\mathbf{Y}) = \left( \mathbf{Q}^{(1)} \boldsymbol{\ell}_1(\mathbf{Z}_1) \right) \odot \left( \mathbf{Q}^{(2)} \boldsymbol{\ell}_2(\mathbf{Z}_2) \right) \tag{CP-layer}$$

where $\mathbf{Q}^{(1)}, \mathbf{Q}^{(2)} \in \mathbb{R}^{R \times R}$ are parameter matrices of sum layers, such that $\mathbf{Q}^{(1)} = \mathbf{B}^{\top} \mathbf{A}^{(1)}$, $\mathbf{Q}^{(2)} = \mathbf{C}^{\top} \mathbf{A}^{(2)}$. That is, we reduced the overall width of each layer from $K$ to the smaller $R$ while still approximately computing a Tucker layer, by assuming that $\boldsymbol{\mathcal{W}}$ was originally low-rank. From now on, we will refer to Eq. (CP-layer) as CP layer. This is a new compositional abstraction we can use instead of Tucker layers in Algorithm 2 to build tensorized circuits out of a RG. For monotonic PCs, one can still recover the Tucker layer factorization above by replacing the CP factorization (Eq. (15)) with its non-negative version (Cichocki & Phan, 2009), which ensures the factors $\mathbf{A}, \mathbf{B}, \mathbf{C}$ and hence $\mathbf{Q}^{(1)}, \mathbf{Q}^{(2)}$ to be non-negative matrices. Furthermore, a folded version of Eq. (CP-layer) can be obtained similarly to the one for Eq. (Tucker-layer) (see Section 4.4).

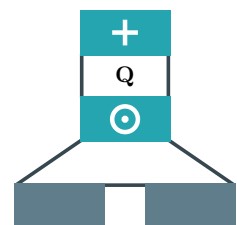

Figure 16: $\mathrm{CP}^{\top}$ layer.

Finally, we introduce another type of layer which is very similar to the CP-layer above except that the Hadamard product is performed before the vector-matrix multiplication. We denote this sum-product layer as $\mathrm{CP}^{\top}$, spelled *CP-transpose* or *CP-T*. Formally, a $\mathrm{CP}^{\top}$ layer $\boldsymbol{\ell}$ computes

$$\boldsymbol{\ell}(\mathbf{Y}) = \mathbf{Q} \left( \boldsymbol{\ell}_1(\mathbf{Z}_1) \odot \boldsymbol{\ell}_2(\mathbf{Z}_2) \right), \tag{$\mathrm{CP}^{\top}$-layer}$$

where $\mathbf{Q} \in \mathbb{R}^{R \times R}$. The main difference between using CP and $\mathrm{CP}^{\top}$ layers is when these are applied on top of input layers, as there might be a slight difference in expressiveness. For instance, the product of two mixtures of Gaussians is different from a mixture of the product of two Gaussians.

Architectures such as HCLTs are latent tree models (Choi et al., 2011) and as such they can be rewritten as tensorized circuits using $\mathrm{CP}^{\top}$ layers (Table 1) plus one additional sum layer, as illustrated in Fig. 18. More specifically, since HCLTs are monotonic circuits, we can interpret each of the parameter matrices $\mathbf{Q} \in \mathbb{R}_{+}^{R \times R}$ in Eq. ($\mathrm{CP}^{\top}$-layer) as conditional probability tables[7] of the form $p(Z_i \mid Z_j)$ with latent variables $Z_i, Z_j$

---

[7]The term $\mathrm{CP}^{\top}$ is indeed a pun on the term conditional probability tables (CPTs) in the Bayesian network terminology.

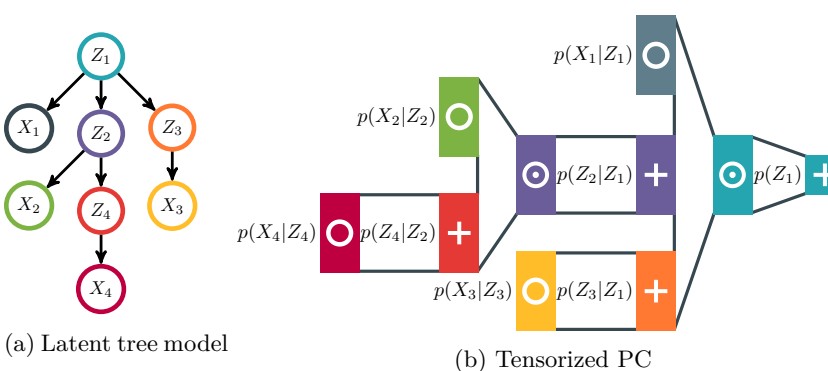

(a) Latent tree model

(b) Tensorized PC

Figure 18: **PGMs can be compiled as tensorized PCs.** We show how the PGM in (a), a latent tree model (LTM), with latent variables $Z_i$ and observable variables $X_i$, can be compiled in the tensorized PC over $\mathbf{X}$ in (b) using input, dense, and $\mathrm{CP}^\top$ layers parameterized by the conditional probability tables of the LTM, following the compilation algorithm proposed by Liu & Van den Broeck (2021b).

attached to the latent tree model the HCLT is compiled from (as we mentioned in Section 4.1). In other words, the difference between these tensorized circuit architectures and others such as EiNets or RAT-SPNs translates to simply a CP factorization of parameters if one fixes the same RG. In Appendix B we show that the same line of thought can be applied to the many tensorized PC architectures that have been developed so far. That is, Appendix B further details how the tensorized PC architectures reported in Table 1 can be understood and built within our pipeline, by specifying which RG and sum and product layer composition to use (Tucker, CP or $\mathrm{CP}^\top$), and whether to fold the computational graph or not. Next, we show how we can further exploit tensor factorizations as to build and compress *folded* tensorized circuit architectures.

## 5.2 Parameter Sharing by Tensor Factorizations

We now focus on the problem of sharing parameters across layers in a tensorized PC. We again exploit tensor factorizations for this task. Consider a tensorized PC built out of a RG as per our pipeline (Section 4.2). It is reasonable to assume that layers located at the same depth might store a similar structure in their parameter tensors. For example, two distinct layers having adjacent pixel patches of the same size as scope may apply a similar transformation to their respective inputs, as we can assume the distributions of the two pixel patches to be quite similar. If the RG is a perfectly balanced binary tree, folding the resulting circuit translates to folding layers located at the same depth, which are likely to share similar structure in parameter space. This motivates us to implement parameter sharing as a factorization across folded layers.

Specifically, we start by compressing a folded Tucker layer (Eq. (Tucker-folded)) and to retrieve a new layer that implements the aforementioned parameter sharing, we again decompose its parameter tensor via a CP factorization (Def. 10). This time, we will have to decompose $\boldsymbol{\mathcal{W}} \in \mathbb{R}^{F \times K \times K \times K}$, i.e., the 4-dimensional tensor obtained by reshaping of the parameter tensor of a folded Tucker layer $\boldsymbol{\ell}$, where $F$ indicates the folding dimension. By applying a rank-$R$ CP factorization such that $R \ll K$, we obtain that

$$\boldsymbol{\mathcal{W}} \approx \sum_{r=1}^{R} \mathbf{d}_{:r} \circ \mathbf{a}_{:r} \circ \mathbf{b}_{:r} \circ \mathbf{c}_{:r} \qquad \text{or in element-wise notation} \qquad w_{nijk} \approx \sum_{r=1}^{R} d_{nr} a_{ir} b_{jr} c_{kr} \qquad (17)$$

where $\mathbf{A}, \mathbf{B}, \mathbf{C} \in \mathbb{R}^{K \times R}$ and $\mathbf{D} \in \mathbb{R}^{F \times R}$. Note that $\mathbf{A}, \mathbf{B}, \mathbf{C}$ are independent of the fold dimension and are effectively shared among folds. By decomposing the parameter tensor in Eq. (Tucker-folded) as in Eq. (17) and by collapsing sum layers as done for the Tucker layer above, we can rewrite Eq. (Tucker-folded) as

$$\boldsymbol{\ell}\left(\bigcup_{n=1}^{F} \mathbf{Y}^{(n)}\right)_{n:} = \mathbf{d}_{n:} \odot \left(\mathbf{Q}^{(1)} \boldsymbol{\ell}_1 \left(\bigcup_{n=1}^{F} \mathbf{Z}_1^{(n)}\right)_{n:}\right) \odot \left(\mathbf{Q}^{(2)} \boldsymbol{\ell}_2 \left(\bigcup_{n=1}^{F} \mathbf{Z}_2^{(n)}\right)_{n:}\right) \qquad n \in [F] \quad (\text{CP}^{\text{S}}\text{-layer})$$

where $\mathbf{Q}^{(1)}, \mathbf{Q}^{(2)} \in \mathbb{R}^{R \times R}$ do not depend on the fold dimension, and $\mathbf{D} \in \mathbb{R}^{F \times R}$. However, we can go further in sharing parameters and drop the fold-dependent matrix $\mathbf{D}$ from Eq. (CP$^{\text{S}}$-layer), hence effectively fixing it to be a matrix of ones. The reason is that its contribution can be "absorbed" by the matrices associated to the following sum layers (i.e., similarly to the "collapse" of consecutive sum layers shown in Fig. 14). We

refer to this layer as $\mathrm{CP}^{\mathrm{XS}}$. Our experiments (Section 6) support this conjecture: as we experiment with both $\mathrm{CP}^{\mathrm{S}}$ and $\mathrm{CP}^{\mathrm{XS}}$, we find they achieve comparable performances for distribution estimation. These new layers are a nice addition to the circuit literature, and possible inspiration for further layer designs.

---

**Opportunity 7. Many new layers and circuit architectures**

We introduced Tucker, CP, $\mathrm{CP}^\top$, $\mathrm{CP}^{\mathrm{S}}$ and $\mathrm{CP}^{\mathrm{XS}}$ as possible composite layers for circuits and tensor factorizations, and we provide PyTorch snippets of them in Appendix F. However, one is not limited to such layers and can design new ones: as long as they are compositions of the building blocks outlined in Def. 7, they can be seamlessly plugged into Algorithm 1 to construct new tensor factorizations as tensorized circuits. Our experiments (Section 6) show that the choice of RG and layer significantly impacts the performance of the resulting architecture (may it be time and memory requirements or accuracy as distribution estimators), hence justifying further exploration of the design space of PC architectures. Lastly, we remark that one is not limited to pick the same composite layer for each node in a RG, according to Algorithm 1. From the point of view of tensor factorizations, this would result in a peculiar "Frankenstein" hierarchical tensor factorization that mixes different local factorizations, as shown in Fig. 6. From an ML perspective, determining which layer structure to select for each RG node can be cast as a *neural architecture search* task (Ren et al., 2021).

---

Table 2: **Distribution estimation results.** We report the test-set bpd of our best architectures, QT-CP-512 and QG-CP-512, and compare them against HCLT (Liu & Van den Broeck, 2021b), RAT-SPN (Peharz et al., 2020c), SparsePC (Dang et al., 2022a), IDF (Hoogeboom et al., 2019), BitSwap (Kingma et al., 2019), BBans (Townsend et al., 2019) and McBits (Ruan et al., 2021). SparsePC is a structure learning algorithm for PCs that iteratively finetunes both structure and parameters of a trained PC and can potentially be applied as a post-processing step to the PCs we are learning with our pipeline. HCLT results are taken from Gala et al. (2024a). Dataset CELEBA* is preprocessed using the lossless YCoCg transform.

| | QT-CP-512 | QG-CP-512 | HCLT (CL-CP) | RAT (RND-Tucker) | Sp-PC | IDF | BitS | BBans | McB |
|---|---|---|---|---|---|---|---|---|---|
| MNIST | 1.17 | 1.17 | 1.21 | 1.67 | 1.14 | 1.90 | 1.27 | 1.39 | 1.98 |
| F-MNIST | 3.38 | 3.32 | 3.34 | 4.29 | 3.27 | 3.47 | 3.28 | 3.66 | 3.72 |
| EMN-MN | 1.70 | 1.64 | 1.70 | 2.56 | 1.52 | 2.07 | 1.88 | 2.04 | 2.19 |
| EMN-LE | 1.70 | 1.62 | 1.75 | 2.73 | 1.58 | 1.95 | 1.84 | 2.26 | 3.12 |
| EMN-BA | 1.73 | 1.66 | 1.78 | 2.78 | 1.60 | 2.15 | 1.96 | 2.23 | 2.88 |
| EMN-BY | 1.54 | 1.47 | 1.73 | 2.72 | 1.54 | 1.98 | 1.87 | 2.23 | 3.14 |
| | QT-CP-256 | QG-CP-128 | | | | | | | |
| CELEBA | 5.33 | 5.33 | | | | | | | |
| CELEBA* | 5.24 | 5.20 | | | | | | | |

---

# 6 Empirical Evaluation: Which RG and Layers to use?

Destructuring modern PC architectures (as well as tensor factorizations) into our pipeline (Fig. 9) allows us to create new tensorized architectures by simply following a mix & match approach (Table 1). At the same time, it helps us understand what really matters between different model classes from the point of views of expressiveness, speed of inference and ease of optimization. We can now in fact easily disentangle key ingredients such as the role of RGs and the choice of composite layers in modern circuit architectures, and pinpoint which is responsible for a boost in performance. For example, HCLTs have been considered as one of the best performing circuit model architectures in recent benchmarks (Liu et al., 2022; 2023a), but until now it has not been clear why they were outperforming other architectures such as RAT-SPNs and EiNets. Within our framework, we can try to answer that question by answering more precise questions: *is it the effect of their RG that is learned from data (Section 4.1)?, the use of their composite sum-product*

*layer parameterization (Section 5.1)?* or *are other hyperparameter choices responsible*? (spoiler: it is going to be the use of CP layers).

Specifically, in this section we are interested in answering the following three research questions following a rigorous empirical investigation. **RQ1)** What are the computational resources needed (time and GPU memory) at test and training time for some of the many tensorized architectures we can now build? **RQ2)** What is the impact of the choice of RG and composite sum-product layer on the performance of tensorized circuits trained as distribution estimators? **RQ3)** Can we retain (most of) the performances of pre-trained tensorized PCs using Tucker layers if we factorize these into CP layers as illustrated in Fig. 14a → Fig. 14b? Note that we are not asking what is the impact of folding (Section 4.4), as we already know the answer: folding is essential for large-scale tensorized architectures. As such, throughout all experiments, we use folded tensorized circuits. *We emphasise that the aim of our experiments is not to reach state-of-the-art results in distribution estimation, but rather to understand the role of the ingredients of tensorized circuit architectures.* All experiments were run on a single NVIDIA RTX A6000 GPU with 48GB of memory. Our code is available at github.com/april-tools/uni-circ-le.

**A new circuit nomenclature.** We remark that HCLT, EiNets, RAT-SPNs, and all the other acronyms in Table 1 do not denote different model *classes* but just different *architectures*. They are instances of the same model class: smooth and (structured-)decomposable circuits. In the following, we will denote a tensorized architecture as [RG]-[sum-product layer], possibly followed by $K$, the number of units used for overparameterizing layers as in Algorithm 1. Under this nomenclature, RAT-SPNs and EiNets will both be encoded as RND-Tucker when they are both build with a random RG. When they are built with a Poon&Domingos RG, they will instead be referred to as PD-Tucker, meanwhile HCLTs will become CL-CP.

**Task & Datasets.** We will mainly evaluate our architectures by performing distribution estimation on image datasets. We use the MNIST-family, which includes 6 datasets of gray-scale $28 \times 28$ images—MNIST (LeCun et al., 2010), FASHIONMNIST (Xiao et al., 2017), and EMNIST with its 4 splits (Cohen et al., 2017)—and the CelebA dataset down-scaled to $64 \times 64$ (Liu et al., 2015), which we explore in two versions: one with RGB pixels and the other with pixels preprocessed by the lossless YCoCg color-coding (Malvar & Sullivan, 2003), as recent results suggested that such a transform can greatly lower bpds.[8] Furthermore, we perform experiments on tabular data with continuous variables. In particular, we will evaluate different tensorized layers by performing density estimation on 5 UCI datasets, as they are typically used to evaluate normalizing flows (Papamakarios et al., 2017). We report the statistics of the UCI dataset in Table E.5.

**Parameter optimization.** We train circuits to estimate the probability distribution that is assumed to have generated the images, considering each pixel as a random variable. As such, the input units in the circuit represent Categorical distributions having 256 values. For RGB images, we associate three Categorical distribution units per pixel (one per color channel). Instead, for the 5 UCI datasets (Table E.5), we use input units representing univariate Gaussian distributions, and we learn both the means and the standard deviations. We perform maximum likelihood by stochastic gradient ascent, i.e., want to maximize the following objective

$$\mathcal{L}(\mathcal{B}, c) = \sum_{\mathbf{x} \in \mathcal{B}} \log(c(\mathbf{x})) - \log(Z), \tag{18}$$

where $Z = \sum_{\mathbf{x}} c(\mathbf{x})$ is the partition function of the PC $c$,[9] and $\mathcal{B}$ a batch of training data. After some preliminary experiments, we found that optimizing PCs with Adam (Kingma & Ba, 2015) using a learning rate of $10^{-2}$ delivered, on average, the best performing models for the datasets we considered. We also settled to reparameterize the circuit sum parameters via clamping and setting $\epsilon = 10^{-19}$ (Eq. (9)) after each optimization step as to keep them non-negative, as it was giving the best learning dynamics among all possible reparameterizations (Section 3.2). In the following, we will summarize our findings when answering RQ1-3, while distilling recommendations for practitioners on how to build and learn circuits.

---

[8]We take this evidence from Liu et al. (2023a;b), which use however a lossy variant of the YCoCg transform that unfortunately artificially inflates likelihoods. As such, their bpds for PCs are not directly comparable with ours, nor with the other deep generative models in their tables. We confirmed this issue in their evaluation protocol via personal communication.

[9]After training, one can efficiently "embed" the normalization constant in the parameters of a PC, effectively renormalizing them (and thus yielding a partition function $Z$ equal to 1), as detailed by Peharz et al. (2015).

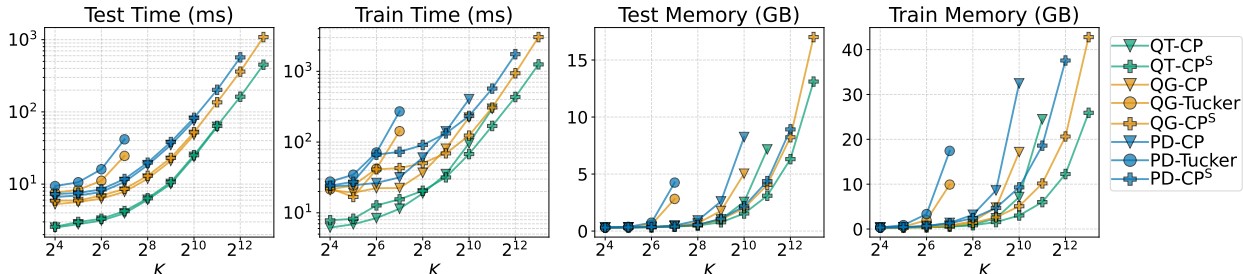

Figure 19: **Benchmarking the role of RGs and composite layers in tensorized circuits.** We report the average time (ms) and GPU memory usage (GiBs) to process a batch of 128 samples from MNIST for different tensorized architectures—listed in the legend on the right—at different values of $K$ (x-axis). The stats are reported for both test and training scenarios, where for training one has to expect additional overhead from performing gradient ascent.

**RQ1) Benchmarking time & space for different tensorized architectures.** For these experiments, we consider the following RGs: PD, as commonly used in architectures such as RAT-SPNs and EiNets, and the two novel light-weight and data-agnostic RGs we introduced in Section 4.1, QTs [10] and QGs . We do not consider RND as it is usually just a balanced binary tree (Peharz et al., 2020c), and as such would yield the same time and memory performance of a QT. For the same reason we do not consider CL as they are tree RGs that end up being quasi-balanced after being rooted.[11] For layers, we consider Tucker (Eq. (Tucker-layer)), CP (Eq. (CP-layer)), CP$^S$ (Eq. (CP$^S$-layer)) and CP$^{XS}$ (Section 5.2).

In Fig. 19, we report the average time and peak GPU memory required to process a data batch from MNIST for several tensorized PC architectures built by mixing & matching different RGs and type of sum-product layers mentioned above, when possible on our GPU budget. For each architecture, we vary the model size by varying $K$, the number of units for each layer, in $\{2^i\}_{i=4}^{14}$. We observe that the QT and QG region graphs deliver more scalable architectures than those based on the commonly used PD which is consistently slower and uses more memory. At the same time, one can see that CP and CP$^S$ layers scale more gracefully: CP can accommodate $K = 2^{10}$ with QT as a RG and CP$^S$ even larger values of $K$, up to $2^{13}$ with QG as well. Doing this is instead computationally impractical for Tucker layers on our GPUs, which allow only for $K = 128$ at most. We underline that this is expected as models using Tucker layers have more parameters than those using CP layers for the same model size $K$. This also explain why the architecture QT-Tucker is missing: QTs iteratively split images in 4 parts (Algorithm D.2) and therefore appling Tucker layers would require $\mathcal{O}(K^4)$ parameters for such architectures, which is unfeasible even for $K = 16$ on our GPUs.

We emphasise that non-folded versions of these architectures, e.g. RAT-SPNs (Peharz et al., 2020c), can be orders of magnitude slower, hindering both learning and deployment in practice. In Fig. E.1, we show the results of the same benchmark reported in Fig. 19 but for the CELEBA dataset, which is more challenging because it is equivalent to perform distribution estimation on a much higher dimensional space ($12,288 = 64 \times 64 \times 3$ instead of $784 = 28 \times 28 \times 1$).[12] From this additional experiment, we conclude that even in higher dimensions the scaling trend of RGs and layers is the same. Finally, in Fig. E.3, we zoom on a comparison between CP$^S$ and CP$^{XS}$. There, we show that for the same choice of RG and $K$, CP$^S$ and CP$^{XS}$ layers require the same time/space resources as expected, with CP$^{XS}$ only being slightly faster at training-time.

**Takeaway 1.**

> QT and QG should be preferred to PD as RGs if we want to scale circuits, with the former being more scalable than the latter. Layer-wise, CP layers scale, as expected, to larger values of $K$ than Tucker layers and for even larger layers parameter sharing (CP$^S$, CP$^{XS}$) is recommended.

---

[10]Throughout our experiments, we will refer to QT-4 simply as QT.
[11]The root is chosen to be the barycenter of the graph to increase parallelism (Dang et al., 2021; 2022c).
[12]Note that for our RQ1, all image datasets with the same resolutions would yield the very same results.

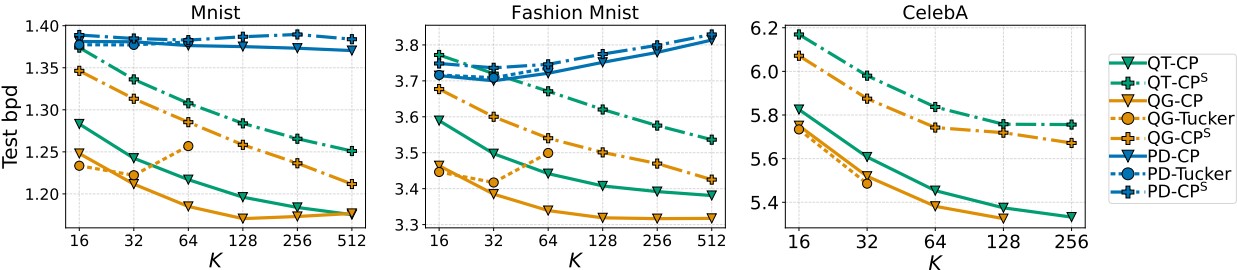

Figure 20: **Overparameterizing tensorized architectures delivers better performing models when using QTs and QGs, but not when using PDs.** We report the test-set bpd (y-axis) at different values of $K$ (x-axis) for MNIST (left), FASHIONMNIST (middle) and CELEBA (right) averaged over 5 runs for different tensorized architectures, which we report in the legend on the right. We keep the mixing layers in QG- and PD-based models fixed and normalized. We use a batch size of 256.

**RQ2) Accuracy as distribution estimators.** We now test our tensorized PCs as distribution estimators and we consider our mixed&matched architecture from RQ1. For each architecture, we vary the model size by varying $K$, the number of units for each layer, in $\{16, 32, 64, 128, 256, 512\}$ for the MNIST family and up to 256 for CELEBA. To assess the effect of learning RGs from data, we compare against HCLTs (CL-CP$^\top$ in our nomenclature) as reported by Dang et al. (2022a). We use a batch size of 256, and train for at most 200 epochs stopping training if the validation log-likelihood does not improve after 5 epochs. We use the average test-set bits-per-dimension (bpd) as the evaluation criterion, i.e. $\mathsf{bpd}(\mathcal{D}, c) = -\mathcal{L}(\mathcal{D}, c)/(d \cdot \log 2)$, where $d$ is the number of features in dataset $\mathcal{D}$ and $\mathcal{L}$ is defined as in Eq. (18).

In Fig. 20 we report the average test-set bpd on MNIST, FASHIONMNIST and CELEBA. An immediate visible pattern emerges when comparing the architectures w.r.t. the choice of RG: Both QT- and QG-based architectures outperform those based on PD, and also manage to scale to larger datasets like CELEBA. On average, the best performing architectures are those built out of QGs. This is expected as such RGs, different from QTs, allow different partitionings for a same region (and therefore require the usage of mixing layers as discussed in Eq. (Mixing-layer)). The PD region graphs, despite being DAG-shaped as QGs, deliver underperforming tensorized architectures, suggesting that bigger models, while being more expressive, are harder to train, a behavior also noted by Liu et al. (2023a). This is particularly evident looking at the trend of PD-based architectures on FASHIONMNIST.

In Table 2, we compare our best performing architectures, with other state-of-the-art models even outside the circuit literature. Our architectures deliver close-to state-of-the-art results, and outperforms some VAE- and flow-based models. When compared with RGs learned from data, as it is the case for HCLT, we note that our simpler, data-agnostic alternatives, QTs and QGs, perform equally well or better. Using them instead of HCLTs avoids the quadratic cost to learn the corresponding Chow-Liu tree (Dang et al., 2021).

As for the choice of type of sum-product layer, Tucker and CP layers deliver very similar performance on PD. We conjecture that this is due to PD being harder to train in general, as for other RGs the trend changes. In fact, with QT and QG, we observe that Tucker delivers the best bpds for the smallest values for $K$. Scaling it to larger $K$s is impractical however. CP and variants not only scale better (see RQ1 and Fig. 19), but are able to deliver the best bpds for larger $K$. As expected, CP consistently outperforms CP$^S$ having more learnable parameters. However, if one has to privilege time over accuracy, CP$^S$ can be a useful alternative. Finally, we report results for CP$^{XS}$ layers and learnable mixing layers in Appendix E, along with the results showed in Fig. 20 in tabular form. We confirm that CP$^S$ and CP$^{XS}$ layers are equivalently accurate and that one does *not* have to learn mixing layer parameters in tensorized PCs with DAG-shaped RGs (Section 4.3). All these conclusions carry over also to a larger image dataset such as CELEBA, with or without the lossless YCoCg color-coding.

**Density estimation on tabular datasets.** Finally, we perform density estimation experiments on UCI datasets (Table E.5), and compare the results achieved by tensorized PCs constructed by our pipeline by

Table 3: **Tucker layers are harder to scale than CP layers on high-dimensional UCI datasets.** We show the best average test log-likelihoods achieved by normalizing flow models (top) and tensorized PCs that can be instantiated from our pipeline (bottom). See main text for their description. Tensorized PCs obtained by parameterizing random binary tree RGs (Section 4.1) with CP (RND-CP) perform better on higher-dimensional datasets Hepmass and MiniBooNE than those with Tucker layers (RND-Tucker), while the latter have an advantage on lower-dimensional datasets such as Power and Gas. For RND-CP and RND-Tucker, we report the layer width ($K$) of the best performing model as a subscript of the log-likelihoods. Fig. E.5 shows training and test log-likelihoods achieved by varying the layer width $K$. Details in Appendix E.1.

|  | Power | Gas | Hepmass | M.BooNE | BSDS300 |
|---|---|---|---|---|---|
| MADE (Germain et al., 2015) | -3.08 | 3.56 | -20.98 | -15.59 | 148.85 |
| RealNVP (Dinh et al., 2017) | 0.17 | 8.33 | -18.71 | -13.84 | 153.28 |
| MAF (Papamakarios et al., 2017) | 0.24 | 10.08 | -17.73 | -12.24 | 154.93 |
| NSF (Durkan et al., 2019) | 0.66 | 13.09 | -14.01 | -9.22 | 157.31 |
| Gaussian | -7.74 | -3.58 | -27.93 | -37.24 | 96.67 |
| EiNet-LRS (Sidheekh et al., 2023, RND-Tucker) | 0.36 | 4.79 | -22.46 | -34.21 | — |
| TTDE Novikov et al. (2021, LT-CP$^\top$) | 0.46 | 8.93 | -21.34 | -28.77 | 143.30 |
| RND-CP | $0.28_{256}$ | $5.01_{256}$ | $-22.52_{64}$ | $-30.69_{128}$ | $120.82_{64}$ |
| RND-Tucker | $0.52_{64}$ | $8.41_{256}$ | $-23.47_{32}$ | $-31.30_{8}$ | $119.09_{64}$ |

parameterizing a RND RG (Section 4.1) with either CP or Tucker layers. To give context to our results, we show the average test log-likelihoods achieved by normalizing flow models (Papamakarios et al., 2021) that are often evaluated on UCI datasets: MADE (Germain et al., 2015), RealNVP (Dinh et al., 2017), MAF (Papamakarios et al., 2017) and NSF (Durkan et al., 2019). As additional baselines, we show results of other PCs supporting tractable marginalization: a single multivariate Gaussian, Einsum networks (Peharz et al., 2020a) with input layers encoding flows (EiNet-LRS) (Sidheekh et al., 2023), and TTDE (Novikov et al., 2021). We emphasize that both EiNet-LRS and TTDE can be built using our pipeline and characterized with our nomenclature, the former as RND RGs parameterized by Tucker layers, and the latter as LT RGs parameterized by CP$^\top$ layers (Table 1). Table 3 shows that CP layers to deliver better performances than Tucker layers on high-dimensional UCI datasets and therefore in the case of deeper tensorized PCs. On the other hand, Tucker layers outperform CP layers on the lower-dimensional UCI datasets. We believe this is due the parameters of Tucker layers being more difficult to train and scale, similarly to our observation for MNIST and FASHIONMNIST in the case of QG RGs in Fig. 20. We further detail in Appendix E.1 the experimental setting, and show in Fig. E.5 the results achieved by varying the layer width $K$.

> **Takeaway 2.**
>
> In the case of image datasets, our recommendation for a go-to architecture is QG-CP-$K$, with the largest possible $K$ one can squeeze in their GPU memory. If computational resources are not enough, one can trade-off accuracy with speed and use QT-{CP,CP$^S$}-$K$. As a general trend, the simpler the architecture the easier training and scaling are. This is also suggested by our results on UCI datasets, where the simpler CP layers can perform better for high-dimensional datasets than Tucker.

**RQ3) Compressing circuits with Tucker layers.** For our last research question, we consider the problem when a trained circuit with Tucker layers is given, and we want to compress it into a smaller one using CP layers by using our compression pipeline as illustrated in Fig. 14a and Fig. 14b. With this in mind, we investigate the change in performance, if any, w.r.t. the number of tunable parameters. Specifically, for each folded Tucker layer (Eq. (Tucker-folded)) in the given circuit, parameterized with a tensor $\mathcal{W}$ of shape $F \times K \times K \times K$ we compress each tensor slice $\mathcal{W}_{f:::}$ by performing non-negative (NN) CP factorization via alternating least squares (Shashua & Hazan, 2005). This optimization eventually delivers a tensor $\mathcal{W}'$ of shape $F \times 3 \times R \times K$ for a $R$-ranked factorization.

Figure 21: **Compressing Tucker layers into CP layers** (Fig. 14a → Fig. 14b) **can yield smaller and accurate models** as seen when we performing non-negative (NN) CP factorization via alternating least squares (Shashua & Hazan, 2005). In each plot, we report the bpd of a pre-trained Tucker-layered PC (dashed blue line), whose RG, size $K$ and dataset on which it was trained on are detailed at the top. We report the bpd of several $R$-ranked NN-CP factorizations of such PCs (red curves), which we then use as initialization for further fine-tuning (green curves). Finally, we report the bpd of Tucker-compressed PCs (Fig. 14b) trained from a random initialization of their parameters (yellow curves).

We sketch the results of our investigation in Fig. 21. As expected, taking a pre-trained Tucker-layered PC (blue dashed line) and compressing its parameters via NN-CP factorization leads to a similar-performing model as the rank $R$ of the approximation increases, as shown by the bpd trend of the red curves in Fig. 21. Interestingly, we observe a key difference between the two region graphs utilized. For tensorized PCs based on PD region graphs, even a rank 1 approximation (i.e. $R = 1$) leads to a relatively small bpd loss, while this is not the case for PCs built out of QGs. We conjecture that PD-based PCs have parameter tensors that are of much lower rank than QG-based PCs, and that very deep PCs learn low-rank parameter matrices.

Next, we investigate whether we can use these compressed models as an effective initialization scheme for smaller circuits, which we further train (fine-tune) to maximize the training data likelihood (Eq. (18)). Again, we see a different trend when comparing w.r.t. the region graph used, as shown by the bpds encoded as green curves in Fig. 21. Specifically, for PD-based PCs such fine-tuning leads to a quick overfitting already in the first optimization steps, leading to much higher bpds on test data. In contrast, fine-tuning QG-based PCs leads to models that consistently match or even outperform the original Tucker-based PCs (blue dashed line), i.e., we observe green curves consistently being below red curves and crossing the dashed blue lines.

As an additional baseline, we use the architecture of these compressed models (Fig. 14b) but train them from scratch: starting from a random initialization of its parameters. Fig. 21 illustrates that the NN-CP initialization can be better than a random one as it leads to better performing models when using QG RGs (yellow curves over green curves). This trend flips when using PD region graphs (yellow curves below green curves), again signaling that much information for these models could be stored in the RG rather than in the parameters of the circuit. This, in turn, suggests that while new hierarchical factorizations with highly intricate RGs but very low-rank inner tensors are possible, they might be harder to learn effectively.

> **Takeaway 3.**
>
> Deep circuits encode distributions in highly-structured factorizations whose parameters can be effectively further compressed, e.g., by NN-CP factorizations. This yields a simple and effective procedure to distill a smaller tractable model from a larger one: compress each layer of the latter, then fine-tune the former by maximum-likelihood estimation.

# 7    Additional Related Work

In the previous sections, we surveyed and bridged the literature of circuit representations and tensor factorizations, and as such we have already reviewed several related works from both communities. Now, we discuss works that partially tried to establish this connection in the past, by trying to connect to probabilistic graphical models.

**Tensor networks and PGMs.** TNs (Orús, 2013) are widely used to model many-body systems in physics and quantum mechanics (Schollwoeck, 2010), and have been used to simulate quantum computations on classical hardware (Markov & Shi, 2008). They have been applied more recently for machine learning applications (Stoudenmire & Schwab, 2016; Han et al., 2018; Efthymiou et al., 2019; Bonnevie & Schmidt, 2021). As they essentially an alternative formalism for probabilistic graphical models over discrete variables (Koller & Friedman, 2009), people have started drawing connections between the two formalisms. For example, Bonnevie & Schmidt (2021) connects non-negative MPS/TTs to PGMs and offers routines for probabilistic reasoning. Similarly, Glasser et al. (2020) explores the same connection, but instead of drawing TNs as PGMs, they draw them as factor graphs (Kschischang et al., 2001).

Interestingly, these works are not aware of the latent variable interpretation of non-negative factorizations (Section 3.1) as they miss the connection through circuits. For the same reason, they are limited to autoregressive sampling (Opportunity 3). To the best of our knowledge, this latent-variable perspective has been (re)discovered only very recently in this concurrent work by Ghalamkari et al. (2024) who proposes the classical expectation-maximization (EM) algorithm to learn them. EM is a consolidated way to learn the parameters of circuits (Peharz et al., 2016; 2020a) by maximum likelihood.

Instead, by representing non-negative tensor factorizations as monotonic PCs, we effortlessly unlock the developed theory and algorithms required to perform complex probabilistic inference, with possible applications in lossless compression (Liu et al., 2022), neuro-symbolic AI with correctness guarantees (Ahmed et al., 2022) and constrained text generation (Zhang et al., 2023). Moreover, results about the succinctness or expressive efficiency of these factorizations (Glasser et al., 2019) have been used recently to prove circuit complexity lowerbounds (Loconte et al., 2024; 2025). Finally, Loconte & Vergari (2025) took inspiration from canonical forms in tensor networks (Schollwoeck, 2010; Bonnevie & Schmidt, 2021) to parameterize already-normalized squared circuits generalizing squared MPS/TTs and TTNs (Section 2.4), and to devise a more efficient marginalization algorithm within the circuit language.

**Probabilistic circuits and PGMs.** The modern formulation of PCs has been introduced for the first time in (Vergari et al., 2019b) as a unifying framework for several existing tractable probabilistic models (TPMs) including arithmetic circuits, (Darwiche, 2001), probabilistic decision graphs (Jaeger, 2004), and-or graphs (Marinescu & Dechter, 2009), cutset networks (Rahman et al., 2014), sum-product networks (Poon & Domingos, 2011) and more (Choi et al., 2020). The aim of PCs has been to abstract away from the different syntaxes and model formalisms of the above TPMs and focus on structural properties that enable tractable inference in each. Non-negative tensor factorizations and tensor networks have been underlooked in this effort so far. Several ways to compile discrete PGMs into PCs (or one of the above formalisms) have been devised in the past (Oztok & Darwiche, 2017; Shen et al., 2016; Choi et al., 2013). These compilation techniques yield sparse deterministic circuits, and only recently PCs have started to be represented first in code (Peharz et al., 2020c;a; Liu & Van den Broeck, 2021b) and then formally (Loconte et al., 2024) as tensorized architectures. Perhaps this lack of tensorized compilation targets has hidden the connection between PCs and matrix and tensor factorizations. The closest connection we are aware of can be found in Jaini et al. (2018b): they bridge sum-product networks to hierarchical mixture models and HMMs and hint at a connection with tensorial mixture models (Sharir et al., 2017) a variant of hierarchical Tucker (Def. 5).

**Matrix factorizations, circuit complexity, and tensor networks expressiveness.** Finding lower bounds to the rank of matrix factorizations can be used as a proxy to prove lower bounds to the size of circuits satisfying particular structural properties (de Colnet & Mengel, 2021). Proving an *exponential* (w.r.t. the number of variables) size lower bound for a class of circuits shows a limitation on which functions they can compute in polynomial time and number of parameters, thus allowing us to precisely separate circuit classes in terms of their expressiveness (Valiant, 1979; Martens & Medabalimi, 2014). Recently, lower bounding the non-negative rank (Gillis, 2020) and the square root rank (Fawzi et al., 2014; Lee & Wei, 2014) of matrices has been used to draw an expressiveness hierarchy of classes of PCs with negative real and complex-valued parameters for distribution estimation (Loconte et al., 2024; 2025). Since circuits generalize many tensor network factorizations (see Section 2.4), showing size lower bounds for a class of circuits can be used to show size lower bounds for tensor networks *regardless of their structure*, e.g. as shown by Loconte et al. (2025) in generalizing a known rank lower bound for Born machines obtained by squaring a MPS/TT with real-valued tensors (Glasser et al., 2019).

# 8 Conclusion

In this paper, we laid the foundations to connect two communities in ML that developed independently but are sharing many research directions: circuits and tensor factorizations. Despite their apparently different syntax, the way they are usually presented, and the tasks in which they are commonly employed, these two formalisms significantly overlap in semantics and potential applications. We create this bridge between communities by first establishing a formal reduction of popular tensor factorizations to circuits in Section 2.

We hope this can propel research on how to design more and more scalable low-rank parameterization for probabilistic inference. To this end, we highlighted a number of possible future avenues for the matrix and tensor factorization communities that leverage the connection with circuits we established: designing hierarchical factorizations with non-tree structures (Opportunity 1); using the property-driven calculus that circuits offer to automatically derive tractable algorithms in a compositional way (Opportunity 2); treat non-negative (hierarchical) factorizations as deep latent variable models (Opportunity 3); devise factorizations over non-discrete and non-linear input spaces (Opportunity 4); embed logical constraints to realize neuro-symbolic systems that can reason with symbolic knowledge (Opportunity 5); devising alternative ways to compactly encode distributions, going beyond probability masses or densities (Opportunity 6); as well as devising flexible factorizations by changing only the structure of (some) layers in a circuit representation (Opportunity 7).

From the point of view of the circuit community, we leveraged this connection to systematize and demystify the construction of modern tensorized and overparameterized circuits (Section 4). We proposed a single pipeline that generalizes existing (tensor factorization and circuit) architectures and introduced a new nomenclature, based on the steps of our pipeline, to understand old but also new architectures that can be created by mixing & matching these steps (see Table 1). Our empirical analysis of popular ways to combine these ingredients highlights how lower-rank structures can be easier to learn and useful to compress higher-rank layers (Section 6). Finally, we distilled our findings in clear-cut recommendations (Takeaways 1 to 3) for practitioners that want to learn and scale circuits on high-dimensional data, and we hope this can foster future rigorous analysis.

**Broader Impact Statement**

This work is fundamental research in probabilistic modeling and reasoning and as such the algorithms and architectures discussed here can impact many possible downstream applications, in ways that go beyond our control. For example, circuits, or tensor factorizations, could be used in computer vision classifiers to amplify the bias already encoded in non-curated datasets or be used in safety-critical applications without eliciting valid safety requirements. Since it is hard to foresee all possible future misuses, we urge practitioners to pay attention to concrete problematic uses of our methodologies: use the time that tractable models save while performing inference to reflect on the direct impact your application can have.

**Author Contributions**

AV and AM devised the original idea of tensorizing and compressing circuits in a unified pipeline. This has yielded a preliminary workshop paper whose content can be traced to Sections 4 and 5. AV and GV wrote the workshop paper and AM designed and run its experiments, which constituted the basis for the experiments in Section 6. LL traced the theoretical connections with tensor factorizations and tensor networks, and lead the writing of all sections in the journal paper with AV. AV and LL are responsible for designing and drawing all circuit pictures in TikZ, with the exception of Fig. 18 done by GG who also provided feedback for several plots. GG simplified the algorithmic pipeline, wrote the sampling algorithm in Appendix C, produced Appendix F, and designed new experiments. GG and AM re-run and extended the workshop experiments to adapt them to the journal version, and GG wrote Section 6 with AV. EQ, RP and CdC critically read the manuscript. AV supervised all the phases of the project and provided feedback.

## Acknowledgments

We would like to acknowledge Nicolas Gillis and Grigorios Chrysos for providing feedback on an early draft of the paper. AV wants to thank Iain Murray for forwarding an email from Raul Garcia-Patron Sanchez inquiring about tensor networks and ML, and the latter for many following discussions about tensor networks. The Eindhoven University of Technology authors thank the support from (i) the Eindhoven Artificial Intelligence Systems Institute, (ii) the Department of Mathematics and Computer Science of TU Eindhoven, and (iii) the EU European Defence Fund Project KOIOS (EDF-2021-DIGIT-R-FL-KOIOS). This project has received funding from the European Union's EIC Pathfinder Challenges 2022 programme under grant agreement No 101115317 (NEO). Views and opinions expressed are however those of the author(s) only and do not necessarily reflect those of the European Union or European Innovation Council. Neither the European Union nor the European Innovation Council can be held responsible for them. AV was supported by the "UNREAL: a Unified Reasoning Layer for Trustworthy ML" project (EP/Y023838/1) selected by the ERC and funded by UKRI EPSRC.

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

# A  Proofs

## A.1  Tucker as a Circuit

**Proposition 1** (Tucker as a circuit)**.** Let $\mathcal{T} \in \mathbb{R}^{I_1 \times \cdots \times I_d}$ be a tensor being decomposed via a multilinear rank-$(R_1, \ldots, R_d)$ Tucker factorization, as in Eq. (2). Then, there exists a circuit $c$ over variables $\mathbf{X} = \{X_j\}_{j=1}^d$ with $\mathsf{dom}(X_j) = [I_j]$, $j \in [d]$ computing the same factorization. Moreover, we have that $|c| \in \mathcal{O}(d \prod_{j=1}^d R_j)$.

*Proof.* We prove it constructively by giving the structure and parameters of $c$. That is, we build a circuit $c$ over variables $\mathbf{X}$ computing

$$c(\mathbf{X}) = c(x_1, \ldots, x_d) = \sum_{r_1=1}^{R_1} \cdots \sum_{r_d=1}^{R_d} w_{r_1 \cdots r_d} c_{1,r_1}(x_1) \cdots c_{d,r_d}(x_d). \tag{19}$$

Note that in Eq. (19) each product $c_{1,r_1}^{\mathsf{in}}(x_1) \cdots c_{d,r_d}^{\mathsf{in}}(x_d)$ can be computed by a product unit $c_{r_1 \cdots r_d}^{\mathsf{prod}}$ over variables $\mathbf{X}$. Moreover, we encode each $c_{j,r_j}^{\mathsf{in}}$ as an input unit, for all $j \in [d]$ and $r_j \in [R_j]$. In addition, the collection of sums $\sum_{r_1=1}^{R_1} \cdots \sum_{r_d=1}^{R_d}$ that are weighted by the $w_{r_1 \cdots r_d}$ can be computed by a single sum unit having $\prod_{j=1}^d R_j$ inputs, i.e., the products $c_{r_1 \cdots r_d}^{\mathsf{prod}}$ with $r_j \in [R_j]$ for all $j$. Since each product units has $d$ inputs, we have that the overall circuit size is $|c| \in \mathcal{O}(d \prod_{j=1}^d R_j)$. Finally, we take $w_{r_1 \cdots r_d}$ as the entries of the core tensor $\mathcal{W}$, and make each input unit $c_{j,r_j}^{\mathsf{in}}$ compute $c_{j,r_j}^{\mathsf{in}}(x_j) = v_{x_j, r_j}^{(j)}$ for the factor matrices $\{\mathbf{V}^{(j)}\}_{j=1}^d$ in the Tucker factorization. That is, $c(\mathbf{X})$ computes the same Tucker factorization given by hypothesis. $\quad\square$

## A.2  Hierarchical Tucker as Deep a Circuit

**Proposition 2** (Hierarchical Tucker as a deep circuit)**.** Let $\mathcal{T} \in \mathbb{R}^{I_1 \times \cdots \times I_d}$ be a tensor being decomposed using hierarchical Tucker factorization according to a RG $\mathcal{R}$ (Def. 5). Then, there exists a circuit $c$ over variables $\mathbf{X} = \{X_j\}_{j=1}^d$ with $\mathsf{dom}(X_j) = [I_j]$, computing the same factorization. Furthermore, given $\{\mathbf{Y}^{(i)}\}_{i=1}^m \subset 2^{\mathbf{X}}$ the set of all non-leaf region nodes $\mathbf{Y}^{(i)} \subseteq \mathbf{X}$ being factorized into $(\mathbf{Z}_1^{(i)}, \mathbf{Z}_2^{(i)})$ in $\mathcal{R}$, with corresponding Tucker factorization multilinear rank $(R_{\mathbf{Y}^{(i)}}, R_{\mathbf{Z}_1^{(i)}}, R_{\mathbf{Z}_2^{(i)}})$, we have that $|c| \in \mathcal{O}\left(\sum_{i=1}^m R_{\mathbf{Y}^{(i)}} R_{\mathbf{Z}_1^{(i)}} R_{\mathbf{Z}_2^{(i)}}\right)$.

*Proof.* Similarly to our proof for Proposition 1, we prove it constructively by giving the structure and parameters of $c$. That is, we rewrite the recursive rules used to define a hierarchical Tucker factorization showed in Def. 5 in terms of equivalent circuit computational units. For every leaf region $\mathbf{Z} = \{X_j\}$ in $\mathcal{R}$, we introduce the input units $c_{j,r_j}^{\mathsf{in}}$, $r_j \in [R_{\mathbf{Z}}]$, each computing $c_{j,r_j}(x_j) = v_{x_j r_j}^{(j)}$ for the factor matrix $\mathbf{V}^{(j)}$ of the hierarchical Tucker factorization given by hypothesis. Next, we recursively introduce sum and product units by following the hierarchical variables factorization defined in $\mathcal{R}$. That is, for every non-leaf region node $\mathbf{Y} \subseteq \mathbf{X}$ being partitioned into $(\mathbf{Z}_1, \mathbf{Z}_2)$ in $\mathcal{R}$, we introduce sum and product units that encode a Tucker factorization related to the region node $\mathbf{Y}$. More formally, given $(R_{\mathbf{Y}}, R_{\mathbf{Z}_1}, R_{\mathbf{Z}_2})$ the multilinear rank associated to the region node $\mathbf{Y}$, we introduce the sum units $c_{\mathbf{Y},s}^{\mathsf{sum}}$, with $s \in [R_{\mathbf{Y}}]$. Moreover, we introduce the product units $c_{\mathbf{Y},r_1,r_2}^{\mathsf{prod}}$, with $r_1 \in [R_{\mathbf{Z}_1}]$ and $r_2 \in [R_{\mathbf{Z}_2}]$. Each sum unit $c_{\mathbf{Y},s}^{\mathsf{sum}}$ has the product units $\{c_{\mathbf{Y},r_1,r_2}^{\mathsf{prod}}\}_{r_1=1,r_2=1}^{R_{\mathbf{Z}_1}, R_{\mathbf{Z}_2}}$ as inputs, and is parameterized by weights $\{w_{s,r_1,r_2}\}_{r_1=1,r_2=1}^{R_{\mathbf{Z}_1}, R_{\mathbf{Z}_2}}$. Furthermore, we recursively define the inputs to each product unit $c_{\mathbf{Y},r_1,r_2}^{\mathsf{prod}}$ to be the pair of sum units $c_{\mathbf{Z}_1,r_1}^{\mathsf{sum}}$ and $c_{\mathbf{Z}_2,r_2}^{\mathsf{sum}}$, for all $r_1 \in [R_{\mathbf{Z}_1}]$ and $r_2 \in [R_{\mathbf{Z}_2}]$. By setting the parameters of each sum unit $\theta_{s,r_1,r_2}$ to be the entries of the core tensor $\mathcal{W}^{(\mathbf{Y})}$ (see Eq. (6)), we recover that the constructed composition of sum and product units encodes the Tucker factorization associated to $\mathbf{Y}$. Finally, in the case of the root region $\mathbf{Y} = \mathbf{X}$ in $\mathcal{R}$, we have that $R_{\mathbf{Y}} = 1$ by hypothesis, and therefore the output of the circuit is given by the sum unit $c_{\mathbf{X},1}^{\mathsf{sum}}$. Since the circuit $c$ built in this way consists of a composition of Tucker factorizations represented as circuits (Proposition 1), the circuit size is $|c| \in \mathcal{O}(\sum_{i=1}^m R_{\mathbf{Y}^{(i)}} R_{\mathbf{Z}_1^{(i)}} R_{\mathbf{Z}_2^{(i)}})$, with $\{\mathbf{Y}^{(i)}\}_{i=1}^m$ being the set of non-leaf regions in $\mathcal{R}$. $\quad\square$

# B   Many Tensorized PC Architectures can be Obtained through our Pipeline

We will consider one tensorized PC architecture at a time, and show how its construction can be understood in terms of simple design choices presented in our pipeline: (1) the region graph to parameterize (Section 4.1), (2) the sum and product layers chosen (Sections 4.2 and 4.3 and Section 5), and (3) whether the architecture is folded or not (Section 4.4).

**Poon & Domingos circuits** (Poon & Domingos, 2011) for image data follow the homonomous region graph structure. While the circuit is *not* tensorized, i.e., the computational units defined over the same variable scope are not replicated and tensorized into layers, we can still see them as a tensorized circuit where the width of each layer is 1. Furthermore, no folding is performed to the best of our knowledge.

**Randomized-and-tensorized circuits (RAT-SPN)** (Peharz et al., 2020c) are obtained by parameterizing a randomly-constructed binary tree region graph (named RND in this paper). In particular, in this architecture Kronecker product layers and sum layers are alternated, thus being equivalent to circuits with Tucker layers (Eq. (Tucker-layer)) in our pipeline. In the original implementation of RAT-SPNs (Peharz et al., 2019), layers are no folded.

**Einsum networks (EiNets)** (Peharz et al., 2020a) include a folded version of RAT-SPNs, as well as tensorized *and* folded circuits obtained by overparameterizing the PD region graph. See Peharz et al. (2020b) and Braun (2021) for known available implementations.

**Hidden Chow-Liu Tree (HCLT) circuits** (Liu & Van den Broeck, 2021b) are tensorized circuits obtained by compiling a tree-shaped graphical model that is learned with the Chow-Liu algorithm (Chow & Liu, 1968a). Therefore, it can be obtained in our pipeline by parameterizing the CL region graph with $\mathrm{CP}^\top$ layers whose parameter matrices encode conditional probability tables. HCLTs have been originally implemented within the Juice.jl Julia library (Liu & Van den Broeck, 2021a), which also includes a parallelization scheme using custom CUDA kernels that fuse sum and products operations.

**Non-negative matrix-product states ($\mathrm{MPS}_{\mathbb{R}\geqslant 0}$)** have been shown to be equivalent to hidden-markov-models (HMMs) (Rabiner & Juang, 1986) up to renormalization (Glasser et al., 2019). Given a total ordering of variables $X_1, \ldots, X_d$, it is known we can compile an HMM into an equivalent structured decomposable circuit (Vergari et al., 2019b), which has the same structure of the tensorized circuit encoding an MPS showed in Fig. 8. Therefore, we can represent an HMM/$\mathrm{MPS}_{\mathbb{R}\geqslant 0}$ in our circuit construction pipeline by parameterizing a linear-tree region graph (called LT in this paper) with $\mathrm{CP}^\top$ layers.

**Born machines (BM)** (Han et al., 2018) **and Tensor-Train Density Estimators (TTDE)** (Novikov et al., 2021) are probabilistic models used to estimate probability mass functions and probability density functions, respectively. They are obtained by efficiently squaring an MPS, which is a structured decomposable tensorized circuit as for Proposition 3. Note that such a tensorized circuit can be obtained using the same region graph and tensorized layer used to construct a non-negative MPS, but instead just relax the non-negativity assumption over its parameters. It is known that squaring a MPS (resp. a structured decomposable tensorized circuit) yields a BM (resp. another structured decomposable tensorized circuit having the same layers but with a quadratic width increase). See e.g. Proposition 3 in Loconte et al. (2024). Therefore, BMs and TTDEs can be retrieved through our circuit construction pipeline by overaparameterizing a linear-tree region graph (LT) with $\mathrm{CP}^\top$ layers, followed by efficiently squaring the resulting circuit (Vergari et al., 2021).

**Squared non-monotonic PCs** ($\mathrm{NPC}^2$) (Loconte et al., 2024) are generalizations of BMs and TTDEs that also include the squaring of tensorized circuits obtained by overparameterizing a random binary tree region graph (as in RAT-SPNs above), as well as using Tucker layers instead of $\mathrm{CP}^\top$ layers. Furthermore, the original implementation of $\mathrm{NPC}^2$ allows circuits to be folded.

**Tree Tensor Networks (TTNs)** (Cheng et al., 2019) are tree-shaped hierarchical tensor factorizations represented through the tensor network formalism. TTNs factorizations are equivalent to hierarchical Tucker, but one choose a particular structure based on the data distribution being modelled. For image data, Cheng et al. (2019) proposed a TTN structure obtained by recursively splitting an image in half, alternating horizontal and vertical splits. This structure is analogous to our quad tree region graph (QT), but allowing splitting image patches in just two parts (rather than four).

## C    Sampling

In Algorithm C.1, we interpret the entries of each non-negative parameter matrix $\mathbf{W}^{S \times K}$ in $c$ as the parameters of categorical distributions associated to $S$ latent variables, each taking values in $\{1, \dots, K\}$. Note that we can always normalize a PC s.t. its normalization constant is equal to 1 thus yielding parameter matrices that sum up to 1 along every row, as detailed in (Peharz et al., 2015). Then, sampling a data point $\mathbf{x}$ translates to iteratively sampling from such latent variables (see L8-13 of the algorithm) according to the hierarchical structure of the circuit, i.e. following a topological order like a breadth first search (BFS). Note that sampling the latent variables corresponding to a sum layer corresponds to choosing (i) a selection of the input layers on which recursively continue sampling, and (ii) a particular computational unit within each selected layer. The information (i) and (ii) for each layer is stored in dictionaries (see L1-4). Due to decomposability (Def. 8), sampling from a product layer $\boldsymbol{\ell}$ translates to choosing a selection of the input computational units, as they will be defined on different variables. Unlike sum layers where we sample from Categoricals to select such units, in product layers they are unequivocally determined by which product unit of $\boldsymbol{\ell}$ has been selected previously and whether $\boldsymbol{\ell}$ is an Kronecker or Hadamard layer (see L14-20). We sample all sum and product layers as explained below. Finally, it remains to sample from the input layers and assign values to the variables the PC is defined on. We sample from an input layer $\boldsymbol{\ell}$ when at least one input units within $\boldsymbol{\ell}$ has been selected by the sampling procedure above for sum and product layers. That is, given $X \in \mathbf{X}$ the variable on which $\boldsymbol{\ell}$ depends on and $n_k$ the $k$-th input unit to sample from, we sample an assignment to $X$ from $n_k$ (see L21-25).

---

**Algorithm C.1** samplingTensorizedPC$(c, N)$

**Input:** A tensorized PC $c$ over $\mathbf{X} = \{X_i\}_{i=1}^D$, a positive integer $N$.
**Output:** Samples $\mathbf{S} \in \mathbb{R}^{N \times D}$ drawn from $c$.
**Assumptions:** (1) $c$ is normalized: all sum layer parameters sum up to 1 over the columns; (2) Each input layer is defined over a single RV; (3) the width of a layer is a multiple of $K$.
**Notes:** (1) All assignments preceded by the symbol $\forall$ can be parallelized; (2) unravel-index is the homonymous numpy function but whose indexing starts from 1 instead of 0.

1:  $\mathcal{S} \leftarrow \{\boldsymbol{\ell} : [\,] \mid \forall \boldsymbol{\ell} \in c\} \triangleright$ mapping layers to sample indices
2:  $\mathcal{U} \leftarrow \{\boldsymbol{\ell} : [\,] \mid \forall \boldsymbol{\ell} \in c\} \triangleright$ mapping layers to unit indices
3:  $\mathcal{S}[c] \leftarrow [N]$
4:  $\mathcal{U}[c] \leftarrow \mathbf{1}_N$
5:  **for each** inner layer $\boldsymbol{\ell} \in$ BFS$(c)$ **do**
6:      $\mathcal{L} \leftarrow$ list of the $L$ input layers of $\boldsymbol{\ell}$
7:      **if** $\mathcal{S}[\boldsymbol{\ell}]$ is empty **then skip**
8:      **else if** $\boldsymbol{\ell}$ is a sum layer with $\mathbf{W} \in \mathbb{R}^{K \times KL}$ **then**
9:          $\mathbf{v} \leftarrow$ vector of size $|\mathcal{S}[\boldsymbol{\ell}]|$ with values in $[KL]$
10:         $v_i \leftarrow$ sample from categorical with $\mathsf{p} = \mathbf{w}_{\mathcal{U}[\boldsymbol{\ell}]_i,:}$
11:         idx1, idx2 $\leftarrow$ unravel-index$(\mathbf{v}, (L, K))$
12:         $\forall i \in [L] : \mathcal{S}[\mathcal{L}[i]]$.extend$(\mathcal{S}[\boldsymbol{\ell}][\text{idx1} == i])$
13:             $\mathcal{U}[\mathcal{L}[i]]$.extend$(\text{idx2}[\text{idx1} == i])$
14:     **else if** $\boldsymbol{\ell}$ is a Kronecker prod. layer **then**
15:         idx-list $\leftarrow$ unravel-index$(\mathcal{U}[\boldsymbol{\ell}], (K, )_{i=1}^L)$
16:         $\forall i \in [L] : \mathcal{S}[\mathcal{L}[i]]$.extend$(\mathcal{S}[\boldsymbol{\ell}])$
17:             $\mathcal{U}[\mathcal{L}[i]]$.extend$(\text{idx-list}[k])$
18:     **else if** $\boldsymbol{\ell}$ is a Hadamard prod. layer **then**
19:         $\forall i \in [L] : \mathcal{S}[\mathcal{L}[i]]$.extend$(\mathcal{S}[\boldsymbol{\ell}])$
20:             $\mathcal{U}[\mathcal{L}[i]]$.extend$(\mathcal{U}[\boldsymbol{\ell}])$
21: $\mathbf{S} \leftarrow \mathbb{R}^{N \times D}$
22: **for each** input layer $\boldsymbol{\ell} \in c$ **s.t.** $\mathcal{S}[\ell] \neq [\,]$ **do**
23:     $j \leftarrow$ sc$(\boldsymbol{\ell})$
24:     pairs $\leftarrow$ vstack$(\mathcal{S}[\boldsymbol{\ell}], \mathcal{U}[\boldsymbol{\ell}])$
25:     $\forall (i, k) \in$ pairs $: \mathbf{S}_{ij} \leftarrow$ sample $k$-th unit of $\boldsymbol{\ell}$
26: **return** samples $\mathbf{S}$

---

## D    Region Graphs: Quad-Graphs and Quad-Trees

Algorithm D.1 details the construction of our proposed RGs for image-data: QTs and QGs. The algorithm takes as input the height ($H$) and width ($W$) of the image, and a flag (isTree), which specifies whether to enforce the output RG to be a tree (QT) or not (QG). The algorithm builds a RG in a bottom-up fashion, merging regions associated to smaller patches to bigger patches, starting from the single pixels. Specifically, to build QTs—QT-4s to be precise—we merge regions using Algorithm D.2, whereas for QGs we merge regions using Algorithm D.3.

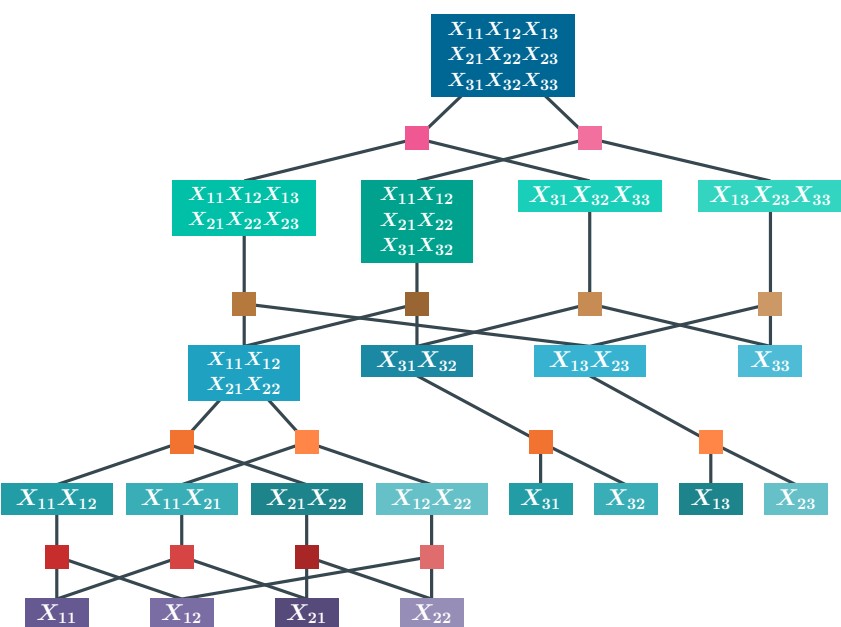

Figure D.1: **The quad graph (QG).** We illustrate the quad graph RG delivered by Algorithm D.1 passing $H = 3, W = 3$ and isTree = False as input arguments. The region graph is *unbalanced* as the image size ($3 \times 3$) is not a power of 2. Differently from our quad trees (QTs), QGs have regions partitioned in more than a single way (e.g., the root region node), and regions can be shared among partitions. For example, in a QT, the top region could only be partitioned in a single way into two or four sub-regions, respectively called QT-2 and QT-4 region graphs.

---

**Algorithm D.1** buildQuadGraph($H, W, \text{isTree}$)

**Input:** Image height $H$, image width $W$, and whether to enforce the output RG to be a tree.
**Output:** a RG over $H \cdot W$ variables.

1: $\mathsf{S} \leftarrow \{\mathbf{Y}_{ij} = \{X_{ij}\} \mid (i,j) \in [H] \times [W]\}$
2: $\mathcal{R} \leftarrow$ a RG having leaf regions $\mathsf{S}$
3: $h \leftarrow H; \quad w \leftarrow W$
4: **while** $h > 1 \vee w > 1$ **do**
5:     $h \leftarrow \lceil h/2 \rceil; \quad w \leftarrow \lceil w/2 \rceil; \quad \mathsf{S}' \leftarrow \varnothing$
6:     **for** $i, j \in [h] \times [w]$ **do**
7:         $\Delta \leftarrow (\{2i-1, 2i\} \times \{2j-1, 2j\}) \bigcap ([H] \times [W])$
8:         **if** $|\Delta| = 1$ **then**
9:             Let $\mathbf{Y}_{pq} \in \mathsf{S}$ s.t. $(p,q) \in \Delta$
10:          addRegion($\mathcal{R}, \mathbf{Y}_{pq}$)
11:         **else if** $|\Delta| = 2$ **then**
12:             Let $\mathbf{Y}_{pq}, \mathbf{Y}_{rs} \in \mathsf{S}$ s.t.
13:               $(p,q), (r,s) \in \Delta, \quad p < r, q < s$
14:          addPartition($\mathcal{R}, \mathbf{Y}_{pq} \cup \mathbf{Y}_{rs}, \{\mathbf{Y}_{pq}, \mathbf{Y}_{rs}\}$)
15:         **else**     ▷ $|\Delta| = 4$
16:             **if** isTree **then** mergeTree($\mathcal{R}, \Delta, \mathsf{S}$)
17:             **else** mergeDAG($\mathcal{R}, \Delta, \mathsf{S}$)
18:         $\mathbf{Y}_{ij} \leftarrow \bigcup_{(r,s) \in \Delta} \mathbf{Y}_{rs}$ s.t. $\mathbf{Y}_{rs} \in \mathsf{S}$
19:         $\mathsf{S}' \leftarrow \mathsf{S}' \cup \{\mathbf{Y}_{ij}\}$
20:     $\mathsf{S} \leftarrow \mathsf{S}'$
21: **return** $\mathcal{R}$

---

**Algorithm D.2** mergeTree($\mathcal{R}, \Delta, \mathsf{S}$)

**Input:** a RG $\mathcal{R}$, a set of four coordinates $\Delta$, and a collection of regions $\mathsf{S}$.
**Behavior:** It merges the regions indexed by $\Delta$ in $\mathcal{R}$ by forming a tree structure.

1: Let $\mathbf{Z}_{uv} = \mathbf{Y}_{p+u\ q+v} \in \mathsf{S}$ s.t.
2:     $(p+u, q+v) \in \Delta, \quad u, v \in \{0, 1\}$
3: $\mathbf{Y} \leftarrow \mathbf{Z}_{00} \cup \mathbf{Z}_{01} \cup \mathbf{Z}_{10} \cup \mathbf{Z}_{11}$
4: addPartition($\mathcal{R}, \mathbf{Y}, \{\mathbf{Z}_{00}, \mathbf{Z}_{01}, \mathbf{Z}_{10}, \mathbf{Z}_{11}\}$)

---

**Algorithm D.3** mergeDAG($\mathcal{R}, \Delta, \mathsf{S}$)

**Input:** a RG $\mathcal{R}$, a set of four coordinates $\Delta$, and a collection of regions $\mathsf{S}$.
**Behavior:** It merges the regions indexed by $\Delta$ in $\mathcal{R}$ by forming a DAG structure.

1: Let $\mathbf{Z}_{uv} = \mathbf{Y}_{p+u\ q+v} \in \mathsf{S}$ s.t.
2:     $(p+u, q+v) \in \Delta, \quad u, v \in \{0, 1\}$
3: $\mathbf{Y} \leftarrow \mathbf{Z}_{00} \cup \mathbf{Z}_{01} \cup \mathbf{Z}_{10} \cup \mathbf{Z}_{11}$
4: addPartition($\mathcal{R}, \mathbf{Y}, \{\mathbf{Z}_{00} \cup \mathbf{Z}_{01}, \mathbf{Z}_{10} \cup \mathbf{Z}_{11}\}$)
5: addPartition($\mathcal{R}, \mathbf{Y}, \{\mathbf{Z}_{00} \cup \mathbf{Z}_{10}, \mathbf{Z}_{01} \cup \mathbf{Z}_{11}\}$)
6: addPartition($\mathcal{R}, \mathbf{Z}_{00} \cup \mathbf{Z}_{01}, \{\mathbf{Z}_{00}, \mathbf{Z}_{01}\}$)
7: addPartition($\mathcal{R}, \mathbf{Z}_{10} \cup \mathbf{Z}_{11}, \{\mathbf{Z}_{10}, \mathbf{Z}_{11}\}$)
8: addPartition($\mathcal{R}, \mathbf{Z}_{00} \cup \mathbf{Z}_{10}, \{\mathbf{Z}_{00}, \mathbf{Z}_{10}\}$)
9: addPartition($\mathcal{R}, \mathbf{Z}_{01} \cup \mathbf{Z}_{11}, \{\mathbf{Z}_{01}, \mathbf{Z}_{11}\}$)

---

We illustrate in Fig. D.1 the resulting QG obtained via Algorithm D.1 with $H = 3, W = 3$ and isTree = False. The QG is unbalanced as $HW$ is not a power of 2.

# E   Additional Results

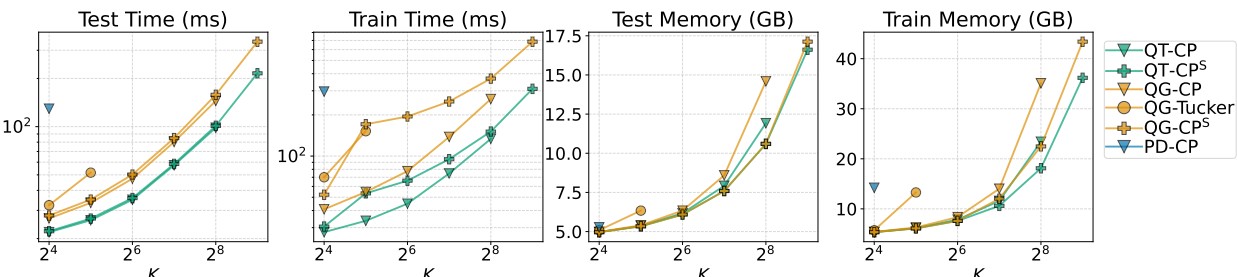

Figure E.1: **Benchmarking the role of RGs and composite layers in tensorized circuits on CelebA.** We report the average time (ms) and GPU memory usage (GiBs) to process a batch of samples for different tensorized architectures—listed in the legend on the right—at different values of $K$ (x-axis). The stats are reported both at test and training time. The benchmark is conducted using the CELEBA dataset with a batch size of 128.

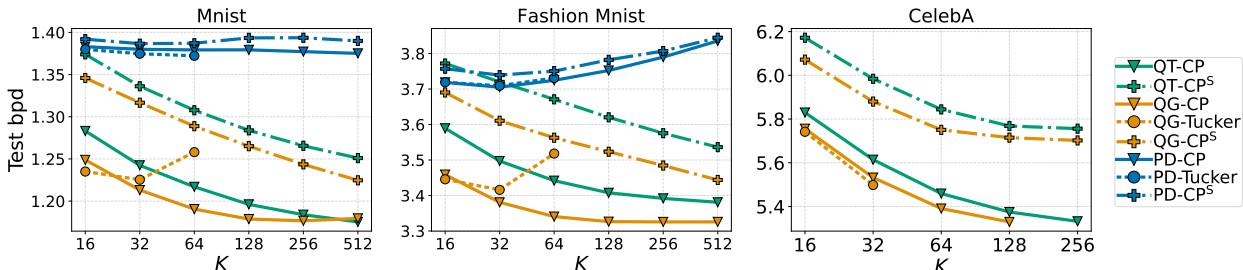

Figure E.2: **Overparameterizing tensorized architectures delivers better performing models when using QTs and QGs, but not when using PDs. Different from Fig. 20, we here learn the mixing layers in QG- and PD-based models.** We report the test-set bpd (y-axis) at different values of $K$ (x-axis) for MNIST (left), FASHIONMNIST (middle) and CELEBA (right) averaged over 5 runs for different tensorized architectures, which we report in the legend on the right. We use a batch size of 256.

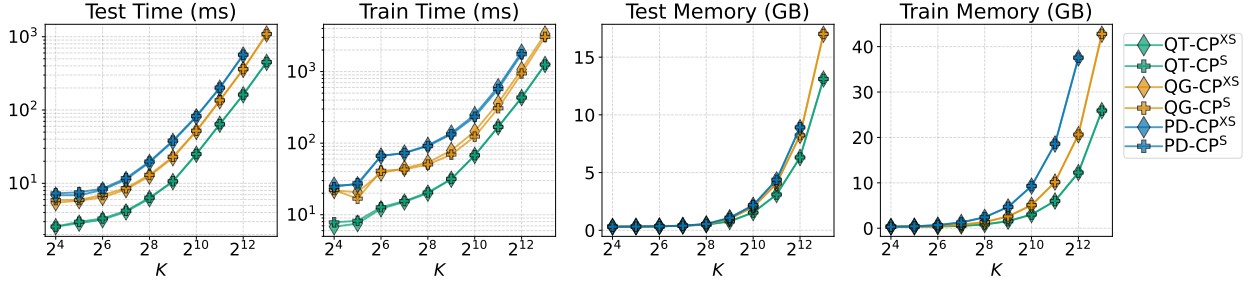

Figure E.3: **For the same choice of RG and $K$, CP$^\mathbf{S}$ and CP$^\mathbf{XS}$ layers require the same time/space resources, with CP$^\mathbf{XS}$ only being slightly faster at training-time.** We report time (ms) and GPU memory usage (GiBs) at different values of $K$ (x-axis) at both test-time and training-time for different tensorized architectures listed in the legend on the right. The benchmark is conducted on MNIST using a batch size of 128.

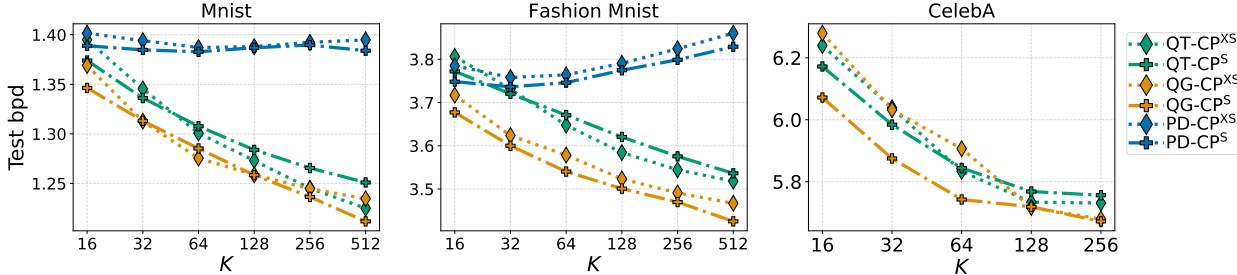

Figure E.4: **CP$^{\text{XS}}$ and CP$^{\text{S}}$ layers are equivalently accurate when used in different tensorized architectures.** We report the test-set bpd (y-axis) averaged over 5 runs for different tensorized architectures—listed in the legend on the right—at different values of $K$ (x-axis). We use the Mnist, FashionMnist and CelebA datasets, and a batch size of 256.

Table E.1: **Mnist distribution estimation results.** Test-set bpd on Mnist averaged over 5 runs for different tensorized PC architectures. We report 3 standard deviations from the mean.

| RG | Learn Mixing-Layer | $K$ | CP | CP$^{\text{XS}}$ | CP$^{\text{S}}$ | Tucker |
|---|---|---|---|---|---|---|
| PD | Yes | 16 | $1.383 \pm 0.008$ | $1.392 \pm 0.008$ | $1.392 \pm 0.007$ | $1.380 \pm 0.006$ |
| | | 32 | $1.380 \pm 0.007$ | $1.387 \pm 0.005$ | $1.387 \pm 0.008$ | $1.375 \pm 0.004$ |
| | | 64 | $1.379 \pm 0.009$ | $1.384 \pm 0.005$ | $1.387 \pm 0.009$ | $1.372 \pm 0.004$ |
| | | 128 | $1.379 \pm 0.003$ | $1.386 \pm 0.006$ | $1.394 \pm 0.006$ | OOM |
| | | 256 | $1.377 \pm 0.005$ | $1.386 \pm 0.008$ | $1.394 \pm 0.009$ | OOM |
| | | 512 | $1.375 \pm 0.009$ | $1.385 \pm 0.007$ | $1.390 \pm 0.011$ | OOM |
| PD | No | 16 | $1.381 \pm 0.007$ | $1.402 \pm 0.008$ | $1.389 \pm 0.006$ | $1.377 \pm 0.005$ |
| | | 32 | $1.381 \pm 0.009$ | $1.394 \pm 0.011$ | $1.385 \pm 0.003$ | $1.377 \pm 0.004$ |
| | | 64 | $1.376 \pm 0.002$ | $1.387 \pm 0.005$ | $1.383 \pm 0.004$ | $1.381 \pm 0.006$ |
| | | 128 | $1.375 \pm 0.003$ | $1.388 \pm 0.004$ | $1.387 \pm 0.003$ | OOM |
| | | 256 | $1.373 \pm 0.005$ | $1.392 \pm 0.006$ | $1.390 \pm 0.009$ | OOM |
| | | 512 | $1.370 \pm 0.002$ | $1.395 \pm 0.014$ | $1.384 \pm 0.008$ | OOM |
| QT | N/A | 16 | $1.283 \pm 0.004$ | $1.395 \pm 0.008$ | $1.374 \pm 0.008$ | N/A |
| | | 32 | $1.242 \pm 0.004$ | $1.345 \pm 0.030$ | $1.336 \pm 0.009$ | N/A |
| | | 64 | $1.217 \pm 0.002$ | $1.301 \pm 0.019$ | $1.308 \pm 0.003$ | N/A |
| | | 128 | $1.196 \pm 0.004$ | $1.273 \pm 0.028$ | $1.284 \pm 0.002$ | N/A |
| | | 256 | $1.184 \pm 0.002$ | $1.245 \pm 0.028$ | $1.266 \pm 0.003$ | N/A |
| | | 512 | $1.175 \pm 0.001$ | $1.225 \pm 0.010$ | $1.251 \pm 0.002$ | N/A |
| QG | Yes | 16 | $1.249 \pm 0.004$ | $1.375 \pm 0.014$ | $1.346 \pm 0.010$ | $1.235 \pm 0.012$ |
| | | 32 | $1.213 \pm 0.003$ | $1.334 \pm 0.010$ | $1.317 \pm 0.004$ | $1.225 \pm 0.011$ |
| | | 64 | $1.190 \pm 0.003$ | $1.280 \pm 0.017$ | $1.289 \pm 0.003$ | $1.258 \pm 0.005$ |
| | | 128 | $1.179 \pm 0.001$ | $1.240 \pm 0.015$ | $1.265 \pm 0.004$ | OOM |
| | | 256 | $1.177 \pm 0.004$ | $1.218 \pm 0.021$ | $1.244 \pm 0.003$ | OOM |
| | | 512 | $1.180 \pm 0.009$ | $1.205 \pm 0.011$ | $1.225 \pm 0.004$ | OOM |
| QG | No | 16 | $1.248 \pm 0.003$ | $1.369 \pm 0.039$ | $1.346 \pm 0.004$ | $1.233 \pm 0.004$ |
| | | 32 | $1.212 \pm 0.003$ | $1.313 \pm 0.027$ | $1.313 \pm 0.006$ | $1.222 \pm 0.004$ |
| | | 64 | $1.185 \pm 0.002$ | $1.276 \pm 0.010$ | $1.285 \pm 0.006$ | $1.257 \pm 0.005$ |
| | | 128 | $1.171 \pm 0.002$ | $1.259 \pm 0.011$ | $1.258 \pm 0.004$ | OOM |
| | | 256 | $1.173 \pm 0.009$ | $1.245 \pm 0.009$ | $1.236 \pm 0.002$ | OOM |
| | | 512 | $1.177 \pm 0.006$ | $1.235 \pm 0.010$ | $1.212 \pm 0.010$ | OOM |

Table E.2: **FashionMnist distribution estimation results.** Test-set bpd on FASHIONMNIST averaged over 5 runs for different tensorized PC architectures. We report 3 standard deviations from the mean.

| RG | Learn Mixing-Layer | $K$ | CP | CP$^{\text{XS}}$ | CP$^{\text{S}}$ | Tucker |
|---|---|---|---|---|---|---|
| PD | Yes | 16 | $3.719 \pm 0.014$ | $3.757 \pm 0.008$ | $3.757 \pm 0.011$ | $3.719 \pm 0.015$ |
| | | 32 | $3.705 \pm 0.012$ | $3.738 \pm 0.011$ | $3.739 \pm 0.005$ | $3.709 \pm 0.004$ |
| | | 64 | $3.725 \pm 0.011$ | $3.749 \pm 0.009$ | $3.750 \pm 0.007$ | $3.731 \pm 0.014$ |
| | | 128 | $3.752 \pm 0.005$ | $3.774 \pm 0.009$ | $3.782 \pm 0.005$ | OOM |
| | | 256 | $3.790 \pm 0.011$ | $3.801 \pm 0.013$ | $3.807 \pm 0.018$ | OOM |
| | | 512 | $3.836 \pm 0.019$ | $3.836 \pm 0.024$ | $3.845 \pm 0.017$ | OOM |
| PD | No | 16 | $3.715 \pm 0.004$ | $3.785 \pm 0.010$ | $3.748 \pm 0.011$ | $3.716 \pm 0.007$ |
| | | 32 | $3.700 \pm 0.017$ | $3.758 \pm 0.009$ | $3.736 \pm 0.005$ | $3.709 \pm 0.004$ |
| | | 64 | $3.721 \pm 0.011$ | $3.764 \pm 0.012$ | $3.746 \pm 0.011$ | $3.736 \pm 0.006$ |
| | | 128 | $3.752 \pm 0.012$ | $3.791 \pm 0.007$ | $3.775 \pm 0.010$ | OOM |
| | | 256 | $3.779 \pm 0.012$ | $3.824 \pm 0.006$ | $3.799 \pm 0.014$ | OOM |
| | | 512 | $3.814 \pm 0.012$ | $3.860 \pm 0.024$ | $3.829 \pm 0.015$ | OOM |
| QT | N/A | 16 | $3.589 \pm 0.005$ | $3.806 \pm 0.042$ | $3.772 \pm 0.031$ | N/A |
| | | 32 | $3.497 \pm 0.003$ | $3.731 \pm 0.032$ | $3.720 \pm 0.007$ | N/A |
| | | 64 | $3.442 \pm 0.003$ | $3.648 \pm 0.019$ | $3.671 \pm 0.005$ | N/A |
| | | 128 | $3.408 \pm 0.003$ | $3.584 \pm 0.011$ | $3.620 \pm 0.009$ | N/A |
| | | 256 | $3.392 \pm 0.001$ | $3.544 \pm 0.014$ | $3.576 \pm 0.013$ | N/A |
| | | 512 | $3.381 \pm 0.002$ | $3.518 \pm 0.018$ | $3.536 \pm 0.007$ | N/A |
| QG | Yes | 16 | $3.459 \pm 0.004$ | $3.741 \pm 0.030$ | $3.690 \pm 0.019$ | $3.446 \pm 0.004$ |
| | | 32 | $3.381 \pm 0.002$ | $3.635 \pm 0.026$ | $3.611 \pm 0.016$ | $3.416 \pm 0.006$ |
| | | 64 | $3.341 \pm 0.004$ | $3.555 \pm 0.020$ | $3.563 \pm 0.020$ | $3.518 \pm 0.012$ |
| | | 128 | $3.326 \pm 0.002$ | $3.487 \pm 0.018$ | $3.523 \pm 0.006$ | OOM |
| | | 256 | $3.326 \pm 0.003$ | $3.449 \pm 0.018$ | $3.484 \pm 0.004$ | OOM |
| | | 512 | $3.326 \pm 0.004$ | $3.409 \pm 0.011$ | $3.444 \pm 0.009$ | OOM |
| QG | No | 16 | $3.464 \pm 0.005$ | $3.717 \pm 0.051$ | $3.677 \pm 0.031$ | $3.446 \pm 0.008$ |
| | | 32 | $3.385 \pm 0.004$ | $3.624 \pm 0.051$ | $3.600 \pm 0.011$ | $3.417 \pm 0.005$ |
| | | 64 | $3.339 \pm 0.004$ | $3.578 \pm 0.032$ | $3.540 \pm 0.009$ | $3.499 \pm 0.006$ |
| | | 128 | $3.319 \pm 0.004$ | $3.523 \pm 0.036$ | $3.501 \pm 0.017$ | OOM |
| | | 256 | $3.317 \pm 0.002$ | $3.491 \pm 0.013$ | $3.470 \pm 0.005$ | OOM |
| | | 512 | $3.317 \pm 0.005$ | $3.467 \pm 0.032$ | $3.425 \pm 0.010$ | OOM |

Table E.3: **CelebA distribution estimation results (using RGB values).** Test-set bpd on CELEBA averaged over 3 runs for different tensorized PC architectures. We report 3 standard deviations from the mean.

| RG | Learn Mixing-Layer | $K$ | CP | $\text{CP}^{\text{XS}}$ | $\text{CP}^{\text{S}}$ | Tucker |
|---|---|---|---|---|---|---|
| QT | N/A | 16 | $5.828 \pm 0.008$ | $6.237 \pm 0.026$ | $6.171 \pm 0.006$ | N/A |
| | | 32 | $5.612 \pm 0.012$ | $6.024 \pm 0.032$ | $5.981 \pm 0.007$ | N/A |
| | | 64 | $5.457 \pm 0.010$ | $5.831 \pm 0.022$ | $5.843 \pm 0.017$ | N/A |
| | | 128 | $5.374 \pm 0.002$ | $5.732 \pm 0.044$ | $5.766 \pm 0.022$ | N/A |
| | | 256 | $5.332 \pm 0.002$ | $5.739 \pm 0.037$ | $5.753 \pm 0.014$ | N/A |
| QG | Yes | 16 | $5.756$ | $6.161$ | $6.072$ | $5.742$ |
| | | 32 | $5.532$ | $5.960$ | $5.880$ | $5.498$ |
| | | 64 | $5.391$ | $5.816$ | $5.751$ | OOM |
| | | 128 | $5.329$ | $5.771$ | $5.715$ | OOM |
| | | 256 | OOM | $5.731$ | $5.702$ | OOM |
| QG | No | 16 | $5.755 \pm 0.010$ | $6.292 \pm 0.037$ | $6.069 \pm 0.006$ | $5.738 \pm 0.011$ |
| | | 32 | $5.528 \pm 0.023$ | $6.056 \pm 0.072$ | $5.875 \pm 0.016$ | $5.494 \pm 0.023$ |
| | | 64 | $5.392 \pm 0.026$ | $5.906 \pm 0.052$ | $5.746 \pm 0.010$ | OOM |
| | | 128 | $5.335 \pm 0.027$ | $5.742 \pm 0.067$ | $5.725 \pm 0.039$ | OOM |
| | | 256 | OOM | $5.691 \pm 0.034$ | $5.667 \pm 0.014$ | OOM |

Table E.4: **CelebA distribution estimation results using lossless YCoCg transform.** Test-set bpd on CELEBA over 1 single run for different tensorized PC architectures. We note how performance are consistently better than those in Table E.3, confirming that using the YCoCg transform helps. Note that results in this table are directly comparable with those in Table E.3 because the transformation used is lossless (and operates on discrete data, hence does not require a correction by the log-determinant).

| RG | Learn Mixing-Layer | $K$ | CP | $\text{CP}^{\text{XS}}$ | $\text{CP}^{\text{S}}$ | Tucker |
|---|---|---|---|---|---|---|
| QT | N/A | 16 | $5.604$ | $5.770$ | $5.831$ | N/A |
| | | 32 | $5.447$ | $5.656$ | $5.648$ | N/A |
| | | 64 | $5.321$ | $5.584$ | $5.589$ | N/A |
| | | 128 | $5.248$ | $5.570$ | $5.549$ | N/A |
| | | 256 | $5.238$ | $5.522$ | $5.548$ | N/A |
| QG | No | 16 | $5.541$ | $5.840$ | $5.757$ | $5.541$ |
| | | 32 | $5.383$ | $5.660$ | $5.622$ | $5.383$ |
| | | 64 | $5.273$ | $5.544$ | $5.510$ | OOM |
| | | 128 | $5.205$ | $5.536$ | $5.500$ | OOM |
| | | 256 | OOM | $5.579$ | $5.489$ | OOM |

### E.1    Results on UCI Tabular Datasets

|        |    | Number of samples | | |
| --- | --- | --- | --- | --- |
|        | $D$ | train | validation | test |
| Power | 6 | 1,659,917 | 184,435 | 204,928 |
| Gas | 8 | 852,174 | 94,685 | 105,206 |
| Hepmass | 21 | 315,123 | 35,013 | 174,987 |
| MiniBooNE | 43 | 29,556 | 3,284 | 3,648 |
| BSDS300 | 63 | 1,000,000 | 50,000 | 250,000 |

Table E.5: **UCI dataset statistics.** Dimensionality $D$ and number of samples of each dataset split after the preprocessing by Papamakarios et al. (2017).

**Density estimation on tabular datasets.**    Following Papamakarios et al. (2017), we evaluate our tensorized architectures for density estimation on five tabular datasets. For each dataset, we randomly construct 8 binary tree region graphs (cf. Section 4.1), and build a mixture of tensorized PCs based of them. Specifically, following our mix-and-match approach Table 1, we build RND-CP and RND-Tucker architectures which we run for several model sizes $K$ and learning rates (see below). Differently from images, all these datasets contain continuous features, which we model using input layers encoding Gaussian likelihoods. We train all PCs for up to 1000 epochs or until convergence, using Adam as optimizer and 512 as batch size. Furthermore, we perform the experiments using three different learning rates: $10^{-3}$, $5 \cdot 10^{-3}$, and $10^{-2}$, and report the best results according to the validation set log-likelihood.

**Results.**    Fig. E.5 reports the best results from our models, where we see that Tucker layers outperform CP layers on the two lowest dimensional datasets – Power and Gas – which also have the highest number of training data points (see Table E.5). On the other hand, CP-based architectures outperform Tucker-based ones on the other three datasets (Hepmass, MiniBooNE and BSDS300), even though the latter have a much higher number of trainable parameters then the former for a fixed $K$ (i.e., $K^2$ for CP while $K^3$ for Tucker). Our results suggest that the more aggressive over-parameterization of Tucker layers lead to a more difficult optimization for high-dimensional datasets and thus for deeper tensorized PCs.

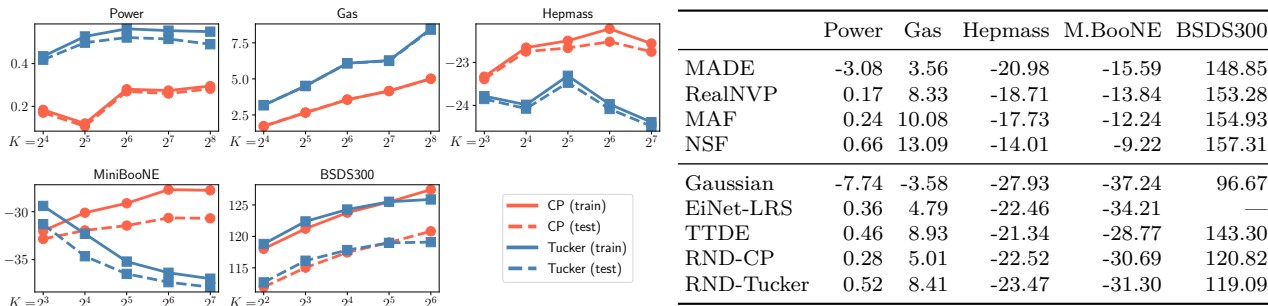

|  | Power | Gas | Hepmass | M.BooNE | BSDS300 |
| --- | --- | --- | --- | --- | --- |
| MADE | -3.08 | 3.56 | -20.98 | -15.59 | 148.85 |
| RealNVP | 0.17 | 8.33 | -18.71 | -13.84 | 153.28 |
| MAF | 0.24 | 10.08 | -17.73 | -12.24 | 154.93 |
| NSF | 0.66 | 13.09 | -14.01 | -9.22 | 157.31 |
| Gaussian | -7.74 | -3.58 | -27.93 | -37.24 | 96.67 |
| EiNet-LRS | 0.36 | 4.79 | -22.46 | -34.21 | — |
| TTDE | 0.46 | 8.93 | -21.34 | -28.77 | 143.30 |
| RND-CP | 0.28 | 5.01 | -22.52 | -30.69 | 120.82 |
| RND-Tucker | 0.52 | 8.41 | -23.47 | -31.30 | 119.09 |

Figure E.5: **Tucker layers are harder to scale than CP layers on high-dimensional UCI datasets.** The **right table** is the one reported in Table 3. The **left plots** show the train and test log-likelihoods of our architectures as the size $K$ of the layers increases. We observe that increasing $K$ is generally beneficial for CP layers in all UCI datasets (left). However, increasing $K$ in Tucker layers can decrease performances for higher-dimensional datasets, as shown for the cases of Hepmass and MiniBooNE. The left plots showing the train set log-likelihoods (dotted lines) are evidence that the decrease of performances of tensorized PCs with Tucker layers is not due to overfitting.

# F  How to implement (folded) layers?

In this section, we provide pytorch snippets to implement a folded Categorical input layer, as well as all the folded sum-product layers we introduced, i.e. Tucker (Eq. (Tucker-layer)), CP (Eq. (CP-layer)), $CP^\top$ (Eq. ($CP^\top$-layer)), $CP^S$ and $CP^{XS}$ (Eq. ($CP^S$-layer)). Our tensorized circuit architectures are nothing more than a sequential application of such layers. We will release our code upon acceptance.

```
1  def categorical_layer(batch, param):
2      """Evaluates a folded Categorical layer with C categories.
3
4      :param batch: Input batch (num_folds, num_channels, batch_size)
5      :param param: Raw parameters (num_folds, K, num_channels, C)
6      :return log_prob: Output log prob (num_folds, K, batch_size)
7      """
8      cat_log_prob = param.log_softmax(-1)
9      idx_fold = torch.arange(batch.size(0))[:, None, None]
10     idx_channel = torch.arange(batch.size(1))[None, :, None]
11     log_prob = cat_log_prob[idx_fold, :, idx_channel, batch].sum(dim=1)
12     return log_prob.transpose(-1, -2)
```

Figure F.1: **Pytorch snippet for a folded categorical layer.** The snippet details the evaluation of a folded categorical layer, which can be used for any type of categorical variable. For instance, it can be used to evaluate the log-likelihood of pixels in image modeling as well as tokens in language modeling. Note how the raw input parameters (`param`) undergo a log-softmax reparameterization (Section 3.2) so as to model valid log-probabilities for many categorical distributions.

```
1  def tucker_layer(log_prob, W):
2      """Evaluates a folded Tucker layer with arity 2 via log-sum-exp.
3
4      :param log_prob: Input log prob (num_folds, 2, K, batch_size)
5      :param W: Layer weights in prob domain (num_folds, K, K, O)
6      :return out_log_prob: Output log prob (num_folds, O, batch_size)
7      """
8      max_log_prob = log_prob.max(dim=2, keepdim=True).values
9      exp_log_prob = torch.exp(log_prob - max_log_prob)
10     out_log_prob = max_log_prob.sum(dim=1) + torch.einsum(
11         "fib,fjb,fijo->fob",
12         exp_log_prob[:, 0], exp_log_prob[:, 1], W).log()
13     return out_log_prob
```

Figure F.2: **Pytorch snippet for a folded Tucker layer.** A folded Tucker layer $\boldsymbol{\ell}$ is parameterized by a tensor $\boldsymbol{\mathcal{W}}$ of shape $F \times O \times K^2$, and computes $F$ Tucker layer (Eq. (Tucker-layer)) $\{\boldsymbol{\ell}^{(n)}\}_{n=1}^F$ *in parallel*. Specifically, the layer $\boldsymbol{\ell}$ computes

$$\boldsymbol{\ell}\left(\bigcup_{n=1}^F \mathbf{Y}^{(n)}\right)_{n:} = \mathbf{W}_{n::}\left[\boldsymbol{\ell}_1\left(\bigcup_{n=1}^F \mathbf{Z}_1^{(n)}\right)_{n:} \otimes \boldsymbol{\ell}_2\left(\bigcup_{n=1}^F \mathbf{Z}_2^{(n)}\right)_{n:}\right],$$

where $\boldsymbol{\ell}_1$ (resp. $\boldsymbol{\ell}_2$) denotes a folded layer computing the $F$ left (resp. right) inputs to $\boldsymbol{\ell}^{(n)}$, each defined over variables $\mathbf{Z}_1^{(n)}$ (resp. $\mathbf{Z}_2^{(n)}$), and $\mathbf{W}_{n::} \in \mathbb{R}^{O \times K^2}$ is the parameter matrix of $\boldsymbol{\ell}^{(n)}$. Note that the snippet shapes the tensor $\boldsymbol{\mathcal{W}}$ as $F \times K \times K \times O$ for convenience with the einsum operation.

```python
def cp_layer(log_prob, W1, W2):
    """Evaluates a folded CP layer with arity 2 via log-sum-exp.

    :param log_prob: Input log probs, (num_folds, 2, K, batch_size)
    :param W1: Folded layer weights, (num_folds, K, O)
    :param W2: Folded layer weights, (num_folds, K, O)
    :return out_log_prob: Output log probs, (num_folds, O, batch_size)
    """
    max_log_prob = log_prob.max(dim=2, keepdim=True).values
    exp_log_prob = torch.exp(log_prob - max_log_prob)
    out_log_prob = max_log_prob.sum(dim=1) + torch.einsum(
        "fib,fjb,fio,fjo->fob",
        exp_log_prob[:, 0], exp_log_prob[:, 1], W1, W2).log()
    return out_log_prob
```

Figure F.3: **Pytorch snippet for a folded CP layer.** A folded CP layer $\boldsymbol{\ell}$ is parameterized by two equally-sized tensors $\mathcal{W}^{(1)}$ and $\mathcal{W}^{(2)}$ of shape $F \times O \times K$, and computes $F$ CP layer (Eq. (CP-layer)) $\{\boldsymbol{\ell}^{(n)}\}_{n=1}^F$ *in parallel.* Specifically, the layer $\boldsymbol{\ell}$ computes

$$\boldsymbol{\ell}\left(\bigcup_{n=1}^F \mathbf{Y}^{(n)}\right)_{n:} = \left(\mathbf{W}_{n::}^{(1)} \boldsymbol{\ell}_1\left(\bigcup_{n=1}^F \mathbf{Z}_1^{(n)}\right)_{n:}\right) \odot \left(\mathbf{W}_{n::}^{(2)} \boldsymbol{\ell}_2\left(\bigcup_{n=1}^F \mathbf{Z}_2^{(n)}\right)_{n:}\right),$$

where $\boldsymbol{\ell}_1$ (resp. $\boldsymbol{\ell}_2$) denotes a folded layer computing the $F$ left (resp. right) inputs to $\boldsymbol{\ell}^{(n)}$, each defined over variables $\mathbf{Z}_1^{(n)}$ (resp. $\mathbf{Z}_2^{(n)}$), and $\mathbf{W}_{n::}^{(1)}, \mathbf{W}_{n::}^{(2)} \in \mathbb{R}^{O \times K}$ are the parameter matrices of $\boldsymbol{\ell}^{(n)}$.

```python
def cpt_layer(log_prob, W):
    """Evaluates a folded CP-T layer with arity H via log-sum-exp.

    :param log_prob: Input log prob (num_folds, H, K, batch_size)
    :param W: Folded layer weights (num_folds, K, O)
    :return out_log_prob: Output log prob (num_folds, O, batch_size)
    """
    log_prob = log_prob.sum(dim=1, keepdim=True)
    max_log_prob = log_prob.max(dim=2, keepdim=True).values
    exp_log_prob = torch.exp(log_prob - max_log_prob)
    out_log_prob = max_log_prob.sum(dim=1) + torch.einsum(
        "fib,fio->fob",
        exp_log_prob[:, 0], W).log()
    return out_log_prob
```

Figure F.4: **Pytorch snippet for a folded $\mathrm{CP}^\top$ layer.** A folded $\mathrm{CP}^\top$ layer $\boldsymbol{\ell}$ is parameterized by tensor $\mathcal{W}$ of shape $F \times O \times K$, and computes $F$ $\mathrm{CP}^\top$ layer (Eq. ($\mathrm{CP}^\top$-layer)) $\{\boldsymbol{\ell}^{(n)}\}_{n=1}^F$ *in parallel.* Specifically, the layer $\boldsymbol{\ell}$ computes

$$\boldsymbol{\ell}\left(\bigcup_{n=1}^F \mathbf{Y}^{(n)}\right)_{n:} = \mathbf{W}_{n::}\left(\boldsymbol{\ell}_1\left(\bigcup_{n=1}^F \mathbf{Z}_1^{(n)}\right)_{n:} \odot \boldsymbol{\ell}_2\left(\bigcup_{n=1}^F \mathbf{Z}_2^{(n)}\right)_{n:}\right),$$

where $\boldsymbol{\ell}_1$ (resp. $\boldsymbol{\ell}_2$) denotes a folded layer computing the $F$ left (resp. right) inputs to $\boldsymbol{\ell}^{(n)}$, each defined over variables $\mathbf{Z}_1^{(n)}$ (resp. $\mathbf{Z}_2^{(n)}$), and $\mathbf{W}_{n::} \in \mathbb{R}^{O \times K}$ are the parameter matrices of $\boldsymbol{\ell}^{(n)}$.

```python
def cpxs_layer(log_prob, W1, W2, D=None):
    """Evaluates a CP-XS layer with arity 2 via log-sum-exp.
       If D is None then evaluates a CP-S layer.

    :param log_prob: Input log probs, (num_folds, 2, K, batch_size)
    :param W1: Folded layer weights, (K, O)
    :param W2: Folded layer weights, (K, O)
    :param D: Optional weights (num_folds, O)
    :return out_log_prob: Output log probs, (num_folds, O, batch_size)
    """
    max_log_prob = log_prob.max(dim=2, keepdim=True).values
    exp_log_prob = torch.exp(log_prob - max_log_prob)
    out_log_prob = max_log_prob.sum(dim=1) + torch.einsum(
        "fib,fjb,io,jo->fob",
        exp_log_prob[:, 0], exp_log_prob[:, 1], W1, W2).log()
    if D:
        out_log_prob += D.log().unsqueeze(-1)
    return out_log_prob
```

Figure F.5: **Pytorch snippet for a CP$^S$ layer.** A CP$^S$ layer $\boldsymbol{\ell}$ is parameterized by two equally-sized matrices $\mathbf{W}^{(1)}$ and $\mathbf{W}^{(2)}$ of shape $O \times K$, and a matrix $\mathbf{D}$ of shape $F \times O$. The layer computes $F$ CP layer (Eq. (CP$^S$-layer)) $\{\boldsymbol{\ell}^{(n)}\}_{n=1}^F$ *in parallel*, each parameterized by $\mathbf{W}^{(1)}$ and $\mathbf{W}^{(2)}$, and then weights their outputs by $\mathbf{D}$. Specifically, the layer $\boldsymbol{\ell}$ computes

$$\boldsymbol{\ell}\left(\bigcup_{n=1}^F \mathbf{Y}^{(n)}\right)_{n:} = \mathbf{d}_{n:} \odot \left(\mathbf{W}^{(1)} \boldsymbol{\ell}_1 \left(\bigcup_{n=1}^F \mathbf{Z}_1^{(n)}\right)_{n:}\right) \odot \left(\mathbf{W}^{(2)} \boldsymbol{\ell}_2 \left(\bigcup_{n=1}^F \mathbf{Z}_2^{(n)}\right)_{n:}\right),$$

where $\boldsymbol{\ell}_1$ (resp. $\boldsymbol{\ell}_2$) denotes a folded layer computing the $F$ left (resp. right) inputs to $\boldsymbol{\ell}^{(n)}$, each defined over variables $\mathbf{Z}_1^{(n)}$ (resp. $\mathbf{Z}_2^{(n)}$), $\mathbf{W}^{(1)}, \mathbf{W}^{(2)} \in \mathbb{R}^{O \times K}$ are the parameter matrices of $\boldsymbol{\ell}^{(n)}$, and $\mathbf{D}$ as a folded-dependent parameter matrix. When $\mathbf{D} = \mathbf{1}$ we retrieve CP$^{XS}$.

