# OpenReview forum: "What is the Relationship between Tensor Factorizations and Circuits (and How Can We Exploit it)?"
_TMLR — Accepted by TMLR_

### Review · Reviewer_uaDR · 2024-10-22

**Summary Of Contributions:**

1. This paper extends tensor factorization methods and their hierarchical formulations into the framework of circuits.
2. Numerous demonstration graphs are provided, clearly illustrating the concepts of tensor factorization and circuit representation.
3. The method proposed in this paper is shown to be faster, more memory-efficient, and capable of strong performance compared to other approaches on datasets such as MNIST and CelebA.

**Audience:**

Yes

**Broader Impact Concerns:**

NA. The tensor factorization problem is a fundamental problem and is really important in various research areas including computer vision, large-scale optimization and deep learning such as video generation.

**Claims And Evidence:**

Yes

**Requested Changes:**

1. Provide examples that illustrate the relationship between the current circuit-based approach and the issue of non-uniqueness in tensor decomposition. Specifically, include a demonstration (possibly in the appendix) that shows how the decomposition of a tensor differs across various methods.

**Strengths And Weaknesses:**

Strengths:
1. The numerous examples provided make the study of tensor factorization quite clear and easy to follow.
2. The method demonstrates efficiency in both speed and GPU usage, showcasing the effectiveness of NN-CP factorization.

Weaknesses:
1. The direct benefit of framing tensor factorization in the language of circuits is not clear, and the paper lacks significant new theoretical contributions.
2. The discussion is limited to the "linear" setting, with the high-order tensor considered purely in terms of decomposition. The deeper implications and practical usage of this approach are not well explored. Is there any possibility of generalizing this method to incorporate nonlinear activation? Some literature suggests that nonlinear approximations can be more efficient than their linear counterparts.
3. A well-known issue with tensor factorization is the non-uniqueness of the factorization. A natural question arises: does this method suffer from the same problem? (Maybe add a bit ablation study by introducing perturbation to discuss the uniqueness) The paper mentions that the methods compared in the empirical experiments use different architectures but fall within the same model class. While this is a challenging question, addressing it would add important context to the study.

---

> ### Author Response · Authors · 2024-11-04
>
> We thank the reviewer for their feedback, and for deeming our paper to be quite clear and easy to follow. Below, we reply to their points and concerns.
>
>
> >the direct benefit of framing tensor factorization in the language of circuits is not clear
>
> We remark that the first benefit by establishing a connection between circuits and tensor factorizations (TFs) is allowing us to connect the literature about the former to the latter, and vice versa. Our contributions to learning circuits (from Section 4 onward) would not be possible without it. From the point of view of “what the tensor factorization community can gain”, we highlight our Opportunity boxes, which offer both **theoretical** and **practical** benefits. This also connects to your question about non-linearities below.
>
> As a summary, here we recall how the circuit perspective now allows researchers in the TF community to automatically devise tractable algorithms to compute quantities of interest in a compositional way (Opportunity 2). Furthermore, it provides a faster sampling algorithm by interpreting non-negative factorizations are latent variable models (Opportunity 3, Appendix C). From a more practical perspective, we present a wide choice of hierarchical factorization architectures developed in the circuit literature (Opportunity 1) that, to the best of our knowledge, are not used in the tensor factorization literature. To further stress this aspect, we have added Figure 6 in the revision, showing how our formalism **allows us to build new factorizations by simply connecting layers**. In addition, we show many possible parameterizations of tensor factorizations as to incorporate non-linearities (Opportunity 4).
>
> If you believe any of these opportunities are not clear, please let us know specifically.
> This would allow us to clarify any misunderstanding and improve the paper.
>
>
> >the paper lacks significant new theoretical contributions.
>
> The theoretical contribution of our paper is drawing a formal connection between circuits and tensor factorizations, which is ``evident’’ in hindsight. The rest of our work has to be intended as a **survey** connecting TFs and circuits. For example, we establish a formal connection among the many different tensorized architectures that have been developed so far (Section 4 and Appendix B).
>
>
> >The discussion is limited to the "linear" setting, with the high-order tensor considered purely in terms of decomposition. Is there any possibility of generalizing this method to incorporate nonlinear activation?
>
> Classical TFs, and therefore their circuit representation, model **multilinear** functions as it evident from Eq. 2, or by looking at the presence of products in the circuit computational graph, which are non-linearities that people sometimes call “multiplicative interactions” (e.g., in deep learning [A]). By generalizing TFs and allowing arbitrary input functions (see Opportunity 4), circuits allow a second form of non-linearity. For instance, in the continuous case, we can have Gaussian likelihoods, splines, or even (deep) generative models as input functions in circuits. We detail this in Opportunity 4.
>
> [A] S. Jayakumar et al. Multiplicative interactions and where to find them. 2020.
>
>
> >A well-known issue with tensor factorization is the non-uniqueness of the factorization. A natural question arises: does this method suffer from the same problem?
>
> As we proved by construction that tensor factorizations can be formulated as circuits, we can answer this immediately (another benefit of our connection): Circuits are non-unique, in the same way a Tucker factorization is not. More in general, in deep machine learning models, the non-uniqueness problem is often expected and unavoidable since loss functions are typically highly non-convex. In our revision, we discuss the fact circuits are non-unique at line L176.
>
>
> >Maybe add a bit ablation study by introducing perturbation to discuss the uniqueness
>
> We gently ask the reviewer to detail how the perturbation should be introduced as to discuss the uniqueness. To our understanding, perturbations can be introduced in different ways: (i) to the parameters at initialization time (we already show this, see below), (ii) to the parameters after optimization.
>
> If the perturbation being referred to is upon initialization, we note that our experiments have been run multiple times with different random initializations of the circuit parameters. Table E.1. shows bits-per-dimensions averaged over 5 independent runs. The three standard deviations we report are in the order of $10^{-3}$, thus suggesting circuits are robust with respect to the initialization noise.
>
> We are willing to execute additional experiments after further clarifications on the setting and the problem being investigated.

---

> > ### Comment · Reviewer_uaDR · 2024-11-29
> >
> > Thank you for the explanations. I have read the replies.

---

### Review · Reviewer_3GRw · 2024-10-22

**Summary Of Contributions:**

The paper studies relationships and connections between circuit representation and tensor factorization approaches to representing, learning, and sampling from probabilistic distributions. Although there have been other works that examined these connections, this current submission appears to be a more comprehensive study of the topic.

Authors also highlight a number of opportunities for further research in the intersection between these areas. Overall, I believe many people in the ML community would find such a survey useful.

**Audience:**

Yes

**Claims And Evidence:**

Yes

**Requested Changes:**

None

**Strengths And Weaknesses:**

Even though the paper is mostly well-written, and arguments are easy to follow, it uses too much heavy notations at points. This might be unavoidable given the topic of the paper.

---

> ### Comment · Action_Editor_EbSN · 2024-10-28
> **Insufficient review**
>
> This review will be flagged. Do you want to offer a more in-depth review? Or shall I assess this review as is. Your status at TMLR may be impacted.

---

### Review · Reviewer_GfCJ · 2024-10-30

**Summary Of Contributions:**

This paper establish a formal connection between tensor factorizations and circuit representations. Within this formal connection, the paper focuses on non-negative tensor factorization and its relations to probabilistic circuits. Furthermore, the paper introduces a "Lego block" modular approach to building tensorized circuit architectures, enabling flexible model construction and tractable exploration of probabilistic models with complex factorization structures.

Key contributions include:
1. Unified Framework for Tensorized Circuits: The paper proposes a general framework that translates tensor factorizations into circuit representations, notably including hierarchical formulations like Tucker factorizations. This framework provides a systematic way to explore and unify various tensor and circuit-based probabilistic models.
2. Parameter-Efficient Architectures: The proposed framework leverages tensor factorizations to improve the parameter efficiency of probabilistic circuits (PCs) by compressing circuit parameters into compact tensor representations.
3. Pipeline for Model Construction: By representing tensor and circuit factorizations as modular components, the authors create a versatile pipeline for assembling tensorized circuit architectures, thus supporting the construction of complex, hierarchically structured models.

Overall, the work presents an exciting new perspective that may open several avenues for future research at the intersection of tensor factorizations and circuit representations, with promising applications in a range of fields where probabilistic modeling is essential.

**Audience:**

Yes

**Broader Impact Concerns:**

The authors might wanna address ethical implications and potential broader impacts, especially given that tensorized circuits are relevant to sensitive applications in autonomous decision-making, and probabilistic modeling for high-stakes domains. Might wanna include a sentence or two about evaluating the fairness and bias of the models after compression.

**Claims And Evidence:**

Yes

**Requested Changes:**

Major changes:

Improve readability by:

1.Move additional related work to the end of the manuscript after the experimental sections (Sections 6 and 7).
2. Move opportunities and Takeaways into Discussion section.

Minor Changes
1. Decrease the mention of applications throughout the text by introducing it into either in introduction or discussion sections for readability.
2. Would be ideal to compare the model beyond image compression. However, given that previous literature seem to focus on image dataset compression I would not penalize the manuscript because of its absence.
3. Include ImageNet into the set of datasets given that HCLTs include it ( (Liu et al., 2022; 2023a)

**Strengths And Weaknesses:**

Strengths
1. Innovative Integration of Two Domains: The framework effectively unifies tensor factorizations and circuits, which could have significant impact on both fields by enabling cross-domain techniques to benefit from advances in the other.
2. Modular and Flexible Model Design: The “Lego block” modularity is a valuable contribution, allowing researchers and practitioners to design and test different architectures systematically and efficiently.
3. Empirical Validation and Efficiency Gains**: The empirical results emphasize the proposed models’ computational efficiency, showing parameter savings while maintaining expressiveness, which could be particularly advantageous in resource-constrained environments.
4.Accessibility of Mathematical Notation: The paper employs extensive and consistent notation, which may make it easier for readers that might not deeply familiar with tensor factorizations or circuits to fully grasp the framework.
5. Broad Application Potential: The framework’s flexibility supports applications in probabilistic modeling, neuro-symbolic AI, and tasks that require reliable and interpretable probabilistic inferences.

Weaknesses and Areas for Improvement

1. Expanded Benchmarking: The empirical evaluations, though thorough, might benefit from additional benchmarks across diverse datasets and tasks, Applications to real-world data (e.g., in text, image, or biological data) would strengthen the impact, especially in domains where tensor factorizations and probabilistic circuits are already in use beyond image dataset compression.
2.Opportunity Boxes: Interesting idea, however, they seem to be better be part of the discussion section of the paper instead of floating in the text. Makes the text read as a review paper instead of a research one.

---

> ### Author Response · Authors · 2024-11-12
>
> We thank the reviewer for providing detailed feedback, and for considering the significant possible impacts of our contributions.
>
>
> >The empirical evaluations, though thorough, might benefit from additional benchmarks across diverse datasets
>
> We performed more experiments on different data sets in our revised version: we added new density estimation experiments on five tabular UCI datasets that include continuous features. There, we investigate the impact of choosing different layers such as CP and Tucker layers. We report the results in Table 3 of the revised paper, where we also compare against a number of baselines, such as normalizing flows and other circuits. In addition, we perform experiments by varying the layer size in Figure E.5 in the appendix. We have found CP layers to perform better and be easier to scale than Tucker layers on the high-dimensional UCI datasets.
>
>
> >Move additional related work to the end of the manuscript after the experimental sections (Sections 6 and 7).
>
> Thank you for the feedback on the presentation. We have moved our Additional Related Work section after the experimental section.
>
>
> >Opportunity Boxes: Interesting idea, however, they seem to be better be part of the discussion section of the paper instead of floating in the text.
>
> We thank the reviewer for their suggestion. However, given the length of the paper, we  believe that moving our opportunity/takeaways boxes in a discussion section would make the contributions of our connections harder to follow: they will be lost in the text. We argue that highlighting them in the text as they are now facilitates readers in pinpointing and retrieving them.
>
>
> >Include ImageNet into the set of datasets given that HCLTs include it ( (Liu et al., 2022; 2023a)
>
> As remarked in our footnote 9, we cannot compare directly to Liu et al. as their results use a *lossy* color space transform that inflates results as it models a not-properly-normalized distribution.  We stress that our experiments on CelebA already fulfil the same role as ImageNet64 in terms of evaluating scaling to a more challenging dataset.

---

### Review · Reviewer_FGZY · 2024-11-27

**Summary Of Contributions:**

This paper makes an exciting and meaningful connection between two areas in machine learning, circuit representations and tensor factorizations, that have traditionally developed independently. By bridging these fields, the authors create a generalized framework that:

- Expands popular tensor factorizations using a circuit-based perspective.
- Offers a modular approach to systematically construct tensorized circuit architectures.
- Establishes a single pipeline for exploring and optimizing these models, bridging gaps in both theory and practice.

The paper combines rigorous theory with comprehensive empirical evaluations, offering clear recommendations for practitioners and opening new perspectives for future research.

**Audience:**

Yes

**Broader Impact Concerns:**

None.

**Claims And Evidence:**

Yes

**Requested Changes:**

I'll recommend acceptance regardless, but please see my suggestions above.

**Strengths And Weaknesses:**

I have a deep appreciation for papers that bridge disparate concepts, and this work is an excellent example of that. I applaud the authors for setting a high standard of quality.

## Strengths

1. **Clarity and Accessibility**
The paper is written with clarity, consistency, and precision. The inclusion of pedagogical examples, paired with beautiful and illustrative figures, further enhances its accessibility.

2. **Thorough Literature Review**
The coverage of related work is remarkably comprehensive. While this makes the paper a long read, it remains highly valuable, especially for readers (like me) who are familiar with only one of the two fields being connected.

3. **Well-Designed Experiments**
The research questions are clearly stated and systematically examined, providing compelling evidence to support the authors’ claims.

4. **Potential for Broader Impact**
I also see potential connections to other diverse areas, such as copulas, multi-marginal optimal transport, and symbolic regression. While these connections are outside the immediate scope of this paper, they represent exciting opportunities for future research that could build on the solid foundation laid here.

## Weaknesses

While this paper is already excellent, I'd suggest the following to make it even stronger:

1. **Practical Guidance for Implementation**
The only thing I found lacking is a few pointers to how one would start exploring these topics. While undoubtedly powerful, the presented methodologically is non-trivial to implement, so it would be helpful to get some more specific recommendations on the practical implementation.

Finally, a minor suggestion would be to clarify that Propositions 1 and 2 are in fact constructive, as shown in the appendix.

---

> ### Author Response · Authors · 2024-12-03
>
> We thank the reviewer for their kind words, and especially for considering the paper excellent and for openly recommending acceptance already.
>
> We agree that some implementation guidelines would be beneficial. As such, we provided pytorch snippets for 5 different folded layers in appendix F, which we reference in Opportunity 7 in the main text. We will also release code upon acceptance.

---

### Comment · Action_Editor_EbSN · 2025-02-11
**malformed**

The author list is malformed. Please look at other examples on TMLR. Also please set MM/YY in header to 02/2025.

---

### Decision · Action_Editor_EbSN · 2025-01-15

**Recommendation:** Accept as is

**Comment:**

The paper provides a novel connection between tensor factorizations and circuits and demonstrates strong theoretical and practical contributions. The authors addressed reviewer concerns thoroughly, improving clarity and experiments.

The reviews strongly support the quality of the paper, with reviewers noting its contributions and addressing areas for improvement. Reviewers agreed that the paper provides novel insights, particularly in establishing a formal connection between tensor factorizations and circuits. This connection was identified as a major strength, enabling advancements in both fields. The modular framework and empirical results further highlight the paper’s practical contributions, which were praised for their clarity and accessibility.

The authors addressed several concerns raised by reviewers. On the issue of non-uniqueness in tensor factorizations, Reviewer uaDR noted this limitation, and the authors responded with a detailed discussion in Section 4 and robust experiments demonstrating stability. Reviewer FGZY suggested adding implementation guidance, which the authors addressed by including PyTorch snippets and committing to release code. To broaden the empirical scope, as requested by Reviewer GfCJ, the authors conducted additional experiments, such as density estimation on UCI datasets, and explained why ImageNet comparisons were omitted.

**Audience:**

This paper will interest researchers working on probabilistic modeling and tensor factorizations. Reviewer FGZY praised its potential to inspire new research. The modular framework provides practical tools for designing circuit-based models.

The “Lego block” approach to building architectures bridges theoretical insights and applications. Reviewer GfCJ highlighted its relevance for resource-efficient probabilistic modeling. Reviewer uaDR emphasized its value in offering a new perspective on tensor factorizations.

**Claims And Evidence:**

The claims in this submission are supported by the theoretical framework, experimental results, and revisions. Reviewers acknowledged the soundness of the work. Reviewer FGZY highlighted the clarity of the theoretical contributions. Pedagogical examples and figures enhance understanding.

The empirical validation is robust. Reviewer GfCJ noted thorough evaluations across datasets. Reviewer uaDR raised concerns about non-uniqueness in tensor factorizations. The authors addressed this by discussing it in Section 4 and Appendix E. They also ran additional experiments showing robustness to initialization noise. The revisions resolved the key concerns raised by reviewers.